

# Hasse diagrams for gapless SPT and SSB phases with non-invertible symmetries

Lakshya Bhardwaj, Daniel Pajer, Sakura Schäfer-Nameki and Alison Warman

Mathematical Institute, University of Oxford,
Woodstock Road, Oxford, OX2 6GG, United Kingdom

## Abstract

We discuss (1+1)d gapless phases with non-invertible global symmetries, also referred to as categorical symmetries. This includes gapless phases showing properties analogous to gapped symmetry protected topological (SPT) phases, known as gapless SPT (or gSPT) phases; and gapless phases showing properties analogous to gapped spontaneous symmetry broken (SSB) phases, that we refer to as gapless SSB (or gSSB) phases. We fit these gapless phases, along with gapped SPT and SSB phases, into a phase diagram describing possible deformations connecting them. This phase diagram is partially ordered and defines a so-called Hasse diagram. Based on these deformations, we identify gapless phases exhibiting symmetry protected criticality, that we refer to as intrinsically gapless SPT (igSPT) and intrinsically gapless SSB (igSSB) phases. This includes the first examples of igSPT and igSSB phases with non-invertible symmetries. Central to this analysis is the Symmetry Topological Field Theory (SymTFT), where each phase corresponds to a condensable algebra in the Drinfeld center of the symmetry category. On a mathematical note, gSPT phases are classified by functors between fusion categories, generalizing the fact that gapped SPT phases are classified by fiber functors; and gSSB phases are classified by functors from fusion to multi-fusion categories. Finally, our framework can be applied to understand gauging of trivially acting non-invertible symmetries, including possible patterns of decomposition arising due to such gaugings.

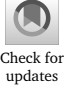
doi:10.21468/SciPostPhys.19.4.113

# 1 Introduction and summary

The well-established Landau paradigm for theories with group-like symmetries states that second order phase transitions are symmetry breaking transitions. Recently, this was extended to include generalized or categorical symmetries, in particular allowing a comprehensive characterisation of gapped phases with categorical symmetries [1–4] using the Symmetry Topological Field Theory (SymTFT) [5–7]. This was extended to include gapless phases, as transitions between gapped phases with fusion category symmetries in [8–12].

In the SymTFT of a $d$-dimensional theory $\mathfrak{T}$ with categorical symmetry $\mathcal{S}$ is a $d + 1$ dimensional gapped theory, defined on an interval with gapped boundary, known as the symmetry boundary $\mathfrak{B}_{\mathcal{S}}^{\text{sym}}$, which encodes the symmetry $\mathcal{S}$. The other boundary is the physical boundary and may or may not be gapped, depending on the properties of the original theory $\mathfrak{T}$.

In (1+1)d the SymTFT description allows for a systematic exploration of all gapped[1] and gapless phases with fusion category symmetry $\mathcal{S}$, by classifying the condensable algebras of the Drinfeld center $\mathcal{Z}(\mathcal{S})$. The latter is the (non-degenerate) braided fusion category formed by the topological defects of the SymTFT. Condensable (or commutative, separable Frobenius) algebras in braided fusion categories were introduced in [13, 14] and play a central role in

---

[1]In this paper, "gapped phases" refers specifically to relativistic, bosonic, oriented quantum systems whose long-distance behaviour is governed by a fully extended TQFT. We assume that the system has a finite fusion-category symmetry $\mathcal{S}$, and is either symmetry-preserving or spontaneously breaks $\mathcal{S}$ to a subcategory $\mathcal{S}'$. Intrinsically topologically ordered (long-range entangled) phases, which arise in higher dimensions, are not treated here, although they are also accessible via the SymTFT framework.

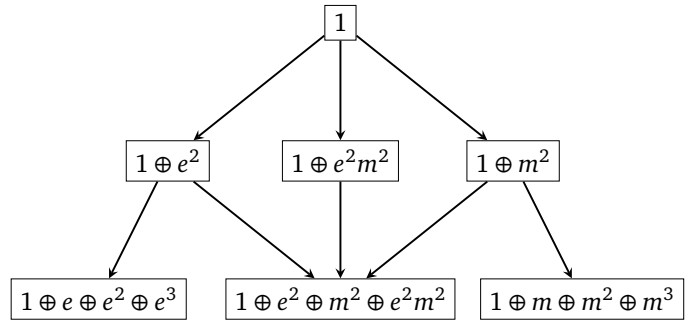

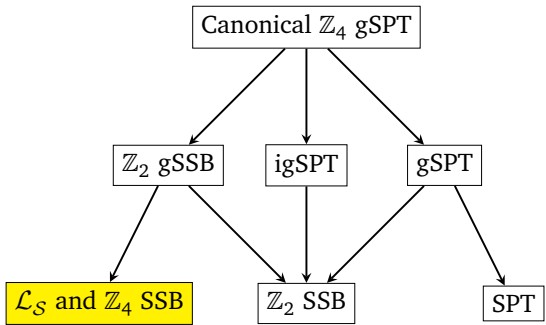

Figure 1: Hasse diagram for $\mathcal{Z}(\mathsf{Vec}_{\mathbb{Z}_4})$: In the top half we show the Hasse diagram of condensable algebras. In each box we show a condensable algebra. The lowest level are the maximal, i.e. Lagrangian, algebras. Picking one of these as the symmetry Lagrangian algebra $\mathcal{L}_\mathcal{S}$ that fixes the symmetry $\mathcal{S}$ allows the classification of all $\mathcal{S}$-symmetric phases (bottom half).

the classification of boundaries and anyon condensations [15, 16]. The symmetry boundary is given in terms of a maximal condensable algebra, i.e. a Lagrangian algebra.[2] In turn, the physical boundary condition is specified by either a Lagrangian algebra (gapped boundary condition) for gapped phases, or a non-maximal condensable algebra for gapless phases.

Every condensable algebra in $\mathcal{Z}(\mathcal{S})$ defines an $\mathcal{S}$-symmetric phase. In turn there is a partial order on the set of condensable algebras, defined as $\mathcal{A}_1 \leq \mathcal{A}_2$ if $\mathcal{A}_1$ is a subalgebra of $\mathcal{A}_2$. This ordering defines a Hasse diagram, which is a graph where each vertex is a condensable algebra and a connection between two vertices is drawn if there is a partial order. This is a directed graph.

The Hasse diagram contains information about the types of phases, e.g. the lowest level are all the gapped phases. The top layer is always the trivial algebra given by the identity line, which corresponds to a canonical gapless phase in which all of symmetry $\mathcal{S}$ acts faithfully in the infrared (IR) and there are no charges for $\mathcal{S}$ missing from the IR, i.e. no charges that are confined. We note here that we employ "generalized charges" in the sense of [17]; every simple object $a \in \mathcal{Z}(\mathcal{S})$ is a topological bulk operator/defect and is simply called a "charge", irrespective of whether in a group-like presentation it would be labelled "electric" or "magnetic".

We can classify phases as spontaneous symmetry breaking (SSB) and symmetry protected topological (SPT) phases, either gapped or gapless. Gapless SPT (gSPT) have been recently studied in [9, 18–27]. A particularly interesting class of gSPTs are intrinsically gapless SPTs (igSPTs), which are gSPTs that cannot be deformed to SPT phases. A summary of all types of phases is given in table 1.

---

[2]See the appendices of [8, 10] for a discussion of the conditions on condensable and Lagrangian algebras.

Table 1: Types of (1+1)d phases, their physical characterization with properties of the energy and symmetry gaps. The last two columns list the conditions on the condensable algebra $\mathcal{A}$ in terms of which the phase is determined and the number of universes (equivalently vacua for gapped systems) $n$. $\mathcal{L}$ denotes a condensable algebra that is Lagrangian, and $\mathcal{L}_{\mathcal{S}}$ denotes the Lagrangian algebra for $\mathcal{S}$ symmetry.

| Phase | Physical characterization | Energy gap $\Delta$ Symmetry gap $\Delta_{\mathcal{S}}$ | Condition on $\mathcal{A}$ | $n$ |
|---|---|---|---|---|
| SPT | Gapped system with energy gap $\Delta > 0$. The IR theory is a trivial TQFT. The charges of $\mathcal{S}$ confined in the IR start to appear at an energy scale (called symmetry gap) $\Delta_{\mathcal{S}} \geq \Delta > 0$. Order parameters (OPs) are all of string type (i.e. in twisted-sectors for $\mathcal{S}$). | $\Delta > 0$ $\Delta_{\mathcal{S}} > 0$ | $\mathcal{A} = \mathcal{L}$ $\mathcal{A} \cap \mathcal{L}_{\mathcal{S}} = 1$ | 1 |
| gSPT | Gapless system with $\Delta = 0$ and a unique ground state on circle. Not all charges of $\mathcal{S}$ appear in the IR. The confined charges appear at a symmetry gap $\Delta_{\mathcal{S}} > 0$. OPs are all of string type. | $\Delta = 0$ $\Delta_{\mathcal{S}} > 0$ | $\mathcal{A} \neq \mathcal{L}$ $\mathcal{A} \cap \mathcal{L}_{\mathcal{S}} = 1$ | 1 |
| igSPT | A gSPT phase that cannot be deformed to a gapped SPT phase, because it has confined charges not exhibited by any of the gapped SPTs. | $\Delta = 0$ $\Delta_{\mathcal{S}} > 0$ | $\mathcal{A} \neq \mathcal{L}$ $\mathcal{A} \cap \mathcal{L}_{\mathcal{S}} = 1$ | 1 |
| SSB | Gapped system with $n$ degenerate vacua (labeled by $i$) permuted by $\mathcal{S}$ action. Each vacuum $i$ has energy gap $\Delta^{(i)} > 0$. Excitations between two vacua $i$ and $j$ cost non-zero energy $\Delta^{(ij)} > 0$. Not all charges are realized in the IR $\implies$ symmetry gap $\Delta_{\mathcal{S}} > 0$. OPs are multiplets involving both conventional (non-string-type) and string-type operators. | $\Delta^{(i)} > 0$ $\Delta^{(ij)} > 0$ $\Delta_{\mathcal{S}} > 0$ | $\mathcal{A} = \mathcal{L}$ $\mathcal{A} \cap \mathcal{L}_{\mathcal{S}} \supsetneq 1$ | $> 1$ |
| gSSB | Gapless system with $n$ degenerate gapless universes labeled by $i$. Each universe has a unique ground state on a circle. Excitations between two universes $i$ and $j$ cost non-zero energy $\Delta^{(ij)} > 0$. Not all charges are realized in the IR $\implies$ symmetry gap $\Delta_{\mathcal{S}} > 0$. OPs are multiplets involving both conventional (non-string-type) and string-type operators. | $\Delta^{(i)} = 0$ $\Delta^{(ij)} > 0$ $\Delta_{\mathcal{S}} > 0$ | $\mathcal{A} \neq \mathcal{L}$ $\mathcal{A} \cap \mathcal{L}_{\mathcal{S}} \supsetneq 1$ | $> 1$ |
| igSSB | A gSSB phase with $n$ universes that cannot be deformed to a gapped SSB phase with $n$ vacua. | $\Delta^{(i)} = 0$ $\Delta^{(ij)} > 0$ $\Delta_{\mathcal{S}} > 0$ | $\mathcal{A} \neq \mathcal{L}$ $\mathcal{A} \cap \mathcal{L}_{\mathcal{S}} \supsetneq 1$ | $> 1$ |

An example of a Hasse diagram for a SymTFT is shown in figure 1 for the group-like symmetry $\mathbb{Z}_4$, which also appeared in [9]. The topological lines in the Drinfeld center are generated by the lines $e$ and $m$ with $e^4 = m^4 = 1$. In the top half we have shown the condensable algebras, with the lowest level given by the Lagrangians. In this Hasse diagram of condensable algebras we have not yet chosen any symmetry $\mathcal{S}$.

In the bottom half of figure 1 we have picked the symmetry $\mathcal{S} = \mathbb{Z}_4$, i.e. Lagrangian $\mathcal{L}_{\mathcal{S}} = 1 \oplus e \oplus e^2 \oplus e^3$, and indicate what type of phase each of these condensable algebras corresponds to. Gapped and gapless SPT phases are given in terms of condensable algebras $\mathcal{A}$ such that $\mathcal{A} \cap \mathcal{L}_{\mathcal{S}} = 1$. If the condensable algebra is maximal (Lagrangian) it is a gapped SPT, otherwise a gapless SPT. In addition, if there is no edge in the Hasse diagram that connects a gSPT with a gapped SPT, then this is an igSPT. For $\mathbb{Z}_4$ there is precisely one igSPT phase, which has been observed in previous literature.

Our approach allows identification of not only gapped and gapless SPT phases, but also gapped and gapless SSB phases for any fusion category symmetry, and the associated phase Hasse diagram describing possible patterns in which these phases can be deformed into each other. We carry this out for $\mathsf{Vec}(S_3)$, and non-invertible symmetries $\mathsf{Rep}(S_3)$ and $\mathsf{Rep}(D_8)$ in figures 2 and 3, and find a new igSPT phase for $\mathsf{Rep}(D_8)$, which to our knowledge is the first **igSPT phase for a non-invertible symmetry**. We also provide igSPT phases for $\mathsf{Rep}(D_{8m})$ symmetries for any $m \geq 1$. The Hasse diagram remains unchanged if we change the symmetry $\mathcal{S}$ to $\mathcal{S}'$ with the same SymTFT, merely the types of phases have a different interpretation.

Physically, the phases can be characterized by the energy gap $\Delta$ (standard gap in the spectrum of states and operators) but also by the symmetry gap $\Delta_{\mathcal{S}}$. When $\Delta_{\mathcal{S}} > 0$, not all charges of $\mathcal{S}$ are realized in the IR phase. The confined charges are realized in such an instance by excited states. The symmetry gap $\Delta_{\mathcal{S}}$ is the energy of the first excited state carrying one of the confined charges. We have summarized this in table 1.

**Generalized superconductivity interpretation.** The gapped and gapless phases discussed here can be given a generalized superconductivity description.[3] Given the symmetry $\mathcal{S}$, the key objects are the charges that can be carried by the operators in the SymTFT bulk. (Such operators may correspond to extended or disorder operators in the microscopic theory.) These charges are captured by anyons of the SymTFT $\mathfrak{Z}(\mathcal{S})$, which form the Drinfeld center $\mathcal{Z}(\mathcal{S})$ [5–7, 17, 28–30]. The mutual braiding between two anyons describes the mutual non-locality between two local operators carrying the charges associated to the two anyons. Each phase (gapped or gapless) is characterized by a set of mutually local charges that condense. This set of condensed charges is characterized by a condensable algebra in $\mathcal{Z}(\mathcal{S})$. Any charge that is non-local with the condensed charges is confined and does not appear in the IR, showing up only after a symmetry gap $\Delta_{\mathcal{S}}$ in the spectrum. This may be viewed as a generalized Meissner effect. On the other hand, any charge that is not condensed and that is local with all the condensed charges is deconfined, i.e. it must be carried by a gapless excitation arising in the IR. The phase under consideration is thus necessarily a gapless phase if there are deconfined non-condensed charges. On the other hand, for a phase to be gapped, we must have a maximal amount of condensed charges such that it is not possible to have deconfined non-condensed charges. This translates to the fact that the corresponding condensable algebra is maximal, or in other words, Lagrangian. A phase $\mathcal{P}_1$ can be obtained by a (small) deformation of another phase $\mathcal{P}_2$ only if $\mathcal{P}_1$ can be obtained from $\mathcal{P}_2$ by further condensing some of the charges deconfined in $\mathcal{P}_2$. This translates to the fact that the condensable algebra corresponding to $\mathcal{P}_2$ must be a subalgebra of the condensable algebra corresponding to $\mathcal{P}_1$. (Gapped or gapless) SSB phases are distinguished from (gapped or gapless) SPT phases depending on whether or not there are untwisted sector (i.e. genuine) local operators that are condensed.

Although our main discussion is in the context of (1+1)d theories and $\mathcal{S}$ a fusion category symmetry, the general setting, already outlined in [1], is applicable in higher dimensions as well for higher fusion category symmetries or non-invertible symmetries recently uncovered in higher dimensions [31–34] (for reviews see [35, 36]), which we intend to explore in the future.

Finally, we note that our approach for the classification of $\mathcal{S}$-symmetric phases in this work relies on the use of condensable algebras at the object level, without explicitly verifying the possible multiplication structures of the algebras which can pose certain limitations in more complex examples than the ones studied here. A forthcoming work [37] by a subset of the authors will compute the algebra multiplication for all group theoretical cases considered here and formally confirm the present results.

## 2 Gapped SPT phases

In this section we review the case of gapped symmetry protected topological (SPT) phases, which will simply be referred to as SPT phases without the additional adjective "gapped". The case of gapless SPT (gSPT) phases will be discussed in section 3.

Consider a symmetry $\mathcal{S}$ that can act on $(D + 1)$-dimensional systems.[4] This could be a 0-form i.e. group symmetry possibly with a 't Hooft anomaly, a higher-form or higher-group symmetry possibly with a 't Hooft anomaly, or a non-invertible (higher-)categorical symmetry.[5]

---

[3]We thank Apoorv Tiwari for a discussion on this point.

[4]Throughout this work, we consider systems only in infinite volume. Although our considerations apply also to large classes of lattice systems, we will always have in mind systems described by relativistic quantum field theories. We will always display the spacetime dimension rather than the spatial dimension. Also, $D$ denotes the space dimension and $d = D + 1$ is spacetime dimension.

[5]We only study finite symmetries in this work, which restricts $\mathcal{S}$ to be a multi-fusion $D$-category. We further restrict our attention only to fusion $D$-categories by disallowing the presence of $D$-form symmetries. We take such a fusion $D$-category to be equipped with a pivotal structure that allows one to consider foldings of topological defects comprising the symmetry.

An $\mathcal{S}$-symmetric $(D+1)$-dimensional gapped system $\mathfrak{T}$ is in an SPT phase for $\mathcal{S}$ symmetry if it lies in the trivial gapped phase once the symmetry $\mathcal{S}$ is forgotten. The IR effective theory for a system lying in a trivially gapped phase is the trivial TQFT, which has a single vacuum, a unique ground state on every closed spatial manifold, and a trivial partition function on every closed spacetime manifold.

Two $\mathcal{S}$-symmetric gapped systems are said to be lying in different SPT phases for $\mathcal{S}$ symmetry if the $\mathcal{S}$ symmetry is realized differently on the trivial IR TQFT. That is, the spectrum of (possibly extended or non-genuine) operators/defects or states in the IR that are charged under $\mathcal{S}$ is different for two systems lying in two different SPT phases. For a non-anomalous finite group symmetry $G$, one often distinguishes SPT phases by inspecting the projective (fractional) charges of ground states in each symmetry-twisted sector on a circle: this however fails for the example of order 128 constructed in [38], where every twisted-sector ground state carries the trivial $G$-charge, even though the phase is non-trivial since its 2-cocycle is non-trivial in $H^2(G, U(1))$.

Also, not all the charges of symmetry $\mathcal{S}$ are realized in the IR. The "confined charges" are simple objects of the bulk theory $\mathcal{Z}(\mathcal{S})$ that do not survive as local boundary excitations, i.e. they are realized by gapped excited states. This allows us to define a symmetry gap $\Delta_{\mathcal{S}}$, which is the energy of the first excited state (which could be in a twisted or untwisted sector) carrying one of the confined charges. This should in general be distinguished from the energy gap $\Delta$, which is the energy of the first excited state irrespective of how it transforms under $\mathcal{S}$. Closing $\Delta_{\mathcal{S}}$ necessarily closes $\Delta$, but the converse is not true. This distinction between $\Delta_{\mathcal{S}}$ and $\Delta$ will be important when we discuss gapless SPT phases, for which $\Delta = 0$ but $\Delta_{\mathcal{S}} \neq 0$.[6]

It is not possible to connect two $\mathcal{S}$-symmetric systems lying in different SPT phases by $\mathcal{S}$-symmetric deformations without closing the energy gap. This is because two different SPT phases have different sets of confined charges. In fact, there always exists a confined charge for one of the SPTs that is realized in the IR for the other SPT. Thus, one necessarily needs to go through a point where $\Delta_{\mathcal{S}} = 0$ to transition between two SPT phases, but that means going through a point with $\Delta = 0$.

As mentioned earlier, the SPT phase occupied by the system is captured by how the $\mathcal{S}$ symmetry is realized in the trivial IR TQFT. The symmetry $\mathcal{S}$ is a collection of topological defects forming the structure of a pivotal[7] fusion $D$-category. A realization of the symmetry on a theory $\mathfrak{T}$ means that we find topological defects of $\mathfrak{T}$ labeled by elements of $\mathcal{S}$ that obey the fusion rules and have the quantum dimensions of the elements of $\mathcal{S}$. Mathematically, this amounts to choosing a pivotal tensor $D$-functor

$$\phi : \ \mathcal{S} \to \mathcal{S}(\mathfrak{T}), \tag{1}$$

where $\mathcal{S}(\mathfrak{T})$ is the $D$-category formed by topological defects of $\mathfrak{T}$. Choosing such a functor $\phi$ converts a theory $\mathfrak{T}$ to an $\mathcal{S}$-symmetric theory $\mathfrak{T}^{\mathcal{S}}$, whose underlying non-symmetric theory is $\mathfrak{T}$. An SPT phase is an $\mathcal{S}$-symmetric TQFT whose underlying TQFT is trivial, which we denote by $\mathfrak{T}_{\text{triv}}$.[8] Thus, an SPT phase can be described as a functor

$$\mathcal{S} \to \mathcal{S}(\mathfrak{T}_{\text{triv}}), \tag{2}$$

where $\mathcal{S}(\mathfrak{T}_{\text{triv}})$ is the $D$-category formed by topological defects of $\mathfrak{T}_{\text{triv}}$, which can be identified with TQFTs of spacetime dimension less than or equal to $D$.

---

[6]We thank Apoorv Tiwari and Sanjay Moudgalya for an insightful discussion on this point.

[7]Throughout we assume the symmetry fusion $D$-category is pivotal, which guarantees that quantum dimensions and traces are isotopy- and orientation-invariant. All unitary examples (Rep($G$), Drinfeld centers, etc.) admit a canonical pivotal (or often implicitly assumed spherical) choice, so the assumption is physically justified while keeping the graphical calculus unambiguous.

[8]Throughout this work we obfuscate the difference between TQFTs and gapped phases for brevity. One should keep in mind that gapped phases are deformation classes of TQFTs.

In $(1 + 1)$-dimensions, i.e. for $D = 1$, we have

$$\mathcal{S}(\mathsf{T}_{\mathrm{triv}}) = \mathsf{Vec}, \tag{3}$$

which is the category formed by vector spaces. Physically such a vector space describes the Hilbert space of a topological quantum mechanics regarded as a topological defect of a trivial $(1+1)$d TQFT. Thus, an SPT phase in $(1+1)$d is characterized by a functor of the form

$$\phi : \mathcal{S} \to \mathsf{Vec}. \tag{4}$$

Such a functor is also referred to as a **fiber functor**, reproducing the description of $(1+1)$d SPT phases appearing in [3].

In order to describe condensed charges, or equivalently the charges realized in the IR, associated to a fiber functor $\phi$, it is useful to first discuss the Symmetry Topological Field Theory (SymTFT) construction, also known as topological holography, of SPT phases. This is because the charges for a symmetry $\mathcal{S}$ are neatly encoded as topological defects of the SymTFT $\mathfrak{Z}(\mathcal{S})$ associated to $\mathcal{S}$ [17].

**Symmetry TFT description.** Any $d$-dimensional $\mathcal{S}$-symmetric theory $\mathfrak{T}^{\mathcal{S}}$ admits a sandwich construction as an interval compactification of the $(d + 1)$-dimensional SymTFT $\mathfrak{Z}(\mathcal{S})$ associated to the $\mathcal{S}$ symmetry, with one end of the interval occupied by a topological boundary condition (known as symmetry boundary) $\mathfrak{B}_{\mathcal{S}}^{\mathrm{sym}}$ of $\mathfrak{Z}(\mathcal{S})$ capturing the symmetry $\mathcal{S}$, and the other end of the interval occupied by a possibly non-topological boundary condition (known as physical boundary) $\mathfrak{B}_{\mathfrak{T}^{\mathcal{S}}}^{\mathrm{phys}}$ of $\mathfrak{Z}(\mathcal{S})$, capturing the dynamical information of $\mathfrak{T}^{\mathcal{S}}$. The topological defects arising at the boundary $\mathfrak{B}_{\mathcal{S}}^{\mathrm{sym}}$ form the fusion $(d - 1)$-category $\mathcal{S}$, and realize the symmetry defects of $\mathfrak{T}^{\mathcal{S}}$ after the interval compactification. The sandwich construction is depicted schematically as

$$
\begin{array}{cc}
\overset{\displaystyle \mathfrak{B}_{\mathcal{S}}^{\mathrm{sym}} \qquad \mathfrak{B}_{\mathfrak{T}^{\mathcal{S}}}^{\mathrm{phys}}}{\boxed{\phantom{XX}\mathfrak{Z}(\mathcal{S})\phantom{XX}}} \qquad = \qquad \overset{\displaystyle \mathfrak{T}^{\mathcal{S}}}{\big|}
\end{array}, \tag{5}
$$

where we display only two of the $(d + 1)$ dimensions for simplicity of exposition. For $d = 2$, a symmetry $\mathcal{S}$ is described by a fusion category and the 3d SymTFT $\mathfrak{Z}(\mathcal{S})$ is obtained by performing Turaev-Viro-Barrett-Westbury construction based on the fusion category $\mathcal{S}$. For $\mathcal{S} = \mathsf{Vec}_G^\omega$, i.e. a symmetry group $G$ with 't Hooft anomaly $\omega$, a lattice model for the 3d SymTFT is provided by a (possibly twisted) quantum double model based on $G$ [39–41]; while for general $\mathcal{S}$ one may use a Levin-Wen string-net model [42] with the fusion category $\mathcal{S}$ as an input.

A $p$-dimensional operator $\mathcal{O}_p$ of $\mathfrak{T}^{\mathcal{S}}$ carrying a non-trivial charge under $\mathcal{S}$ is constructed by compactifying a non-trivial $(p + 1)$-dimensional topological defect $\mathbf{Q}_{p+1}$ of $\mathfrak{Z}(\mathcal{S})$ sandwiched between the symmetry and physical boundaries:

$$
\begin{array}{ccc}
\overset{\displaystyle \mathfrak{T}^{\mathcal{S}}}{\underset{\displaystyle \mathcal{O}_p}{\big|}} & = & \overset{\displaystyle \mathfrak{B}_{\mathcal{S}}^{\mathrm{sym}} \qquad \mathfrak{B}_{\mathfrak{T}^{\mathcal{S}}}^{\mathrm{phys}}}{\boxed{\begin{array}{c} \mathfrak{Z}(\mathcal{S}) \\ \mathbf{Q}_{p+1} \end{array}}}\ \mathcal{O}_p^{\partial}
\end{array}, \tag{6}
$$

where we have suppressed $p$ spacetime dimensions of $\mathcal{O}_p$ and $\mathbf{Q}_{p+1}$. As the symmetry generating topological defects are localized completely along the symmetry boundary $\mathfrak{B}_{\mathcal{S}}^{\mathrm{sym}}$, the

charge of the resulting $p$-dimensional operator $\mathcal{O}_p$ under $\mathcal{S}$ is determined completely by how the topological defects of $\mathfrak{B}^{\text{sym}}_{\mathcal{S}}$ act on the ends of $\mathbf{Q}_{p+1}$ along the boundary $\mathfrak{B}^{\text{sym}}_{\mathcal{S}}$. Thus, the charge of a $p$-dimensional operator $\mathcal{O}_p$ is labeled by the topological defect $\mathbf{Q}_{p+1}$ involved in its sandwich construction. Specializing to $d = 2$, the charges of (untwisted and twisted sector) local operators under a fusion category symmetry $\mathcal{S}$ are described by anyons (i.e. topological line defects) of the Turaev-Viro theory $\mathfrak{Z}(\mathcal{S})$, which mathematically form the modular tensor category $\mathcal{Z}(\mathcal{S})$ known as the Drinfeld center of $\mathcal{S}$.[9]

$\mathcal{S}$-symmetric $d$-dimensional TQFTs are realized by taking the physical boundary $\mathfrak{B}^{\text{phys}}$ to also be topological. Not all topological defects of $\mathfrak{Z}(\mathcal{S})$ can end along a topological boundary: roughly only half of them can end. These ending topological defects characterize the charges for $\mathcal{S}$-symmetry that are realized in the corresponding $\mathcal{S}$-symmetric TQFT. The (possibly extended) operators creating IR states carrying non-trivial charges are recognized as the order parameters for the $\mathcal{S}$-symmetric gapped phase described by the $\mathcal{S}$-symmetric TQFT. These operators also carry the same charges under $\mathcal{S}$, and thus topological defects of the SymTFT ending along the physical boundary capture charges of order parameters [1,2]. The topological defects of $\mathfrak{Z}(\mathcal{S})$ that cannot end along $\mathfrak{B}^{\text{phys}}$ describe the confined charges that do not arise in the IR TQFT, but are carried instead by excited states.

Let us specialize now to $d = 2$. A topological boundary is specified (up to continuous deformations) by a Lagrangian algebra $\mathcal{L}$ in the Drinfeld center $\mathcal{Z}(\mathcal{S})$. The objects of $\mathcal{Z}(\mathcal{S})$ appearing in $\mathcal{L}$ are the anyons that can end along the topological boundary. The charges realized in an $\mathcal{S}$-symmetric 2d TQFT are thus the objects of the Lagrangian algebra $\mathcal{L}_{\text{phys}}$ associated to the topological physical boundary $\mathfrak{B}^{\text{phys}}$, which also capture the charges of local operators acting as order parameters for the associated $\mathcal{S}$-symmetric gapped phase. The objects of $\mathcal{Z}(\mathcal{S})$ not appearing in $\mathcal{L}_{\text{phys}}$ are the confined charges not realized by the TQFT but are realized instead by the excited states. Note that the anyons appearing in a Lagrangian algebra must all be bosons and must have trivial mutual braidings.

SPT phases for $\mathcal{S}$ are special $\mathcal{S}$-symmetric TQFTs whose underlying non-symmetric TQFT is the trivial TQFT. The corresponding physical boundaries are also some special topological boundaries of the SymTFT. For (1+1)d SPT phases, it is easy to describe the special condition that a topological physical boundary for an SPT phase has to satisfy. Let $\mathcal{L}_{\text{sym}}$ be the Lagrangian algebra for the symmetry boundary $\mathfrak{B}^{\text{sym}}_{\mathcal{S}}$ and $\mathcal{L}_{\text{phys}}$ be the Lagrangian algebra for the physical boundary $\mathfrak{B}^{\text{phys}}$. The (non-symmetric) TQFTs in 2d (up to continuous deformations associated to Euler terms) are characterized by the number of vacua (or ground states on a circle) they have. By the state-operator correspondence, this is equal to the number of linearly independent topological local operators of the TQFT. The trivial 2d TQFT corresponds to having a one-dimensional vector space of topological local operators generated by the identity local operator. Since the underlying non-symmetric TQFT for an SPT phase is trivial, we want to choose a topological physical boundary $\mathfrak{B}^{\text{phys}}$ such that the resulting 2d TQFT has no non-identity topological local operators. There are three sources for such operators:

1. Topological local operators of $\mathfrak{Z}(\mathcal{S})$.

2. Topological local operators of $\mathfrak{B}^{\text{sym}}_{\mathcal{S}}$ and $\mathfrak{B}^{\text{phys}}$.

3. Topological line operators of $\mathfrak{Z}(\mathcal{S})$ ending along both $\mathfrak{B}^{\text{sym}}_{\mathcal{S}}$ and $\mathfrak{B}^{\text{phys}}$.

In other words, the last type of local operators are obtained as interval compactifications of bulk line operators. The first type of local operators are all proportional to the identity, which has to do with the fact that $\mathfrak{Z}(\mathcal{S})$ is a SymTFT for a fusion category symmetry $\mathcal{S}$. This may fail

---

[9]For $\mathcal{S} = \mathsf{Vec}_G$ a non-anomalous group symmetry, one can recognize $\mathcal{Z}(\mathcal{S})$ as the category of representations formed by the quantum double $D(G)$ of the group $G$.

for multi-fusion category symmetries. The second type of local operators are also all proportional to identity, which has to do with the fact that the boundary conditions are associated to Lagrangian algebras, or relatedly that the topological defects on the boundaries form a fusion category. If, instead, the topological defects on the boundary form a multi-fusion category, then just by definition, there are non-identity topological local operators hosted by the boundary. The third type of local operators do not arise if we demand

$$\mathcal{L}_{\text{sym}} \cap \mathcal{L}_{\text{phys}} = 1 \,, \tag{7}$$

i.e. the only anyon appearing both in $\mathcal{L}_{\text{sym}}$ and $\mathcal{L}_{\text{phys}}$ is the trivial anyon 1. Thus, the condition (7) provides a SymTFT characterization for **2d SPT phases**.

## 2.1 Example: $\mathbb{Z}_2 \times \mathbb{Z}_2$

Let us discuss a couple of examples. First consider an invertible symmetry $\mathcal{S}$ given by a non-anomalous group $\mathbb{Z}_2 \times \mathbb{Z}_2$, which is denoted as

$$\mathcal{S} = \text{Vec}_{\mathbb{Z}_2 \times \mathbb{Z}_2} \,. \tag{8}$$

The anyons of the SymTFT are

$$\mathcal{Z}(\text{Vec}_{\mathbb{Z}_2 \times \mathbb{Z}_2}) = \left\{ e_1^{s_1} e_2^{s_2} m_1^{s_1'} m_2^{s_2'} \,\middle|\, s_i, s_i' \in \{0, 1\} \right\} \,, \tag{9}$$

where $e_i$ and $m_j$ are bosonic lines, with the braiding between $e_i$ and $m_i$ given by a non-trivial sign. The SymTFT can also be recognized as two toric codes stacked together. The symmetry Lagrangian algebra is

$$\mathcal{L}_{\text{Vec}_{\mathbb{Z}_2 \times \mathbb{Z}_2}}^{\text{sym}} = 1 \oplus e_1 \oplus e_2 \oplus e_1 e_2 \,. \tag{10}$$

There are two possible SPT phases since for a group symmetry $G = \mathbb{Z}_2 \times \mathbb{Z}_2$ in 2d they are simply characterized by the cohomology group

$$H^2(\mathbb{Z}_2 \times \mathbb{Z}_2, U(1)) = \mathbb{Z}_2 \,. \tag{11}$$

The Lagrangian algebras for the two SPT phases are

$$\begin{aligned} \mathcal{L}_0^{\text{phys}} &= 1 \oplus m_1 \oplus m_2 \oplus m_1 m_2 \,, \\ \mathcal{L}_1^{\text{phys}} &= 1 \oplus e_2 m_1 \oplus e_1 m_2 \oplus e_1 e_2 m_1 m_2 \,. \end{aligned} \tag{12}$$

Indeed the two physical Lagrangian algebras satisfy (7).

Let us now discuss the $\mathbb{Z}_2^{(1)} \times \mathbb{Z}_2^{(2)}$ (generalized) charges realized in the IR for these SPT phases, and the charges confined in the IR. All possible charges are parametrized by the SymTFT anyons as follows:

- The interval compactification of the anyon $e_1^k e_2^l$ constructs an untwisted sector local operator with charge $k$ under $\mathbb{Z}_2^{(1)}$ and charge $l$ under $\mathbb{Z}_2^{(2)}$.

- The interval compactification of the anyon $e_1^k e_2^l m_1$ constructs a local operator in the twisted sector for $\mathbb{Z}_2^{(1)}$ with charge $k$ under $\mathbb{Z}_2^{(1)}$ and charge $l$ under $\mathbb{Z}_2^{(2)}$.

- The interval compactification of the anyon $e_1^k e_2^l m_2$ constructs a local operator in the twisted sector for $\mathbb{Z}_2^{(2)}$ with charge $k$ under $\mathbb{Z}_2^{(1)}$ and charge $l$ under $\mathbb{Z}_2^{(2)}$.

- The interval compactification of the anyon $e_1^k e_2^l m_1 m_2$ constructs a local operator in the twisted sector for the diagonal $\mathbb{Z}_2$ in $\mathbb{Z}_2^{(1)} \times \mathbb{Z}_2^{(2)}$ with charge $k$ under $\mathbb{Z}_2^{(1)}$ and charge $l$ under $\mathbb{Z}_2^{(2)}$.

The charges realized in the IR are captured by the corresponding Lagrangian algebra. The SPT phase for $\mathcal{L}_0^{\text{phys}}$ thus realizes only uncharged twisted sector operators, while the SPT phase for $\mathcal{L}_1^{\text{phys}}$ realizes

- a $\mathbb{Z}_2^{(1)}$ twisted sector operator charged under $\mathbb{Z}_2^{(2)}$ but not under $\mathbb{Z}_2^{(1)}$,

- a $\mathbb{Z}_2^{(2)}$ twisted sector operator charged under $\mathbb{Z}_2^{(1)}$ but not under $\mathbb{Z}_2^{(2)}$,

- an operator in the twisted sector for the diagonal $\mathbb{Z}_2$ which is charged under both $\mathbb{Z}_2^{(1)}$ and $\mathbb{Z}_2^{(2)}$.

It is actually a general fact that the only non-trivial operators realized by an SPT phase are all in twisted sectors for the symmetry. The corresponding order parameters are referred to as string order parameters.

## 2.2  Example: $\text{Rep}(D_8)$

Next consider a non-invertible symmetry

$$\mathcal{S} = \text{Rep}(D_8), \tag{13}$$

formed by representations of the dihedral group $D_8$ of order 8. We denote the elements of $D_8$ as

$$D_8 = \{1, a, a^2, a^3, x, ax, a^2x, a^3x\}, \tag{14}$$

with multiplication rule

$$a^4 = 1 = x^2, \qquad xa = a^3x. \tag{15}$$

Its irreducible representations, which are taken to be the symmetry generators, are

$$1, \quad 1_a, \quad 1_x, \quad 1_{ax}, \quad E, \tag{16}$$

where $1$ is the trivial one-dimensional representation, $1_a$ is a one-dimensional representation in which $a$ acts trivially but $x$ and $ax$ act by a non-trivial sign, $1_x$ is a one-dimensional representation in which $x$ acts trivially but $a$ and $ax$ act by a non-trivial sign, $1_{ax}$ is a one-dimensional representation in which $ax$ acts trivially but $a$ and $x$ act by a non-trivial sign, and $E$ is a two-dimensional representation comprised of basis vectors $v_1, v_2$ having transformations

$$\begin{aligned} a: \quad & v_1 \to iv_1, \; v_2 \to -iv_2, \\ x: \quad & v_1 \to v_2, \; v_2 \to v_1. \end{aligned} \tag{17}$$

The fusion rules of symmetry generators are described by tensor products of representations. The one-dimensional representations form a group $\mathbb{Z}_2 \times \mathbb{Z}_2$ under tensor products, and the other tensor products are

$$\begin{aligned} E \otimes 1_i &= 1_i \otimes E = E, \\ E \otimes E &= 1 \oplus 1_a \oplus 1_x \oplus 1_{ax}, \end{aligned} \tag{18}$$

for all $i \in \{a, x, ax\}$.

The Drinfeld center for $\text{Rep}(D_8)$ is equivalent to the Drinfeld center for the group $D_8$ as a braided fusion category, i.e.

$$\mathcal{Z}(\text{Rep}(D_8)) \simeq \mathcal{Z}(\text{Vec}_{D_8}). \tag{19}$$

Yet for our intents and purposes, we may crudely equate these two Drinfeld centers as even though these two categories are technically not the same, they define a bulk SymTFT with

the same braided monoidal structure. For any finite group $G$, the Drinfeld center has simple objects labeled by

$$([g], \rho), \tag{20}$$

where $[g]$ is a conjugacy class in $G$ and $\rho$ is an irreducible representation of the centralizer $H_g$ of an element $g \in [g]$. The conjugacy classes for $D_8$ are

$$1 = \{1\}, \qquad a^2 = \{a^2\}, \qquad a = \{a, a^3\}, \qquad x = \{x, a^2 x\}, \qquad ax = \{ax, a^3 x\}, \tag{21}$$

where we have also labeled them. Note that the labeling is in terms of group elements, and it should be clear from the context whether a certain label refers to a group element or a conjugacy class. The centralizers are

$$H_1 = H_{a^2} = D_8, \qquad H_x \cong H_{ax} = \mathbb{Z}_2^2, \qquad H_a = \mathbb{Z}_4. \tag{22}$$

Table 2 lists the anyons in the Drinfeld center of $\mathrm{Rep}(D_8)$, $\mathcal{Z}(\mathrm{Rep}(D_8))$, as $([g], \rho)$ along with the labels of reference [43] (which we shall also adopt in the following), their quantum dimensions and $T$-matrix elements (which encode the anyon spins). The modular data ($S$, $T$ matrices and fusion coefficients $N_{jk}^i$) for $\mathcal{Z}(\mathrm{Vec}_G) = \mathcal{Z}(\mathrm{Rep}(G))$, with $G$ a finite group, can be computed from the expressions in references [44, 45].

As shown in [46] and later discussed in [43], $\mathcal{Z}(\mathrm{Rep}(D_8))$ exhibits the same topological order as the twisted Drinfeld center $\mathcal{Z}(\mathrm{Vec}_{\mathbb{Z}_2^3}^\omega)$ of three copies of the toric code, each of which is generated by the bosons

$$e_C, m_C, \quad C \in \{R, G, B\}, \tag{23}$$

labeled by a color index. The $e_C$'s satisfy invertible fusion rules, whereas those of the $m_C$'s are non-invertible. Each $m_C$ braids non-trivially with the corresponding $e_C$. Furthermore, there are 3+3 non-invertible fermions

$$\begin{aligned}
f_C &= m_C \otimes e_C, \\
f_{C_1 C_2} &= m_{C_1 C_2} \otimes e_{C_1} = m_{C_1 C_2} \otimes e_{C_2},
\end{aligned} \tag{24}$$

where $C, C_1, C_2 \in \{R, G, B\}$, a semion $s_{RGB}$ (which braids with all $e_C$'s) and an anti-semion

$$\bar{s}_{RGB} = s_{RGB} \otimes e_{RBG}. \tag{25}$$

The Lagrangian algebra that corresponds to the symmetry $\mathrm{Rep}(D_8)$ is

$$\mathcal{L}_{\mathrm{Rep}(D_8)}^{\mathrm{sym}} = 1 \oplus e_{RGB} \oplus m_{GB} \oplus m_{RB} \oplus m_{RG}. \tag{26}$$

There are three SPT phases for the $\mathcal{S} = \mathrm{Rep}(D_8)$ symmetry corresponding to Lagrangian algebras

$$\begin{aligned}
\mathcal{L}_0^{\mathrm{phys}} &= 1 \oplus e_{RG} \oplus e_R \oplus e_G \oplus 2m_B, \\
\mathcal{L}_1^{\mathrm{phys}} &= 1 \oplus e_B \oplus e_R \oplus e_{RB} \oplus 2m_G, \\
\mathcal{L}_2^{\mathrm{phys}} &= 1 \oplus e_B \oplus e_{GB} \oplus e_G \oplus 2m_R,
\end{aligned} \tag{27}$$

which satisfy condition (7).

Let us now discuss all the charges for $\mathrm{Rep}(D_8)$ symmetry, and the ones that can arise in these three SPT phases, which are also the charges of the (string) order parameters for these SPT phases. But first to connect with the anyon notation we introduce the following relabeling of the $\mathrm{Rep}(D_8)$ generators (16) as

$$R \equiv 1_x, \qquad G \equiv 1_{ax}, \qquad RG \equiv 1_a, \qquad B \equiv E, \tag{28}$$

Table 2: Anyons in $\mathcal{Z}(\text{Rep}(D_8))$ can be classified by a choice of conjugacy class $[g]$ and an irreducible representation $\rho$ of the corresponding centralizer $H_g$ (first column). Equivalently, they can be labeled in terms of three copies of the toric code (second column) [43]. The quantum dimensions of each anyon and diagonal $T$-matrix elements (which encode the anyon spins) are listed in columns 3 and 4 respectively.

| $([g],\rho)$ | Anyon label | Dim | $T$ |
|:---:|:---:|:---:|:---:|
| $(1,1)$ | $1$ | 1 | 1 |
| $(1,1_a)$ | $e_{RG}$ | 1 | 1 |
| $(1,1_x)$ | $e_R$ | 1 | 1 |
| $(1,1_{ax})$ | $e_G$ | 1 | 1 |
| $(1,E)$ | $m_B$ | 2 | 1 |
| $(a^2,1)$ | $e_{RGB}$ | 1 | 1 |
| $(a^2,1_a)$ | $e_B$ | 1 | 1 |
| $(a^2,1_x)$ | $e_{GB}$ | 1 | 1 |
| $(a^2,1_{ax})$ | $e_{RB}$ | 1 | 1 |
| $(a^2,E)$ | $f_B$ | 2 | $-1$ |
| $(a,1)$ | $m_{RG}$ | 2 | 1 |
| $(a,i)$ | $s_{RGB}$ | 2 | $i$ |
| $(a,-1)$ | $f_{RG}$ | 2 | $-1$ |
| $(a,-i)$ | $\bar{s}_{RGB}$ | 2 | $-i$ |
| $(x,+,+)$ | $m_{GB}$ | 2 | 1 |
| $(x,+,-)$ | $m_G$ | 2 | 1 |
| $(x,-,-)$ | $f_G$ | 2 | $-1$ |
| $(x,-,+)$ | $f_{GB}$ | 2 | $-1$ |
| $(ax,+,+)$ | $m_{RB}$ | 2 | 1 |
| $(ax,+,-)$ | $m_R$ | 2 | 1 |
| $(ax,-,-)$ | $f_R$ | 2 | $-1$ |
| $(ax,-,+)$ | $f_{RB}$ | 2 | $-1$ |

where the 1d representations $\rho \in \{1,R,G,RG\}$ are related to the invertible anyons $e_\rho$ and the 2d representation $B$ is related to the non-invertible anyon $m_B$ rather than $e_B$ which is an invertible anyon. We therefore write the fusion rules (17) for $\text{Rep}(D_8)$ as

$$\begin{aligned} B \otimes \rho &= \rho \otimes B = B\,, \\ B \otimes B &= 1 \oplus R \oplus G \oplus RG\,, \end{aligned} \tag{29}$$

for all $\rho \in \{1,R,G,RG\}$.

As we discussed earlier, the charges are labeled by anyons of the SymTFT:

- *Multiplets uncharged under* $\text{Rep}(D_8)$: $1$, $e_R$, $e_G$, $e_{RG}$, $m_B$

  The anyons $e_\rho$ for 1d irreducible representations $\rho$ of $D_8$ describe multiplets of (single) twisted sector operators attached to the line operator $\rho \in \text{Rep}(D_8)$ generating the symmetry, where we defined $e_1 \equiv 1$. On the other hand, the $m_B$ multiplet includes two linearly independent operators, both in the $B$-twisted sector. All of these operators are uncharged under $\text{Rep}(D_8)$. These anyons correspond to $(1,\rho)$ in the alternative notation.

- *Multiplets charged under B: $e_B$, $e_{GB}$, $e_{RB}$, $e_{RGB}$, $f_B$*

  The anyons $e_{RGB \otimes \rho}$ for 1d irreducible representations $\rho$ of $D_8$ describe multiplets of (single) twisted sector operators attached to the line operator $\rho \in \text{Rep}(D_8)$ generating the symmetry. Note that the $e_{RGB}$ multiplet contains an untwisted local operator. The $f_B$ multiplet includes two linearly independent operators, both in the $B$-twisted sector. All of these operators transform by a sign $-1$ under the action of $B$ but are uncharged under the rest of $\text{Rep}(D_8)$. These anyons correspond to $(a^2, \rho)$ in the alternative notation.

- *Multiplets charged under R, G, B: $m_{RG}$, $f_{RG}$, $s_{RGB}$, $\bar{s}_{RGB}$*

  The anyon $m_{RG}$ describes a multiplet of two operators: one untwisted and one in $RG$ twisted sector. The anyon $f_{RG}$ describes a multiplet of two operators: one in $R$ twisted sector and one in $G$ twisted sector. The anyons $s_{RGB}$ and $\bar{s}_{RGB}$ describe multiplets of two operators, both in the $B$-twisted sector. The two operators in each multiplet are exchanged by the action of $B$, transform by a sign $-1$ under the action of $R$ and $G$, and are uncharged under $RG$. Note that the $s_{RGB}$ and $\bar{s}_{RGB}$ multiplets are distinguished by the precise exchange action of $B$. These anyons correspond to $(a, i^p)$ in the alternative notation.

- *Multiplets charged under G, RG, B: $m_{GB}$, $f_{GB}$, $m_G$, $f_G$*

  The anyon $m_{GB}$ describes a multiplet of two operators: one untwisted and one in $R$ twisted sector. The anyon $f_{GB}$ describes a multiplet of two operators: one in $G$ twisted sector and one in $RG$ twisted sector. The anyons $m_G$ and $f_G$ describe multiplets of two operators, both in the $B$-twisted sector. The two operators in each multiplet are exchanged by the action of $B$, transform by a sign $-1$ under the action of $G$ and $RG$, and are uncharged under $R$. Note that the $m_G$ and $f_G$ multiplets are distinguished by the precise exchange action of $B$. These anyons correspond to $(x, s, s')$ in the alternative notation.

- *Multiplets charged under R, RG, B: $m_{RB}$, $f_{RB}$, $m_R$, $f_R$*

  The anyon $m_{RB}$ describes a multiplet of two operators: one untwisted and one in $G$ twisted sector. The anyon $f_{GB}$ describes a multiplet of two operators: one in $R$ twisted sector and one in $RG$ twisted sector. The anyons $m_R$ and $f_R$ describe multiplets of two operators, both in the $B$-twisted sector. The two operators in each multiplet are exchanged by the action of $B$, transform by a sign $-1$ under the action of $R$ and $RG$, and are uncharged under $G$. Note that the $m_R$ and $f_R$ multiplets are distinguished by the precise exchange action of $B$. These anyons correspond to $(xa, s, s')$ in the alternative notation.

The three SPT phases are thus distinguished by the charges that they host:

1. The SPT phase corresponding to $\mathcal{L}_0^{\text{phys}}$ only hosts twisted sector local operators uncharged under the full $\text{Rep}(D_8)$. In this sense, it may be viewed as a trivial SPT phase.

2. The SPT phase corresponding to $\mathcal{L}_1^{\text{phys}}$ hosts twisted sector operators charged under $G$, $RG$ and $B$, but uncharged under $R$.

3. The SPT phase corresponding to $\mathcal{L}_2^{\text{phys}}$ hosts twisted sector operators charged under $R$, $RG$, $B$, but uncharged under $G$.

# 3 Gapless SPT phases

In this section, we discuss $(D+1)$-dimensional $\mathcal{S}$-symmetric gapless phases that generalize the SPT phases discussed above. By definition, the energy gap $\Delta = 0$ for gapless systems. We say

that a gapless system lies in a **gapless SPT (gSPT) phase** if it has a unique ground state on every closed spatial manifold.[10] This requirement ensures that we do not have spontaneous symmetry breaking or topological order.

Given an $\mathcal{S}$-symmetric gapless system lying in a gSPT phase, we can ask what are the charges realized in the IR of the system. If all possible charges of $\mathcal{S}$ are realized in the IR, then we say that the system lies in the **canonical gSPT** phase for $\mathcal{S}$ symmetry. Otherwise, if some charges are confined in the IR, we have a non-zero symmetry gap $\Delta_{\mathcal{S}} > 0$, which is the energy of the lowest excited state carrying one of the confined charges. In such a situation, the system lies in a **non-canonical gSPT** phase.

Physically, in the SymTFT picture each simple object $Q$ of the center $\mathcal{Z}(\mathcal{S})$ may or may not end on the physical boundary $\mathcal{B}^{\text{phys}}$. If $Q$ can end, its endpoint is a local, finite-energy operator; inserting a segment of $Q$ parallel to the boundary costs energy proportional to its length and thus does not generate low-energy modes. A boundary realizing a Lagrangian condensable algebra condenses a maximal set of mutually local lines: all lines mutually local with the condensate are either condensed or confined, so no deconfined lines remain. The collapsed one-dimension-lower theory is therefore fully gapped. Conversely, if the physical boundary is non-Lagrangian, at least one line $Q_*$ is not condensed or confined and remains deconfined: it can slide along the boundary with arbitrarily small energy cost, giving rise to a continuum of low-energy states whose energies accumulate at zero and whose correlation functions decay as power laws. The presence of any such deconfined line implies the theory is gapless, whereas their absence guarantees a finite spectral gap.

Considering the (1+1)d examples in this work, different gSPT phases are distinguished by the confined charges. More generally, however, the set of confined charges may not suffice to distinguish phases, particularly in higher dimensions where e.g. symmetry fractionalization data can also play a role. Constraining ourselves to the cases highlighted in this work in (1+1)d, then if we have two gapless systems $\mathfrak{T}_1$ and $\mathfrak{T}_2$ lying in two different gSPT phases with sets $Q_1$ and $Q_2$ of confined charges respectively, such that neither of the two is included in the other

$$Q_1 \not\subset Q_2, \quad \text{and} \quad Q_2 \not\subset Q_1, \tag{30}$$

then one cannot deform the two systems $\mathfrak{T}_1$ and $\mathfrak{T}_2$ into each other without breaking $\mathcal{S}$ symmetry and without closing the symmetry gap $\Delta_{\mathcal{S}}$. However, if $Q_1 \subset Q_2$ then there are no obstructions to deforming $\mathfrak{T}_1$ into $\mathfrak{T}_2$[11] without breaking $\mathcal{S}$ and without closing $\Delta_{\mathcal{S}}$, but $\mathfrak{T}_2$ cannot be deformed into $\mathfrak{T}_1$ without breaking $\mathcal{S}$ and without closing $\Delta_{\mathcal{S}}$.

Note that the confined charges exhibited by gapped SPT phases can also be exhibited by gapless SPT phases. The only way such gapless SPTs differ from gapped SPTs is that the former have $\Delta = 0$ while the latter have $\Delta > 0$; however, both have $\Delta_{\mathcal{S}} > 0$. There is no obstruction to deforming such gapless SPTs to gapped SPTs, which can in principle be done by simply opening up the energy gap $\Delta$.

We define an **intrinsically gapless SPT (igSPT) phase** to be a gSPT phase that cannot be deformed to a gapped SPT phase because it exhibits a confined charge that is not exhibited by any of the gapped SPTs. If a gSPT phase has no such obstruction to be deformed to a gapped SPT phase, we refer to it as a (non-intrinsic) gSPT phase.

The physical idea behind the consideration of igSPTs is that of **symmetry protected criticality**. Since an igSPT cannot be deformed to a gapped SPT, the gaplessness $\Delta = 0$ is protected under deformations as long as we do not explicitly *or spontaneously* break the symmetry $\mathcal{S}$. In (1+1)d, this obstruction guarantees a genuinely gapless phase, i.e. one that cannot be made gapped without breaking or extending the symmetry. However, in $d \geq 3$, the same anomaly may instead be saturated by a gapped symmetry-enriched topologically ordered (SET) phase.

---

[10]Note that gaplessness is a feature of infinite volume. At finite volume, we have a gap in the spectrum.

[11]The actual existence of such a deformation is still a dynamical property of the systems $\mathfrak{T}_1$ and $\mathfrak{T}_2$.

Let us emphasize, then, that although an igSPT cannot be deformed to an SPT phase while preserving $\mathcal{S}$, it can still be gapped either by spontaneously breaking the symmetry $\mathcal{S}$, or by entering an SET phase that retains $\mathcal{S}$. In the rest of this work, we restrict our use of the term "symmetry-protected criticality" to the $d = 2$ setting, where the absence of a gapped symmetric endpoint enforces true gaplessness.

Let us now describe the mathematical structure of gSPT phases. Consider a gapless system lying in a gSPT phase. Due to the presence of confined charges, the symmetry $\mathcal{S}$ does not act faithfully on the gapless IR degrees of freedom, but only a smaller symmetry $\mathcal{S}'$ acts faithfully on the IR degrees of freedom.[12] Mathematically, $\mathcal{S}'$ is some quotient of $\mathcal{S}$ and we have a "projection"

$$\phi : \ \mathcal{S} \to \mathcal{S}'. \tag{31}$$

More precisely $\phi$ is a pivotal tensor $D$-functor between two pivotal fusion $D$-categories having the property that every object or (higher-)morphism of $\mathcal{S}'$ (up to isomorphisms) lies in the image of $\phi$. Thus, the problem of classification of gapless SPTs in (1+1)d is essentially the problem of classification of functors between fusion categories.

Such a functor specifies a functor in the opposite direction on the Drinfeld centers (i.e. the generalized charges)

$$\mathcal{Z}(\phi) : \ \mathcal{Z}(\mathcal{S}') \to \mathcal{Z}(\mathcal{S}). \tag{32}$$

If $\phi$ is not injective, i.e. if $\mathcal{S}'$ is strictly smaller than $\mathcal{S}$, then $\mathcal{Z}(\phi)$ is not surjective. The elements of $\mathcal{Z}(\mathcal{S})$ not lying in the image of $\mathcal{Z}(\phi)$ are the confined charges. A gSPT phase for $\mathcal{S}$ symmetry is thus specified by a pair

$$(\mathcal{S}', \phi), \tag{33}$$

comprising of an IR symmetry $\mathcal{S}' \le \mathcal{S}$ and a projection $\phi$ of $\mathcal{S}$ onto $\mathcal{S}'$.

There are no obstructions to deform an $\mathcal{S}$-gSPT $(\mathcal{S}'_1, \phi_1)$ into another $\mathcal{S}$-gSPT $(\mathcal{S}'_2, \phi_2)$ if there exists a functor

$$\phi_{21} : \ \mathcal{S}'_1 \to \mathcal{S}'_2, \tag{34}$$

such that

$$\phi_2 = \phi_{21} \circ \phi_1, \tag{35}$$

i.e. we can decompose $\mathcal{S} \to \mathcal{S}'_2$ as

$$\mathcal{S} \to \mathcal{S}'_1 \to \mathcal{S}'_2. \tag{36}$$

Thus, an intrinsically gapless SPT phase is one for which the IR symmetry $\mathcal{S}'$ does not admit any gapped SPT, or equivalently using (2), $\mathcal{S}'$ does not admit a functor of the form

$$\phi' : \ \mathcal{S}' \to \mathcal{S}(\mathfrak{T}_{\text{triv}}). \tag{37}$$

Let us now discuss the SymTFT description of gSPT phases. A $D$-functor $\phi$ of the form (31) allows one to regard an $\mathcal{S}'$-symmetric system as an $\mathcal{S}$-symmetric system, on which the $\mathcal{S}$ symmetry acts non-faithfully. A canonical system with $\mathcal{S}'$ symmetry is the symmetry boundary $\mathfrak{B}^{\text{sym}}_{\mathcal{S}'}$ for the SymTFT $\mathfrak{Z}(\mathcal{S}')$. Using the functor $\phi$ we can regard the topological boundary $\mathfrak{B}^{\text{sym}}_{\mathcal{S}'}$ as an $\mathcal{S}$-symmetric boundary condition of $\mathfrak{Z}(\mathcal{S}')$. This means that we can obtain $\mathfrak{B}^{\text{sym}}_{\mathcal{S}'}$ as an interval compactification of the SymTFT $\mathfrak{Z}(\mathcal{S})$ taking the following form

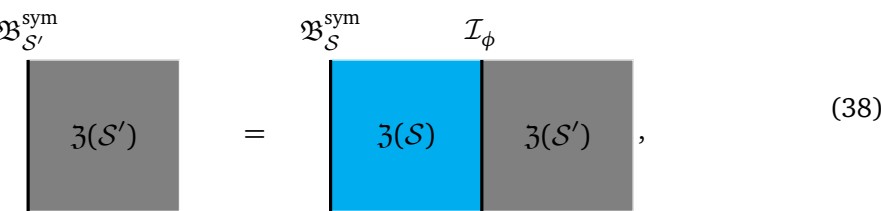

$$\tag{38}$$

---

[12]Even though the IR degrees of freedom do not transform faithfully under $\mathcal{S}$, we assume that $\mathcal{S}$ acts faithfully on the full spectrum (including states at all energies).

where $\mathcal{I}_\phi$ is a topological interface from $\mathfrak{Z}(\mathcal{S})$ to $\mathfrak{Z}(\mathcal{S}')$, associated to the functor $\phi$.

The IR theory $\mathfrak{T}^{\text{IR}}$ of a gapless system lying in the gSPT phase $(\mathcal{S}', \phi)$ is obtained by inserting a suitable physical boundary $\mathfrak{B}^{\text{phys}}_{\mathfrak{T}^{\text{IR}}}$[13] on the right and performing the full interval compactification

$$
\mathfrak{T}^{\text{IR}} \quad \bigg| \quad = \quad \boxed{\begin{array}{cc} \mathfrak{B}^{\text{sym}}_{\mathcal{S}} & \mathcal{I}_\phi \quad \mathfrak{B}^{\text{phys}}_{\mathfrak{T}^{\text{IR}}} \\ \mathfrak{Z}(\mathcal{S}) & \mathfrak{Z}(\mathcal{S}') \end{array}} \quad , \tag{39}
$$

which also captures how the $\mathcal{S}$ symmetry acts on $\mathfrak{T}^{\text{IR}}$. The physical boundary for $\mathfrak{T}^{\text{IR}}$ viewed as an $\mathcal{S}$-symmetric system is obtained by colliding $\mathcal{I}_\phi$ with $\mathfrak{B}^{\text{phys}}_{\mathfrak{T}^{\text{IR}}}$, which may be denoted as $\mathcal{I}_\phi \otimes \mathfrak{B}^{\text{phys}}_{\mathfrak{T}^{\text{IR}}}$.

We have seen that the functor $\phi$ determines a topological interface $\mathcal{I}_\phi$ from $\mathfrak{Z}(\mathcal{S})$ to $\mathfrak{Z}(\mathcal{S}')$. This interface has to satisfy a special property described below. A general topological interface $\mathcal{I}$ from $\mathfrak{Z}(\mathcal{S})$ to $\mathfrak{Z}(\mathcal{S}')$ can be acted upon by the topological boundary $\mathfrak{B}^{\text{sym}}_{\mathcal{S}}$ of $\mathfrak{Z}(\mathcal{S})$ to produce a topological boundary $\mathfrak{B}^{\text{sym}}_{\mathcal{S}} \otimes \mathcal{I}$ of $\mathfrak{Z}(\mathcal{S}')$. For an arbitrary $\mathcal{I}$, the resulting boundary $\mathfrak{B}^{\text{sym}}_{\mathcal{S}} \otimes \mathcal{I}$ hosts a multi-fusion $D$-category of topological defects localized along it. However, $\mathcal{I}_\phi$ has to be a special topological interface for which the resulting boundary $\mathfrak{B}^{\text{sym}}_{\mathcal{S}} \otimes \mathcal{I}_\phi$ hosts a fusion $D$-category, which is called $\mathcal{S}'$ above, leading to identification

$$
\mathfrak{B}^{\text{sym}}_{\mathcal{S}} \otimes \mathcal{I}_\phi = \mathfrak{B}^{\text{sym}}_{\mathcal{S}'} . \tag{40}
$$

The interface $\mathcal{I}_\phi$ has to satisfy another property following from the fact that the functor $\phi$ is surjective in a suitable sense described above. Equivalently, the functor $\mathcal{Z}(\phi)$ must have a trivial kernel. Physically this means that a non-trivial charge of $\mathcal{S}'$ has to be a non-trivial charge of $\mathcal{S}$ when an $\mathcal{S}'$-symmetric system is viewed as an $\mathcal{S}$-symmetric system. In the SymTFT description, the functor $\mathcal{Z}(\phi)$ is captured as a transformation of topological defects of $\mathfrak{Z}(\mathcal{S}')$ into topological defects of $\mathfrak{Z}(\mathcal{S})$ as they pass through the interface $\mathcal{I}_\phi$

$$
\boxed{\begin{array}{cc} & \mathcal{I}_\phi \\ \mathfrak{Z}(\mathcal{S}) & \mathfrak{Z}(\mathcal{S}') \\ \mathcal{Z}(\phi) \cdot \mathbf{Q} & \mathbf{Q} \end{array}} \quad . \tag{41}
$$

Indeed, consider a $p$-dimensional operator $\mathcal{O}_p$ in $\mathfrak{T}^{\text{IR}}$ whose $\mathcal{S}'$ charge is captured by a $(p+1)$-dimensional topological defect $\mathbf{Q}_{p+1}$ of $\mathfrak{Z}(\mathcal{S}')$, i.e. $\mathcal{O}_p$ is constructed as an interval compactification

$$
\mathcal{O}_p \quad \bigg| \quad = \quad \boxed{\begin{array}{c} \mathfrak{B}^{\text{sym}}_{\mathcal{S}'} \quad \mathfrak{B}^{\text{phys}}_{\mathfrak{T}^{\text{IR}}} \\ \mathfrak{Z}(\mathcal{S}') \\ \mathbf{Q}_{p+1} \end{array}} \quad . \tag{42}
$$

---

[13]Here we assume that a given $\mathcal{S}$-symmetric IR theory $\mathfrak{T}^{\text{IR}}$ admits a SymTFT description and thus all the dynamics of the theory are captured by the physical boundary $\mathfrak{B}^{\text{phys}}_{\mathfrak{T}^{\text{IR}}}$. Hence if a given theory $\mathfrak{T}^{\text{IR}}$ is gapless, the "gaplessness" is not imposed but rather an inherent property of the theory in question. The given physical boundary then lies in a deformation class of a non-maximal (non-Lagrangian) condensable algebra of the SymTFT which in turn specifies confined charges of the theory. This then brings us to the arguments made around (31) for the classification of $\mathcal{S}$-symmetric gapless phases using the SymTFT construction.

Note that to avoid confusion, we depict boundaries and domain walls as vertical lines while charges/topological defects of the SymTFT are depicted as horizontal lines. Using (41), we can equivalently construct $\mathcal{O}_p$ as

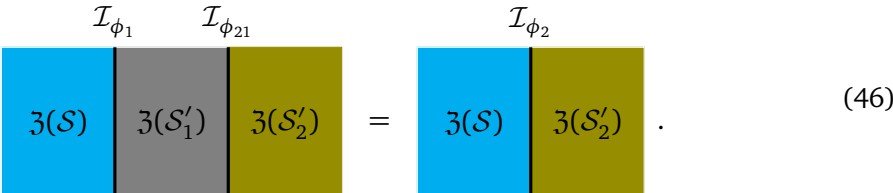

(43)

which means that the $\mathcal{S}$ charge of $\mathcal{O}_p$ is captured by the topological defect $\mathcal{Z}(\phi)\cdot\mathbf{Q}_{p+1}$ of $\mathfrak{Z}(\mathcal{S})$.

The fact that the kernel of $\mathcal{Z}(\phi)$ is trivial means that no non-identity (non-condensation) $q$-dimensional topological defect $\mathbf{Q}_q$ of $\mathfrak{Z}(\mathcal{S}')$ can end along $\mathcal{I}_\phi$, which is equivalent to the requirement

$$\mathcal{Z}(\phi)\cdot\mathbf{Q}_q \neq \mathcal{Z}(\phi)\cdot 1_q\,, \tag{44}$$

where $1_q$ is the $q$-dimensional identity defect. Thus, gSPTs for $\mathcal{S}$ symmetry are classified by topological interfaces $\mathcal{I}_\phi$ from the SymTFT $\mathfrak{Z}(\mathcal{S})$ to other SymTFTs $\mathfrak{Z}(\mathcal{S}')$, satisfying the two conditions (40) and (44).

A gSPT phase described by a topological interface $\mathcal{I}_{\phi_1}$ from $\mathfrak{Z}(\mathcal{S})$ to $\mathfrak{Z}(\mathcal{S}'_1)$ has no obstruction to being deformed to a gSPT phase described by a topological interface $\mathcal{I}_{\phi_2}$ from $\mathfrak{Z}(\mathcal{S})$ to $\mathfrak{Z}(\mathcal{S}'_2)$ if there exists a topological interface $\mathcal{I}_{\phi_{21}}$ from $\mathfrak{Z}(\mathcal{S}'_1)$ to $\mathfrak{Z}(\mathcal{S}'_2)$ such that

$$\mathcal{I}_{\phi_1}\otimes\mathcal{I}_{\phi_{21}}=\mathcal{I}_{\phi_2}\,, \tag{45}$$

or pictorially we have

(46)

Let us now specialize to (1+1)d gSPT phases. The SymTFT description of (1+1)d gSPT phases for non-anomalous group symmetries was discussed in [9, 25]. Here we discuss the SymTFT setup for (1+1)d gSPT phases with arbitrary, possibly non-invertible, symmetries described by fusion categories. The general setup is a special case of the club quiches discussed recently in [10].

**Characterization by condensable algebras.** In (1+1)d, a topological interface $\mathcal{I}_\phi$ corresponding to a gSPT phase for fusion category symmetry $\mathcal{S}$ can be described in terms of a special type of (not necessarily Lagrangian) condensable algebra $\mathcal{A}_\phi$ in the Drinfeld center $\mathcal{Z}(\mathcal{S})$, as explained below. A topological interface from $\mathfrak{Z}(\mathcal{S})$ to $\mathfrak{Z}(\mathcal{S}')$ can be described as a topological boundary condition of the folded TQFT $\mathfrak{Z}(\mathcal{S})\boxtimes\overline{\mathfrak{Z}(\mathcal{S}')}$ whose anyons are described by the MTC $\mathcal{Z}(\mathcal{S})\boxtimes\overline{\mathcal{Z}(\mathcal{S}')}$. Thus $\mathcal{I}_\phi$ is described by a Lagrangian algebra $\mathcal{L}_\phi$ in $\mathcal{Z}(\mathcal{S})\boxtimes\overline{\mathcal{Z}(\mathcal{S}')}$. Let us express it as

$$\mathcal{L}_\phi=\bigoplus_{a,a'}n_{aa'}\mathbf{Q}_a\overline{\mathbf{Q}'_{a'}}\,, \tag{47}$$

where $\mathbf{Q}_a$ are simple anyons of $\mathfrak{Z}(\mathcal{S})$, $\mathbf{Q}'_{a'}$ are simple anyons of $\mathfrak{Z}(\mathcal{S}')$ and $n_{aa'}$ are non-negative integers. This expression concretely captures the information of the functor $\mathcal{Z}(\phi)$ describing

what happens to anyons of $\mathfrak{Z}(\mathcal{S}')$ as they pass through the interface $\mathcal{I}_\phi$. We have

$$\mathcal{Z}(\phi) \cdot \mathbf{Q}'_{a'} = \bigoplus_a n_{aa'} \mathbf{Q}^*_a, \tag{48}$$

where $\mathbf{Q}^*_a$ is the orientation reversed dual of $\mathbf{Q}_a$. Pictorially, we have

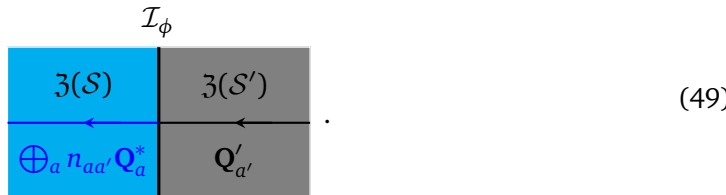

$$. \tag{49}$$

The condensable algebra $\mathcal{A}_\phi$ is specified by

$$\mathcal{Z}(\phi) \cdot 1 = \mathcal{A}^*_\phi, \tag{50}$$

where 1 is the trivial anyon of $\mathfrak{Z}(\mathcal{S}')$. That is, $\mathcal{A}_\phi$ captures the anyons of $\mathfrak{Z}(\mathcal{S})$ that end along $\mathcal{I}_\phi$. Note that for $\mathcal{I}_\phi$ to correspond to a gSPT phase, no anyons of $\mathfrak{Z}(\mathcal{S}')$ are allowed to end along $\mathcal{I}_\phi$, i.e. we have

$$n_{1a'} = \delta_{1a'}. \tag{51}$$

A Lagrangian algebra $\mathcal{L}_\phi$ satisfying the above condition is determined fully by the condensable algebra $\mathcal{A}_\phi$ up to the action of 0-form symmetries of $\mathfrak{Z}(\mathcal{S}')$ which do not change physical results.

For the condensable algebra $\mathcal{A}_\phi$ to describe a gSPT phase, we also need to satisfy condition (40), which translates into the requirement that there should not be a non-identity anyon of $\mathfrak{Z}(\mathcal{S})$ that can end both along $\mathfrak{B}^{\text{sym}}_\mathcal{S}$ and $\mathcal{I}_\phi$, i.e.

$$\mathcal{L}^{\text{sym}}_\mathcal{S} \cap \mathcal{A}_\phi = 1. \tag{52}$$

Otherwise, compactifying such an anyon stretched between $\mathfrak{B}^{\text{sym}}_\mathcal{S}$ and $\mathcal{I}_\phi$ produces a non-identity topological local operator along the resulting topological boundary $\mathfrak{B}^{\text{sym}}_\mathcal{S} \otimes \mathcal{I}_\phi$ of $\mathfrak{Z}(\mathcal{S}')$, which means that the topological defects along the boundary form a multi-fusion category, in violation of condition (40).

In conclusion, while (1+1)d gapped SPT phases with $\mathcal{S}$ symmetry are characterized by **Lagrangian algebras** in $\mathcal{Z}(\mathcal{S})$ having trivial intersection with $\mathcal{L}^{\text{sym}}_\mathcal{S}$, the (1+1)d gapless SPT phases with $\mathcal{S}$ symmetry are characterized by (not necessarily Lagrangian) **condensable algebras** in $\mathcal{Z}(\mathcal{S})$ having trivial intersection with $\mathcal{L}^{\text{sym}}_\mathcal{S}$. The canonical gSPT corresponds to $\mathcal{A}_\phi = 1$, and gSPTs showing the same confined charges as gapped SPTs correspond to $\mathcal{A}_\phi$ being Lagrangian.

The confined $\mathcal{S}$ charges for a gSPT phase correspond to the anyons $\mathbf{Q}_a$ of $\mathfrak{Z}(\mathcal{S})$ that do not appear in $\mathcal{L}_\phi$, i.e. for which we have

$$n_{aa'} = 0, \qquad \forall\, a'. \tag{53}$$

On the other hand, the order parameters are local operators of the gapless system that appear as topological local operators of the IR effective theory (but are not topological in the full system). The charges of order parameters are thus captured by the condensable algebra $\mathcal{A}_\phi$.

In terms of condensable algebras, a gSPT $\mathcal{A}_{\phi_1}$ can be deformed into a gSPT $\mathcal{A}_{\phi_2}$ if

$$\mathcal{A}_{\phi_1} \subset \mathcal{A}_{\phi_2}, \tag{54}$$

i.e. if $\mathcal{A}_{\phi_1}$ is a subalgebra of $\mathcal{A}_{\phi_2}$. An igSPT is a gSPT such that it cannot be deformed to an SPT phase, i.e. the associated condensable algebra is not contained in any Lagrangian algebra that has a trivial intersection with the symmetry Lagrangian algebra.

Finally, we note that the categorical framework constrains the possible phases and transitions, it does not by itself determine the spectrum or dispersion of low-energy excitations. In gapped systems, the algebraic classification of topological charges is sufficient to infer spectral gaps. In contrast, for gapless systems, the nature of the excitations, such as their scaling or spectrum, requires additional physical input, such as a specific lattice model or a CFT description of the IR fixed point.

## 3.1 Example: $\mathbb{Z}_4$ igSPT

A well-known example of a (1+1)d igSPT phase is for $\mathbb{Z}_4$ symmetry [22, 24], for which the fusion category is $\mathcal{S} = \text{Vec}_{\mathbb{Z}_4}$. The anyons of the SymTFT $\mathfrak{Z}(\text{Vec}_{\mathbb{Z}_4})$ are

$$\mathcal{Z}(\text{Vec}_{\mathbb{Z}_4}) = \left\{ e^k m^l \mid k, l \in \{0, 1, 2, 3\} \right\}, \tag{55}$$

where $e^k$ and $m^l$ are bosons, with the braiding between $e$ and $m$ given by the fourth root of unity $i$. The symmetry Lagrangian algebra is

$$\mathcal{L}^{\text{sym}}_{\text{Vec}_{\mathbb{Z}_4}} = 1 \oplus e \oplus e^2 \oplus e^3. \tag{56}$$

The various condensable algebras are

$$\begin{aligned}
\mathcal{A}_1 &= 1, \\
\mathcal{A}_{e^2} &= 1 \oplus e^2, \\
\mathcal{A}_{m^2} &= 1 \oplus m^2, \\
\mathcal{A}_{e^2 m^2} &= 1 \oplus e^2 m^2, \\
\mathcal{A}_e &= 1 \oplus e \oplus e^2 \oplus e^3, \\
\mathcal{A}_m &= 1 \oplus m \oplus m^2 \oplus m^3, \\
\mathcal{A}_{em} &= 1 \oplus e^2 \oplus m^2 \oplus e^2 m^2.
\end{aligned} \tag{57}$$

The first four are non-Lagrangian, while the last three are Lagrangian. Out of these the gSPTs are the ones satisfying the condition (52) of trivial intersection with $\mathcal{L}^{\text{sym}}_{\text{Vec}_{\mathbb{Z}_4}}$ are

$$\text{gSPTs for } \mathbb{Z}_4 \text{ Symmetry} = \{\mathcal{A}_1, \mathcal{A}_{m^2}, \mathcal{A}_{e^2 m^2}, \mathcal{A}_m\}, \tag{58}$$

while there is only one gapped SPT corresponding to $\mathcal{A}_m$. Looking at which of these are subalgebras of other algebras, the various possible deformation patterns are determined to be

$$\begin{aligned}
\mathcal{A}_1 &\to \mathcal{A}_{m^2} \to \mathcal{A}_m, \\
\mathcal{A}_1 &\to \mathcal{A}_{e^2 m^2}.
\end{aligned} \tag{59}$$

Thus $\mathcal{A}_{m^2}$ is a non-intrinsic gSPT, while $\mathcal{A}_{e^2 m^2}$ is an igSPT.

The IR symmetries $\mathcal{S}'$ for these two gSPTs are

$$\mathcal{S}'_{m^2} = \text{Vec}_{\mathbb{Z}_2}, \qquad \mathcal{S}'_{e^2 m^2} = \text{Vec}^{\omega}_{\mathbb{Z}_2}, \tag{60}$$

i.e. $\mathcal{S}'_{m^2}$ is non-anomalous $\mathbb{Z}_2$ symmetry, while $\mathcal{S}'_{e^2 m^2}$ is $\mathbb{Z}_2$ symmetry with a non-trivial 't Hooft anomaly

$$\omega \in H^3(\mathbb{Z}_2, U(1)) = \mathbb{Z}_2. \tag{61}$$

That these are the IR symmetries in these cases can be shown by exhibiting appropriate Lagrangian algebras $\mathcal{L}_{m^2}$ and $\mathcal{L}_{e^2m^2}$ completing $\mathcal{A}_{m^2}$ and $\mathcal{A}_{e^2m^2}$. These can be taken to be

$$
\begin{aligned}
\mathcal{L}_{m^2} = {}& 1 \oplus m\,\overline{m}' \oplus m^2 \oplus m^3\,\overline{m}' \oplus e^2\,\overline{e}' \oplus e^2 m\,\overline{e}'\overline{m}' \\
& \oplus e^2 m^2\,\overline{e}' \oplus e^2 m^3\,\overline{e}'\overline{m}' \in \mathcal{Z}(\mathsf{Vec}_{\mathbb{Z}_4}) \boxtimes \overline{\mathcal{Z}(\mathsf{Vec}_{\mathbb{Z}_2})}, \\
\mathcal{L}_{e^2m^2} = {}& 1 \oplus em\,\overline{s} \oplus e^2 m^2 \oplus e^3 m^3\,\overline{s} \oplus em^3\,s \oplus e^2\,s\overline{s} \\
& \oplus e^3 m\,s \oplus m^2\,s\overline{s} \in \mathcal{Z}(\mathsf{Vec}_{\mathbb{Z}_4}) \boxtimes \overline{\mathcal{Z}(\mathsf{Vec}^{\omega}_{\mathbb{Z}_2})},
\end{aligned}
\tag{62}
$$

as reproduced from [10], where we have represented the anyons in $\mathcal{Z}(\mathsf{Vec}_{\mathbb{Z}_2})$ as

$$
\mathcal{Z}(\mathsf{Vec}_{\mathbb{Z}_2}) = \{1, e', m', e'm'\},
\tag{63}
$$

and the anyons in $\mathcal{Z}(\mathsf{Vec}^{\omega}_{\mathbb{Z}_2})$ as

$$
\mathcal{Z}(\mathsf{Vec}^{\omega}_{\mathbb{Z}_2}) = \{1, s, \overline{s}, s\overline{s}\},
\tag{64}
$$

where $s$ and $\overline{s}$ are semion and anti-semion respectively, and $s\overline{s}$ is a boson, with the braiding between $s$ and $s\overline{s}$ being a minus sign.

The non-anomalous $\mathbb{Z}_2$ symmetry admits a gapped SPT phase, while the anomalous $\mathbb{Z}_2$ symmetry does not. This is another way to understand why the gSPT $\mathcal{A}_{e^2m^2}$ is intrinsic and cannot be deformed to a gapped SPT for $\mathbb{Z}_4$ symmetry, but the gSPT $\mathcal{A}_{m^2}$ is not intrinsic and has no obstruction to being deformed to a gapped SPT.

The corresponding functors

$$
\begin{aligned}
\phi_{m^2} &: \ \mathsf{Vec}_{\mathbb{Z}_4} \to \mathsf{Vec}_{\mathbb{Z}_2}, \\
\phi_{e^2m^2} &: \ \mathsf{Vec}_{\mathbb{Z}_4} \to \mathsf{Vec}^{\omega}_{\mathbb{Z}_2},
\end{aligned}
\tag{65}
$$

are described in detail respectively in sections IV.B.2 and IV.B.3 of [10].

As a generalized charge, the anyon $e^k m^l$ in $\mathcal{Z}(\mathsf{Vec}_{\mathbb{Z}_4})$ describes a local operator in $P^l$-twisted sector carrying charge $k$ under $\mathbb{Z}_4$, where $P$ is the generator of $\mathbb{Z}_4$ symmetry. The charges of the order parameters for the gSPT phases are captured by the associated condensable algebras. Thus, the order parameter for the gSPT phase corresponding to $\mathcal{A}_{m^2}$ is an uncharged local operator in $P^2$-twisted sector, while the order parameter for the gSPT phase corresponding to $\mathcal{A}_{e^2m^2}$ is a local operator in $P^2$-twisted sector carrying charge 2 under $\mathbb{Z}_4$.

The confined charges for these gSPT phases are

$$
\begin{aligned}
Q_{m^2} &= \{e, em, em^2, em^3, e^3, e^3 m, e^3 m^2, e^3 m^3\}, \\
Q_{e^2m^2} &= \{m, m^3, e, em^2, e^2 m, e^2 m^3, e^3, e^3 m^2\},
\end{aligned}
\tag{66}
$$

which are the charges not lying in the images of the functors $\mathcal{Z}(\phi)$ on the centers, following from the Lagrangian algebras (62)

$$
\begin{aligned}
\mathcal{Z}(\phi_{m^2}): \quad & \mathcal{Z}(\mathsf{Vec}_{\mathbb{Z}_2}) \to \mathcal{Z}(\mathsf{Vec}_{\mathbb{Z}_4}), \\
& 1 \mapsto 1 \oplus m^2, \\
& e' \mapsto e^2 \oplus e^2 m^2, \\
& m' \mapsto m \oplus m^3, \\
& e'm' \mapsto e^2 m \oplus e^2 m^3, \\
\mathcal{Z}(\phi_{e^2m^2}): \quad & \mathcal{Z}(\mathsf{Vec}^{\omega}_{\mathbb{Z}_2}) \to \mathcal{Z}(\mathsf{Vec}_{\mathbb{Z}_4}), \\
& 1 \mapsto 1 \oplus e^2 m^2, \\
& s \mapsto em \oplus e^3 m^3, \\
& \overline{s} \mapsto em^3 \oplus e^3 m, \\
& s\overline{s} \mapsto e^2 \oplus m^2.
\end{aligned}
\tag{67}
$$

The associated club quiches are as follows:

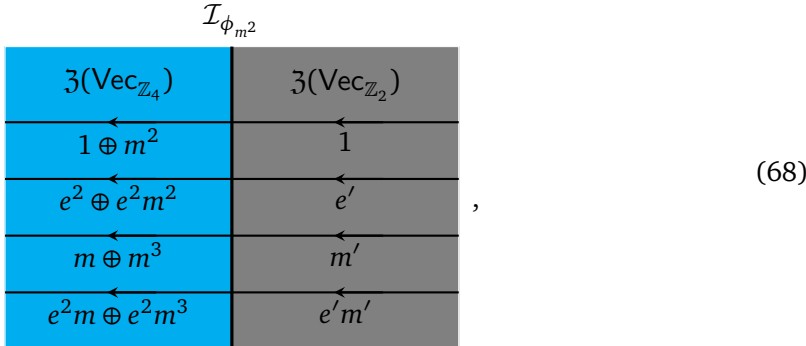

$$
\mathcal{I}_{\phi_{m^2}} \tag{68}
$$

$$
\mathcal{I}_{\phi_{e^2m^2}} \tag{69}
$$

## 3.2 Example: $\mathsf{Rep}(D_8)$ igSPT

Let us now consider a non-invertible symmetry, namely the representation category of the non-Abelian group $D_8$

$$
\mathcal{S} = \mathsf{Rep}(D_8), \tag{70}
$$

which corresponds to the Lagrangian algebra (26). There are several condensable algebras that satisfy the trivial intersection with the symmetry Lagrangian algebra (52). We will here discuss an algebra corresponding to an **intrinsically** gapless SPT. Other algebras are discussed in section 5.2. The transformation properties of local operators carrying $\mathsf{Rep}(D_8)$ charges are discussed in detail after (27).

The condensable algebra we consider is

$$
\mathcal{A}_\phi = 1 \oplus e_{RG} \oplus e_{GB} \oplus e_{RB}, \tag{71}
$$

satisfying the gSPT condition (52). This algebra appears inside the Lagrangian algebra

$$
\begin{aligned}
\mathcal{L}_\phi = {} & 1 \oplus e_{RG} \oplus e_{GB} \oplus e_{RB} \\
& \oplus s\bar{s} \left[ e_R \oplus e_G \oplus e_{RGB} \oplus e_B \right] \\
& \oplus 2 s\bar{s}_{RGB} \oplus 2 \bar{s} s_{RGB} \\
& \in \mathcal{Z}(\mathsf{Rep}(D_8)) \boxtimes \overline{\mathcal{Z}(\mathsf{Vec}_{\mathbb{Z}_2}^\omega)},
\end{aligned} \tag{72}
$$

meaning that the IR symmetry is anomalous $\mathbb{Z}_2$

$$
\mathcal{S}' = \mathsf{Vec}_{\mathbb{Z}_2}^\omega, \tag{73}
$$

and hence $\mathcal{A}_\phi$ describes an intrinsic gSPT phase, as $\mathcal{S}'$ does not allow gapped SPTs. Equivalently, one can check that $\mathcal{A}_\phi$ is not a subalgebra of any of the Lagrangian algebras (27) corresponding to gapped SPT phases for $\mathsf{Rep}(D_8)$ symmetry.

From (71), we find that the order parameters for this gSPT phase are:

- a local operator in $RG$-twisted sector which is uncharged under $\mathrm{Rep}(D_8)$,

- a local operator in $R$-twisted sector which transforms by a sign $-1$ under $B \in \mathrm{Rep}(D_8)$,

- and a local operator in $G$-twisted sector which transforms by a sign $-1$ under $B \in \mathrm{Rep}(D_8)$.

From (72), we compute the map of the charges to be

$$
\begin{aligned}
\mathcal{Z}(\phi): \quad & \mathcal{Z}(\mathrm{Vec}^\omega_{\mathbb{Z}_2}) \to \mathcal{Z}(\mathrm{Rep}(D_8)), \\
& 1 \mapsto 1 \oplus e_{RG} \oplus e_{GB} \oplus e_{RB}, \\
& s \mapsto 2 s_{RGB}, \\
& \bar{s} \mapsto 2 \bar{s}_{RGB}, \\
& s\bar{s} \mapsto e_R \oplus e_G \oplus e_{RGB} \oplus e_B,
\end{aligned}
\tag{74}
$$

which implies that the confined $\mathrm{Rep}(D_8)$ charges exhibited by this igSPT are

$$
Q = \{ m_{GB}, m_G, f_G, f_{GB}, m_{RB}, m_R, f_R, f_{RB} \}.
\tag{75}
$$

The associated quiches are:

$$
\mathcal{I}_\phi
$$

| $\mathfrak{Z}(\mathrm{Rep}(D_8))$ | $\mathfrak{Z}(\mathrm{Vec}^\omega_{\mathbb{Z}_2})$ |
|---|---|
| $1 \oplus e_{RG} \oplus e_{GB} \oplus e_{RB}$ | $1$ |
| $2 s_{RGB}$ | $s$ |
| $2 \bar{s}_{RGB}$ | $\bar{s}$ |
| $e_R \oplus e_G \oplus e_B \oplus e_{RGB}$ | $s\bar{s}$ |

$$\tag{76}$$

$$
\mathcal{I}_{\phi_1} \qquad \mathcal{I}_{\phi_{e^2 m^2}}
$$

| $\mathfrak{Z}(\mathrm{Rep}(D_8))$ | $\mathfrak{Z}(\mathrm{Vec}_{\mathbb{Z}_4})$ | $\mathfrak{Z}(\mathrm{Vec}^\omega_{\mathbb{Z}_2})$ |
|---|---|---|
| $1 \oplus e_{RG}$ | $1$ | $1$ |
| $e_{GB} \oplus e_{RB}$ | $e^2 m^2$ | $1$ |
| $s_{RGB}$ | $em$ | $s$ |
| $s_{RGB}$ | $e^3 m^3$ | $s$ |
| $\bar{s}_{RGB}$ | $em^3$ | $\bar{s}$ |
| $\bar{s}_{RGB}$ | $e^3 m$ | $\bar{s}$ |
| $e_R \oplus e_G$ | $e^2$ | $s\bar{s}$ |
| $e_B \oplus e_{RGB}$ | $m^2$ | $s\bar{s}$ |

$$\tag{77}$$

**Generalization: $\mathsf{Rep}(D_{8m})$ igSPTs.**  There is a generalization of the above igSPT phase to

$$\mathcal{S} = \mathsf{Rep}(D_{8m}), \tag{78}$$

where

$$D_{8m} = \mathbb{Z}_{4m} \rtimes \mathbb{Z}_2 \tag{79}$$

is the dihedral group of order $8m$. We represent elements of $D_{8m}$ as

$$D_{8m} = \left\{ a^i, a^j x \mid i, j \in \{0, 1, 2, \ldots, 4m-1\} \right\}, \tag{80}$$

such that we have

$$a^{4m} = 1, \qquad x^2 = 1, \qquad xax = a^{4m-1}. \tag{81}$$

The irreducible representations of $D_{8m}$, which are the symmetry generators, are

$$1, \quad 1_a, \quad 1_x, \quad 1_{ax}, \quad E_k, \qquad k \in \{1, 2, \ldots, 2m-1\}, \tag{82}$$

where 1 is the trivial one-dimensional representation, $1_a$ is a one-dimensional representation in which $a$ acts trivially but $x, ax$ act by a non-trivial sign, $1_x$ is a one-dimensional representation in which $x$ acts trivially but $a, ax$ act by a non-trivial sign, $1_{ax}$ is a one-dimensional representation in which $ax$ acts trivially but $a, x$ act by a non-trivial sign, and $E_k$ is a two-dimensional representation comprised of basis vectors $v_1, v_2$ having transformations

$$\begin{aligned} a: \quad & v_1 \to \omega_{4m}^k v_1, \qquad v_2 \to \omega_{4m}^{-k} v_2, \\ x: \quad & v_1 \to v_2, \qquad\qquad v_2 \to v_1, \end{aligned} \tag{83}$$

where $\omega_{4m} = \exp(2\pi i/4m)$.

The conjugacy classes are labeled as

$$\begin{aligned} & 1 = \{1\}, \quad a^{2m} = \{a^{2m}\}, \quad a^k = \{a^k, a^{2n-k}\}, \\ & x = \{x, a^2 x, a^4 x, \ldots, a^{4m-2} x\}, \\ & ax = \{ax, a^3 x, a^5 x, \ldots, a^{4m-1} x\}, \end{aligned} \tag{84}$$

where $1 \le k \le 2m-1$, with centralizers

$$H_1 = H_{a^{2m}} = D_{8m}, \qquad H_{a^k} = \mathbb{Z}_{4m}, \qquad H_x \cong H_{ax} = \mathbb{Z}_2^2. \tag{85}$$

The Drinfeld center $\mathcal{Z}(\mathsf{Rep}(D_{8m}))$ is thus comprised of anyons

$$(1, R), \quad (a^{2m}, R), \quad (x, s, s'), \quad (ax, s, s'), \quad (a^k, \omega_{4m}^p), \tag{86}$$

where $R$ is an irreducible representation of $D_{8m}$, $s, s' \in \{+, -\}$ capturing irreducible representations of $\mathbb{Z}_2^2$, and $p \in \{0, 1, 2, \ldots, 4m-1\}$ capturing irreducible representations of $\mathbb{Z}_{4m}$. Our convention is that $(x, +, s')$ and $(ax, +, s')$ are bosons, while $(x, -, s')$ and $(ax, -, s')$ are fermions.

The symmetry Lagrangian algebra is

$$\mathcal{L}_{\mathsf{Rep}(D_{8m})}^{\mathrm{sym}} = (1, 1) \oplus (a^{2m}, 1) \oplus (x, +, +) \oplus (ax, +, +) \bigoplus_{k=1}^{2m-1} (a^k, 1). \tag{87}$$

The following non-Lagrangian condensable algebra

$$\mathcal{A}_\phi = (1, 1) \oplus (1, 1_a) \oplus (a^{2m}, 1_x) \oplus (a^{2m}, 1_{ax}), \tag{88}$$

has trivial intersection with $\mathcal{L}^{\text{sym}}_{\text{Rep}(D_{8m})}$, and thus gives rise to a gSPT phase with $\mathcal{S} = \text{Rep}(D_{8m})$ symmetry. We claim that the symmetry acting faithfully on IR degrees of freedom is

$$\mathcal{S}' = \text{Vec}^\omega_{\mathbb{Z}_{2m}}, \tag{89}$$

where

$$\omega = m \in H^3(\mathbb{Z}_{2m}, U(1)) = \mathbb{Z}_{2m}, \tag{90}$$

is the anomaly of order two. Thus the gSPT under discussion is actually an intrinsic gSPT.

The Drinfeld center $\mathcal{Z}(\text{Vec}^\omega_{\mathbb{Z}_{2m}})$ is comprised of anyons

$$e^i m^j, \qquad i, j \in \{0, 1, 2, \ldots, 2m-1\}, \tag{91}$$

where $e^i$ are bosons, the spins of $m^i$ are

$$\theta(m^i) = \omega^{i^2}_{4m}, \tag{92}$$

and the braiding between $e$ and $m$ is $\omega^2_{4m}$. A Lagrangian algebra completion $\mathcal{L}_\phi$ of $\mathcal{A}_\phi$ is

$$\mathcal{L}_\phi = \bigoplus_{i,j=0}^{2m-1} \left[ (a^{2j+i}, \omega^i_{4m})_\sigma + (a^{2m-2j+i}, \omega^{2m+i}_{4m})_\sigma \right] \overline{e^j m^i}, \tag{93}$$

where we define

$$(a^k, \omega^p_{4m})_\sigma = \begin{cases} (a^k, \omega^p_{4m}), & 1 \le k \le 2m-1, \\ (a^{4m-k}, \omega^{4m-p}_{4m}), & 2m+1 \le k \le 4m-1, \\ (1, 1) \oplus (1, 1_a), & k = 0, \ p = 0, \\ (1, 1_x) \oplus (1, 1_{ax}), & k = 0, \ p = 2m, \\ (a^{2m}, 1) \oplus (a^{2m}, 1_a), & k = 2m, \ p = 0, \\ (a^{2m}, 1_x) \oplus (a^{2m}, 1_{ax}), & k = 2m, \ p = 2m. \end{cases} \tag{94}$$

## 4 SSB phases

Now let us study gapped and gapless phases in which the symmetry $\mathcal{S}$ is spontaneously broken.[14] We will restrict our attention to (1+1)d as in this paper we do not study systems with topological order, for simplicity. Indeed, a fusion $D$-category $\mathcal{S}$ for $D \ge 2$ generically contains (possibly non-invertible) $p$-form symmetries for $p \ge 1$, which if spontaneously broken, lead to topological order in the IR. In (1+1)d, since there is no topological order, the definition of a gapless or gapped SSB phase is taken to be simply a phase which is not a gapless or gapped SPT phase respectively.

### 4.1 Gapped SSB phases

A (1+1)d system in a gapped SSB phase, which would simply be referred to as an SSB phase from now on, has multiple gapped vacua $i \in \{0, 1, 2, \ldots, n-1\}$ in the IR. The excited states in each vacuum $i$ cost at least an energy $\Delta^{(i)} > 0$. Moreover, we can have domain wall excitations that go from vacuum $i$ to vacuum $j$ costing at least an energy $\Delta^{(ij)} > 0$. The symmetry fusion category $\mathcal{S}$ mixes the vacua into each other.

---

[14]We note that the correspondence between symmetry-breaking (SSB) gapped phases and indecomposable module categories over $\mathcal{S}$ was first clearly formulated in [3], where (1+1)d gapped phases under fusion-category symmetry are classified via module-category data, and further explored in continuum field-theoretic examples in [4].

The IR effective theory is a TQFT $\mathfrak{T}^{\text{IR}}$ with $n$ vacua that describes physics below all these energy scales. Each vacuum $i$ itself is governed by an invertible 2d TQFT $\mathfrak{T}_i^{\text{IR}}$ with a single vacuum, which is an Euler term. These Euler terms are continuous parameters, so naively one would think that they are invisible at the level of phases, but the fact that the system has to be $\mathcal{S}$ symmetric rules out deformations that change the relative Euler terms between two different vacua. If $\mathcal{S}$ is a (possibly anomalous) group symmetry, then these relative Euler terms are all trivial, and the IR TQFT $\mathfrak{T}^{\text{IR}}$ is just a sum of trivial TQFTs $\mathfrak{T}_i^{\text{IR}}$. On the other hand, when $\mathcal{S}$ is a non-invertible symmetry, then the relative Euler terms can be non-trivial, and hence one cannot choose all $\mathfrak{T}_i^{\text{IR}}$ to be trivial TQFTs (i.e. invertible 2d TQFTs with trivial Euler terms). An example of this phenomenon studied in detail in [2] is provided by $\mathcal{S} = \text{Rep}(S_3)$ for which there are two SSB phases, both of which have three vacua, which are called $\text{Rep}(S_3)/\mathbb{Z}_2$ SSB and $\text{Rep}(S_3)$ SSB phases in [2]. The three vacua of $\text{Rep}(S_3)/\mathbb{Z}_2$ SSB phase have trivial relative Euler terms between them, while the three vacua of $\text{Rep}(S_3)$ SSB phase have non-trivial relative Euler terms between them. These phases will be discussed in more detail below.

We emphasize that the presence of non-trivial Euler terms does not reduce to whether the anyons labelling vacua have quantum dimension greater than one. Rather, non-trivial Euler terms arise when deformations preserving the symmetry $\mathcal{S}$ permute degenerate vacua in a way incompatible with trivial TQFT data between them. In this sense, the phenomenon is sharper than quantum dimension alone and encodes information about symmetry action across the IR vacuum sector.

Just like SPT phases, SSB phases can also be distinguished in terms of confined charges, i.e. the charges under $\mathcal{S}$ that are not realized by states/operators in the IR theory. In fact, the confined charges differentiate all SPT and SSB phases from each other, and all (1+1)d $\mathcal{S}$-symmetric gapped phases are classified by their confined charges under $\mathcal{S}$.

Mathematically, what differentiates SSB phases from SPT phases is the multiplicity of vacua. An SSB phase corresponds to an indecomposable module category over the symmetry category $\mathcal{S}$ that is not equivalent to Vec [3,4], in contrast to SPT phases, which are classified by fiber functors $\mathcal{S} \to \text{Vec}$. Physically, one can also describe these SSB phases via functors $\mathcal{S} \to \mathcal{M}$, where $\mathcal{M}$ is a multi-fusion category that encodes domain wall and defect operators between the symmetry-broken vacua.[15] In this setting, the Vec-valued functors describe symmetry-preserving SPTs, while more general targets reflect vacuum degeneracy due to spontaneous symmetry breaking. The multi-fusion category $\mathcal{M}$ describes the topological defects of spacetime dimension 1 and 0, i.e. topological line operators and local operators of the IR 2d TQFT $\mathfrak{T}^{\text{IR}}$, and the functor describes the subset of line operators that generate the symmetry $\mathcal{S}$. Essentially the multi-fusion category is comprised of line operators $1_{ij}$ that transforms vacuum $i$ to vacuum $j$ while annihilating all other vacua. The only non-zero fusion rules of these line operators are

$$1_{ij} \otimes 1_{jk} = 1_{ik}. \tag{95}$$

In addition to this there is information about the pivotal structure, or physically speaking relative Euler terms, captured in the quantum dimensions of $1_{ij}$. If the quantum dimension of $1_{ij}$ is 1, then there is no relative Euler term between vacua $i$ and $j$. On the other hand, if the quantum dimension of $1_{ij}$ is not equal to 1, then there is a relative Euler term between vacua $i$ and $j$ specified by the quantum dimension. We refer the reader to [2] for more details. The

---

[15]While we use functors $\mathcal{S} \to \mathcal{M}$ into a multi-fusion category $\mathcal{M}$ to describe topological line or domain wall operators between the broken vacua, it is important to note that this formalism assumes more structure than is strictly present: general module categories over $\mathcal{S}$ do not themselves carry fusion rules, and so are not fusion or multi-fusion categories in their own right. In this sense, the multi-fusion target should be understood as a physical realization rather than the mathematical definition of symmetry-breaking phases.

functor then specifies each symmetry generator $S_a \in \mathcal{S}$ as some combination of these lines

$$S_a = \bigoplus_{i,j} n_a^{ij} 1_{ij} \, . \tag{96}$$

In terms of the SymTFT, an $\mathcal{S}$-symmetric gapped SSB phase is specified by a Lagrangian algebra $\mathcal{L}_{\text{phys}}$ that does not satisfy the condition (7), instead satisfying

$$\mathcal{L}_{\text{phys}} \cap \mathcal{L}_{\mathcal{S}}^{\text{sym}} \supsetneq 1 \, , \tag{97}$$

i.e. the intersection of $\mathcal{L}_{\text{phys}}$ with the symmetry Lagrangian $\mathcal{L}_{\mathcal{S}}^{\text{sym}}$ contains non-identity anyons (along with the identity anyon). The generalized charges realized in the IR are the anyons appearing in $\mathcal{L}_{\text{phys}}$, which correspond to the order parameters of the phase. The anyons not appearing in $\mathcal{L}_{\text{phys}}$ are the confined charges associated to the SSB phase, which correspond to gapped excitations of the phase. The number of vacua involved in the SSB phase are the anyons determined in terms of $\mathcal{L}_{\text{phys}}$ and $\mathcal{L}_{\mathcal{S}}^{\text{sym}}$ as follows. Let $\mathbf{Q}_a \in \mathcal{Z}(\mathcal{S})$ be an anyon and $n_{\text{phys}}^a, n_{\text{sym}}^a$ be its multiplicities in the Lagrangian algebras $\mathcal{L}_{\text{phys}}$ and $\mathcal{L}_{\mathcal{S}}^{\text{sym}}$ respectively. Then, the number of vacua in the SSB phase are

$$n = \sum_a n_{\text{phys}}^{a^*} n_{\text{sym}}^a \, , \tag{98}$$

where $\mathbf{Q}_{a^*}$ is the dual of the anyon $\mathbf{Q}_a$. This follows from the fact that $n_{\text{phys}}^a$ describes the dimension of the vector space of local operators living at the end of $\mathbf{Q}_a$ along $\mathfrak{B}^{\text{phys}}$ and $n_{\text{sym}}^a$ describes the dimension of the vector space of local operators living at the end of $\mathbf{Q}_a$ along $\mathfrak{B}_{\mathcal{S}}^{\text{sym}}$, and so the sandwich interval compactification of $\mathbf{Q}_a$ produces $n_{\text{phys}}^{a^*} n_{\text{sym}}^a$ dimensional vector space of (untwisted sector) topological local operators. Finally, the number of vacua is the same as the dimension of the space of topological local operators. Other information about the SSB phase, like the determination of $n_a^{ij}$ in (96) requires a more detailed analysis as discussed in [2]. The charges of order parameters are encoded in $\mathcal{L}_{\text{phys}}$.

## 4.2 Example: $\mathsf{Rep}(S_3)$ SSB

Let us now describe the examples for symmetry

$$\mathcal{S} = \mathsf{Rep}(S_3) \, , \tag{99}$$

discussed above. We denote the elements of $S_3$ as

$$S_3 = \{1, a, a^2, b, ab, a^2 b\} \, , \tag{100}$$

with multiplication rule

$$a^3 = 1 = b^2 \, , \qquad ba = a^2 b \, . \tag{101}$$

Its irreducible representations, which are taken to be the symmetry generators, are

$$1 \, , \quad 1_- \, , \quad E \, , \tag{102}$$

where $1$ is the trivial one-dimensional representation, $1_-$ is a one-dimensional representation in which $a$ acts trivially but $b$, $ab$ and $a^2 b$ act by a non-trivial sign, and $E$ is a two-dimensional representation comprised of basis vectors $v_1, v_2$ having transformations

$$\begin{aligned} a : & \quad v_1 \to \omega v_1 \, , & v_2 \to \omega^2 v_2 \, , \\ x : & \quad v_1 \to v_2 \, , & v_2 \to v_1 \, , \end{aligned} \tag{103}$$

where $\omega = e^{2\pi i/3}$. The fusion rules of symmetry generators are described by tensor products of representations. The one-dimensional representations form a group $\mathbb{Z}_2$ under tensor products, and the other tensor products are

$$
\begin{aligned}
E \otimes 1_- &= 1_- \otimes E = E, \\
E \otimes E &= 1 \oplus 1_- \oplus E.
\end{aligned}
\tag{104}
$$

The conjugacy classes for $S_3$ are

$$
1 = \{1\}, \qquad a = \{a, a^2\}, \qquad b = \{b, ab, a^2 b\},
\tag{105}
$$

where we have also labeled them. The centralizers are

$$
H_1 = S_3, \qquad H_b = \mathbb{Z}_2, \qquad H_a = \mathbb{Z}_3.
\tag{106}
$$

The Drinfeld center $\mathcal{Z}(\mathrm{Rep}(S_3))$ is thus comprised of anyons

$$
(1, R), \quad (b, s), \quad (a, \omega^p),
\tag{107}
$$

where $R$ is an irreducible representation of $S_3$, $s \in \{+, -\}$ captures irreducible representations of $\mathbb{Z}_2$, and $p \in \{0, 1, 2\}$ captures irreducible representations of $\mathbb{Z}_3$. The symmetry Lagrangian algebra for $\mathrm{Rep}(S_3)$ is

$$
\mathcal{L}^{\mathrm{sym}}_{\mathrm{Rep}(S_3)} = (1, 1) \oplus (a, 1) \oplus (b, +).
\tag{108}
$$

Consider the physical Lagrangian algebras

$$
\begin{aligned}
\mathcal{L}^{\mathrm{phys}}_1 &= (1, 1) \oplus (1, 1_-) \oplus 2(a, 1), \\
\mathcal{L}^{\mathrm{phys}}_2 &= (1, 1) \oplus (a, 1) \oplus (b, +),
\end{aligned}
\tag{109}
$$

which respectively describe $\mathrm{Rep}(S_3)/\mathbb{Z}_2$ SSB and $\mathrm{Rep}(S_3)$ SSB phases.

Note that both phases have three vacua each, which we label by $i \in \{0, 1, 2\}$. The $\mathrm{Rep}(S_3)$ symmetry is realized in the $\mathrm{Rep}(S_3)/\mathbb{Z}_2$ SSB phase as

$$
\begin{aligned}
1_- &= 1_{00} \oplus 1_{11} \oplus 1_{22}, \\
E &= 1_{01} \oplus 1_{02} \oplus 1_{12} \oplus 1_{10} \oplus 1_{20} \oplus 1_{21},
\end{aligned}
\tag{110}
$$

and on $\mathrm{Rep}(S_3)$ SSB phase as

$$
\begin{aligned}
1_- &= 1_{00} \oplus 1_{12} \oplus 1_{21}, \\
E &= 1_{00} \oplus 1_{01} \oplus 1_{02} \oplus 1_{10} \oplus 1_{20}.
\end{aligned}
\tag{111}
$$

Thus, even though both the SSB phases have three vacua, they are distinguished by the action of $\mathrm{Rep}(S_3)$ on them. Moreover, the underlying TQFTs for the two phases are also different; in particular the two phases carry different relative Euler terms. The $\mathrm{Rep}(S_3)/\mathbb{Z}_2$ SSB phase has trivial relative Euler terms between all three vacua. On the other hand, the $\mathrm{Rep}(S_3)$ SSB phase has trivial relative Euler terms between vacua 1 and 2, but non-trivial relative Euler terms between vacua 0 and $i \in \{1, 2\}$, which are encoded in the quantum dimensions of vacua changing line operators

$$
\dim(1_{0i}) = 2, \qquad \dim(1_{i0}) = 1/2,
\tag{112}
$$

for $i \in \{1, 2\}$. These quantum dimensions are essential for the quantum dimension of the symmetry generator $E$ being 2. See [2] for more details.

The order parameters for $\mathrm{Rep}(S_3)/\mathbb{Z}_2$ SSB are determined from $\mathcal{L}^{\mathrm{phys}}_1$ to be:

- A $1_-$-twisted sector local operator uncharged under $\mathrm{Rep}(S_3)$.

- Two linearly independent $(a, 1)$ multiplets of local operators. Each $(a, 1)$ multiplet contains an untwisted sector local operator and a $1_-$-twisted sector local operator which are exchanged by the action of $E \in \mathrm{Rep}(S_3)$. See [2] for more details.

Similarly, the order parameters for $\mathrm{Rep}(S_3)$ SSB are determined from $\mathcal{L}_2^{\mathrm{phys}}$ to be an $(a, 1)$ multiplet and a $(b, +)$ multiplet, for which more details can be found in [2]. The confined charges for the two phases are all the charges in $\mathcal{Z}(\mathrm{Rep}(S_3))$ which are not the charges of the respective order parameters.

## 4.3 Gapless SSB phases (gSSB)

A (1+1)d system in a gapless SSB (gSSB) phase for $\mathcal{S}$ symmetry has multiple gapless universes[16] $i \in \{0, 1, 2, \ldots, n-1\}$ in the IR. There are gapless excitations in each universe, which means we have $\Delta^{(i)} = 0$ for all $i$, but excitations transitioning between two different universes $i$ and $j$ cost a minimum non-zero energy $\Delta^{(ij)} > 0$.[17] The symmetry fusion category $\mathcal{S}$ mixes the universes into each other.

The IR effective theory describing each universe $i$ is some conformal field theory (CFT) $\mathfrak{T}_i^{\mathrm{IR}}$ whose only topological local operators are multiples of the identity operator. Thus the total IR effective theory describing all universes is a CFT $\mathfrak{T}^{\mathrm{IR}}$ which is a sum of all these CFTs

$$\mathfrak{T}^{\mathrm{IR}} = \bigoplus_i \mathfrak{T}_i^{\mathrm{IR}}. \tag{113}$$

The total CFT $\mathfrak{T}^{\mathrm{IR}}$ has an $n$-dimensional vector space of topological local operators.

Different gSSB phases are characterized by the charges for symmetry $\mathcal{S}$ that are not realized in the IR CFT $\mathfrak{T}^{\mathrm{IR}}$, i.e. by the confined charges. These charges again define a symmetry gap $\Delta_{\mathcal{S}} > 0$ which is the lowest energy of an excitation carrying one of these confined charges. A gSSB phase $\mathfrak{T}_1$ with confined charges $Q_1$ has no obstructions to being deformed to a gSSB phase $\mathfrak{T}_2$ with confined charges $Q_2$ without breaking $\mathcal{S}$ and without closing the symmetry gap $\Delta_{\mathcal{S}}$ if $Q_1 \subset Q_2$. On the other hand, if $Q_1 \not\subset Q_2$ then $\mathfrak{T}_1$ cannot be deformed to $\mathfrak{T}_2$ without either closing $\Delta_{\mathcal{S}}$ or breaking $\mathcal{S}$ at some point along the deformation. In such a deformation, the number of universes monotonically increases (and may remain preserved).

Note that the confined charges exhibited by gapped SSB phases can also be exhibited by gapless SSB phases. The only way such gapless SSBs differ from gapped SSBs is that the former have $\Delta^{(i)} = 0$ while the latter have $\Delta^{(i)} > 0$; however, both have $\Delta_{\mathcal{S}} > 0$. There is no obstruction to deforming such gapless SSBs to gapped SSBs, which can in principle be done by simply opening up the energy gaps $\Delta^{(i)}$.

We define an **intrinsically gapless SSB (igSSB) phase** to be a gSSB phase with $n$ universes that cannot be deformed to a gapped SSB phase with $n$ vacua, but only to gapped SSB phases with more than $n$ vacua. An igSSB phase exhibits symmetry protected criticality in the sense that criticality is preserved as long as we do not explicitly break $\mathcal{S}$ or do not further

---

[16]For gapless systems, we use the term "universes" instead of "vacua" as the gapless IR theory in each universe may have various kinds of vacua and even moduli spaces of vacua. The information of the precise vacuum for each universe is not needed for the analysis.

[17]We emphasize that in (1+1)d, a domain wall corresponds to a kink excitation that is point-like in space but has a one-dimensional worldline in spacetime. It interpolates between two distinct vacua $i \neq j$ and carries a tension determined by the difference of Euler terms for those vacua. Because those Euler terms remain nonzero in the phases considered, the domain-wall excitation has finite energy and thus remains gapped ($\Delta^{(ij)} > 0$). In contrast, the gaplessness of an igSSB phase stems from the dynamics within each vacuum universe ($\Delta^{(i)} = 0$), reflecting uncondensed symmetry charges rather than kink excitations. A gapless domain wall would require tuning the Euler terms to zero, which collapses the distinct vacua into a single phase and moves the system to a different (multicritical) point outside the gSSB/igSSB regime discussed here.

spontaneously break $\mathcal{S}$ (thus increasing the number of universes).[18]

Given two different universes $i$ and $j$, we must have at least one topological interface between them, taking us from $\mathfrak{T}_i^{\text{IR}}$ to $\mathfrak{T}_j^{\text{IR}}$. This is because there must be some symmetry in $\mathcal{S}$ that sends universe $i$ to universe $j$, which has to be realised in the IR by such a topological interface. This means that the CFT $\mathfrak{T}_j^{\text{IR}}$ can be obtained from the CFT $\mathfrak{T}_i^{\text{IR}}$ by gauging (possibly non-invertible) topological defects of $\mathfrak{T}_i^{\text{IR}}$.

The mathematical structure of the full IR CFT $\mathfrak{T}^{\text{IR}}$ is thus as follows. There is a fusion category $\mathcal{S}'$ of topological defects in $\mathfrak{T}_0^{\text{IR}}$ which are responsible for realising the part of the symmetry $\mathcal{S}$ that keeps the universe 0 invariant. Other IR CFTs $\mathfrak{T}_i^{\text{IR}}$ are obtained from $\mathfrak{T}_0^{\text{IR}}$ by gauging $\mathcal{S}'$. Different gaugings of $\mathcal{S}'$ are parametrised by indecomposable module categories of $\mathcal{S}'$. Thus $\mathfrak{T}_i^{\text{IR}}$ is obtained by performing a gauging of $\mathfrak{T}_0^{\text{IR}}$ corresponding to some module category $\mathcal{M}_i$, which we express as

$$\mathfrak{T}_i^{\text{IR}} = \mathfrak{T}_0^{\text{IR}}/\mathcal{M}_i. \tag{114}$$

The module category $\mathcal{M}_i$ describes a set of topological interfaces from $\mathfrak{T}_0^{\text{IR}}$ to $\mathfrak{T}_i^{\text{IR}}$ on which $\mathcal{S}'$ acts from the left. After gauging we obtain a dual symmetry of $\mathfrak{T}_i^{\text{IR}}$ that we label as $\mathcal{S}_i'$, which are topological defects of $\mathfrak{T}_i^{\text{IR}}$ acting from the right on the interfaces described by $\mathcal{M}_i$ such that $\mathcal{M}_i$ is a bimodule category of $\mathcal{S}'$ and $\mathcal{S}_i'$. Mathematically, $\mathcal{S}_i'$ is computed as the category of endofunctors of $\mathcal{M}_i$ compatible with the action of $\mathcal{S}'$. It is these topological defects $\mathcal{S}_i'$ that realise the part of the symmetry $\mathcal{S}$ leaving the universe $i$ invariant.

We can in fact express any $\mathfrak{T}_j^{\text{IR}}$ as a gauging of $\mathfrak{T}_i^{\text{IR}}$ by some module category $\mathcal{M}_{ij}$ of $\mathcal{S}_i'$

$$\mathfrak{T}_j^{\text{IR}} = \mathfrak{T}_i^{\text{IR}}/\mathcal{M}_{ij}. \tag{115}$$

This module category $\mathcal{M}_{ij}$ is fixed by the condition that

$$\mathcal{M}_j = \mathcal{M}_i \boxtimes_{\mathcal{S}_i'} \mathcal{M}_{ij}, \tag{116}$$

where $\boxtimes_{\mathcal{S}'}$ is the relative Deligne product. The fusion categories $\mathcal{S}_i'$ and (bi)module categories $\mathcal{M}_{ij}$ combine to form a multi-fusion category that we label as $\mathcal{S}_{\text{multi}}'$. The symmetry $\mathcal{S}$ is realised as a pivotal tensor functor

$$\phi: \mathcal{S} \to \mathcal{S}_{\text{multi}}'. \tag{117}$$

Thus the problem of **classification of (1+1)d gSSB** phases is essentially the problem of **classification of functors from fusion categories to multi-fusion categories**.

A gSSB phase can thus be labeled by a tuple of the form

$$(\mathcal{S}', \mathcal{M}_1, \mathcal{M}_2, \dots, \mathcal{M}_{n-1}; \phi), \tag{118}$$

where $(\mathcal{S}', \mathcal{M}_1, \mathcal{M}_2, \dots, \mathcal{M}_{n-1})$ specifies the pivotal multi-fusion category $\mathcal{S}_{\text{multi}}'$. For a gapped SSB phase, we have

$$\mathcal{S}' = \mathcal{M}_i = \mathsf{Vec}, \tag{119}$$

for all $i$, however the pivotal structure can be non-trivial which accounts for the presence of relative Euler terms in the IR.

In terms of the SymTFT, an $\mathcal{S}$-symmetric gapless SSB phase is specified by a condensable algebra $\mathcal{A}_\phi$ that does not satisfy the condition (52), instead satisfying

$$\mathcal{A}_\phi \cap \mathcal{L}_{\mathcal{S}}^{\text{sym}} \supsetneq 1, \tag{120}$$

---

[18]In this work, we use the term "critical" to refer exclusively to intrinsically gapless phases such as igSPT and igSSB. These are phases where the gaplessness is protected by the symmetry category $\mathcal{S}$, and cannot be lifted without explicitly or spontaneously breaking the symmetry.

i.e. the intersection of $\mathcal{A}_\phi$ with the symmetry Lagrangian $\mathcal{L}_{\mathcal{S}}^{\text{sym}}$ contains non-identity anyons (along with the identity anyon). The IR theory $\mathfrak{T}^{\text{IR}}$ of a gapless system lying in the gSSB phase $(\mathcal{S}', \phi)$ is obtained by inserting a suitable physical boundary $\mathfrak{B}_{\mathfrak{T}^{\text{IR}}}^{\text{phys}}$ to the club quiche defined by $\mathcal{A}_\phi$ as in (39).

The information on $\mathcal{S}'$ is encoded in the reduced topological order $\mathfrak{Z}(\mathcal{S}')$ arising on the right of the topological interface $\mathcal{I}_\phi$ defined by $\mathcal{A}_\phi$. As before, $\mathfrak{Z}(\mathcal{S}')$ can be determined by seeking a Lagrangian algebra completion $\mathcal{L}_\phi \in \mathcal{Z}(\mathcal{S}) \boxtimes \overline{\mathcal{Z}(\mathcal{S}')}$ of the condensable algebra $\mathcal{A}_\phi$ which takes a similar form as in (47). The topological order $\mathfrak{Z}(\mathcal{S}')$ only determines $\mathcal{S}'$ up to gauging (or in other words Morita equivalence). To completely determine $\mathcal{S}'$, we need to compactify the interval occupied by $\mathfrak{Z}(\mathcal{S})$ in the club quiche. The procedure for doing this is discussed in detail in [10], to which we refer the reader. This results in a reducible topological boundary condition $\mathfrak{B}_{\mathcal{S}'_{\text{multi}}}$ of $\mathfrak{Z}(\mathcal{S}')$ hosting the multi-fusion category $\mathcal{S}'_{\text{multi}}$ from which the information on $\mathcal{S}'$ and $\mathcal{M}_i$ can be deduced. The number of irreducible topological boundary conditions involved in $\mathfrak{B}_{\mathcal{S}'_{\text{multi}}}$ is the same as the number of universes $n$ participating in the gSSB phase and can be determined simply as

$$n = \sum_a n_{a^*}^\phi n_a^{\text{sym}}, \tag{121}$$

where $n_a^{\text{sym}}$ are the number of linearly independent topological ends of a simple anyon $\mathbf{Q}_a$ in $\mathcal{Z}(\mathcal{S})$ along the symmetry boundary $\mathfrak{B}_{\mathcal{S}}^{\text{sym}}$, as reflected by the coefficient of $\mathbf{Q}_a$ in the symmetry Lagrangian algebra $\mathcal{L}_{\mathcal{S}}^{\text{sym}}$, $n_a^\phi$ are the number of linearly independent topological ends of a simple anyon $\mathbf{Q}_a$ in $\mathcal{Z}(\mathcal{S})$ along the interface $\mathcal{I}_\phi$, as reflected by the coefficient of $\mathbf{Q}_a$ in

$$\mathcal{A}_\phi = \bigoplus_a n_a^\phi \mathbf{Q}_a, \tag{122}$$

and $\mathbf{Q}_{a^*}$ denotes the dual of $\mathbf{Q}_a$. The functor $\phi$ is also deduced by compactifying the club quiche, as described in detail in [10].

The confined charges for a gSSB phase are encoded in $\mathcal{A}_\phi$ in exactly the same way as for a gSPT phase. That is, we have a functor on the centers

$$\mathcal{Z}(\phi): \ \mathcal{Z}(\mathcal{S}') \to \mathcal{Z}(\mathcal{S}), \tag{123}$$

which is determined in terms of $\mathcal{A}_\phi$ as in (48). The confined charges are the charges in $\mathcal{Z}(\mathcal{S})$ not in the image of $\mathcal{Z}(\phi)$. Just like for gSPT phases, possible deformations of gSSB phases are captured by inclusion of associated condensable algebras as discussed around (54).

## 4.4 Example: $\text{Rep}(S_3)$ gSSB

For $\mathcal{S} = \text{Rep}(S_3)$, the condensable algebras are

$$\begin{aligned}
\mathcal{A}_1 &= (1,1), \\
\mathcal{A}_a &= (1,1) \oplus (a,1), \\
\mathcal{A}_- &= (1,1) \oplus (1,1_-), \\
\mathcal{A}_E &= (1,1) \oplus (1,E), \\
\mathcal{A}_{a,b} &= (1,1) \oplus (a,1) \oplus (b,+), \\
\mathcal{A}_{-,a} &= (1,1) \oplus (1,1_-) \oplus 2(a,1), \\
\mathcal{A}_{-,E} &= (1,1) \oplus (1,1_-) \oplus 2(1,E), \\
\mathcal{A}_{E,b} &= (1,1) \oplus (1,E) \oplus (b,+),
\end{aligned} \tag{124}$$

with $\mathcal{L}^{\text{sym}}_{\text{Rep}(S_3)} = \mathcal{A}_{a,b}$. Using (120), we find that gSSBs are given by

$$\text{gSSBs for Rep}(S_3) \text{ Symmetry} = \{\mathcal{A}_a, \mathcal{A}_{a,b}, \mathcal{A}_{-,a}, \mathcal{A}_{E,b}\}, \tag{125}$$

and gapped SSBs are

$$\text{SSBs for Rep}(S_3) \text{ Symmetry} = \{\mathcal{A}_{a,b}, \mathcal{A}_{-,a}, \mathcal{A}_{E,b}\}, \tag{126}$$

as these algebras are also Lagrangian. These gapped SSBs were discussed in the previous section, and the structure of the corresponding gSSBs is exactly the same, with each universe realizing a CFT instead of a TQFT in the IR.

This leaves behind the sole gSSB phase $\mathcal{A}_a$ exhibiting confined charges not exhibited by any of the gapped SSB phases. The structure of this gSSB phase was worked out in section V.J of [10]. We have two universes $i \in \{0, 1\}$. The relevant IR symmetry in both universes is a non-anomalous $\mathbb{Z}_2$

$$\mathcal{S}' = \mathcal{S}'_0 = \text{Vec}_{\mathbb{Z}_2}, \qquad \mathcal{S}'_1 = \text{Vec}_{\mathbb{Z}_2}. \tag{127}$$

However, the two $\mathbb{Z}_2$ symmetries are duals of each other, i.e. one $\mathbb{Z}_2$ symmetry is obtained as the quantum/dual symmetry after gauging the other $\mathbb{Z}_2$ symmetry. This is reflected in the module categories being

$$\begin{aligned} \mathcal{M}_1 = \mathcal{M}_{01} = \text{Vec}, \\ \mathcal{M}_{10} = \text{Vec}. \end{aligned} \tag{128}$$

If the two $\mathbb{Z}_2$ symmetries were not related by gauging, then the module categories would have been $\mathcal{M}_{ij} = \text{Vec}_{\mathbb{Z}_2}$. The two IR theories in the two universes are thus related as

$$\begin{aligned} \mathfrak{T}^{\text{IR}}_0 = \mathfrak{T}^{\text{IR}}_1 / \mathbb{Z}_2, \\ \mathfrak{T}^{\text{IR}}_1 = \mathfrak{T}^{\text{IR}}_0 / \mathbb{Z}_2. \end{aligned} \tag{129}$$

The resulting multi-fusion category was denoted in [10] as

$$\mathcal{S}'_{\text{multi}} = \text{Ising}^{\sqrt{2}}_{2 \times 2}, \tag{130}$$

which contains lines

$$\{1_{00}, P_{00}, 1_{11}, P_{11}, S_{01}, S_{10}\}, \tag{131}$$

where $\{1_{ii}, P_{ii}\}$ form the sub-fusion category $\mathcal{S}'_i = \text{Vec}_{\mathbb{Z}_2}$, and $S_{ij}$ forms the sub-category $\mathcal{M}_{ij}$. The fusion rules are

$$\begin{aligned} S_{ij} \otimes X_{jj} = X_{ii} \otimes S_{ij} = S_{ij}, \\ S_{ij} \otimes S_{ji} = 1_{ii} \oplus P_{ii}. \end{aligned} \tag{132}$$

The $\sqrt{2}$ in the superscript on the RHS of (130) denotes that there is a non-trivial pivotal structure on this multi-fusion category which reflects in quantum dimension of $S_{ij}$ being

$$\dim(S_{01}) = 2, \qquad \dim(S_{10}) = 1, \tag{133}$$

which differ by a factor of $\sqrt{2}$ from their naive quantum dimensions $\dim = \sqrt{2}$. From the point of view of SymTFT, the reducible topological boundary condition of $\mathfrak{Z}(\text{Vec}_{\mathbb{Z}_2})$ is $\mathfrak{B}_e \oplus \mathfrak{B}_m$, where $\mathfrak{B}_e$ is a topological boundary condition on which $e$ is condensed, and $\mathfrak{B}_m$ is a topological boundary condition on which $m$ is condensed.

The functor $\phi$ is such that the $\text{Rep}(S_3)$ lines are identified as

$$\begin{aligned} 1_- = 1_{00} \oplus P_{11}, \\ E = S_{01} \oplus S_{10} \oplus P_{00}. \end{aligned} \tag{134}$$

The Lagrangian algebra completion of $\mathcal{A}_a$ in the folded setup is

$$\mathcal{L}_a = (1,1) \oplus (a,1) \oplus (1,1_-)\overline{e} \oplus (a,1)\overline{e} \oplus (b,+)\overline{m} \oplus (b,-)\overline{em}, \tag{135}$$

which implies that the map of generalized charges is

$$\begin{aligned}
\mathcal{Z}(\phi): \ \mathcal{Z}(\mathsf{Vec}_{\mathbb{Z}_2}) &\to \mathcal{Z}(\mathsf{Rep}(S_3)), \\
1 &\mapsto (1,1) \oplus (a,1), \\
e &\mapsto (1,1_-) \oplus (a,1), \\
m &\mapsto (b,+), \\
em &\mapsto (b,-),
\end{aligned} \tag{136}$$

and thus the confined charges associated to the gSSB phase are

$$Q_\phi = \{(1,E),(a,\omega),(a,\omega^2)\}. \tag{137}$$

Let us note that this gSSB phase $\mathcal{A}_a$ is actually an intrinsic gSSB (igSSB) phase. Its only deformations are $\mathcal{A}_{-,a}$ and $\mathcal{A}_{a,b}$, both of which are associated to gapped SSB phases with 3 vacua. Thus, any gapped deformation of the $\mathcal{A}_a$ gSSB phase necessarily increases the number of universes, further spontaneously breaking the $\mathsf{Rep}(S_3)$ symmetry.

The order parameter for this igSSB phase is an $(a,1)$ multiplet of local operators. The confined charges are those that do not appear in the image of the functor $\mathcal{Z}(\phi)$ described above, which are $(1,E)$, $(a,\omega)$ and $(a,\omega^2)$. As we deform this igSSB phase to $\mathsf{Rep}(S_3)/\mathbb{Z}_2$ SSB phase, we add $(b,+)$ and $(b,-)$ to the list of confined charges. On the other hand, as we deform it to $\mathsf{Rep}(S_3)$ SSB phase, we add $(1,1_-)$ and $(b,-)$ to the list of confined charges.

## 5 Hasse diagram of phases

In this work, we have discussed at length (1+1)d gapped and gapless SPT and SSB phases. We have also discussed possible deformation patterns of these phases. All of this information can be collected together in a Hasse diagram – i.e. a graph depicting a partially ordered set – as we will describe below.

### 5.1 Partial order on algebras and phases

We first describe a mathematical Hasse diagram of condensable algebras in a (2+1)d topological order $\mathfrak{Z}$ that admits a topological boundary condition. This Hasse diagram becomes a **Hasse diagram for physical phases** after choosing a specific symmetry fusion category $\mathcal{S}$ such that $\mathfrak{Z}$ is the SymTFT for $\mathcal{S}$, i.e.

$$\mathfrak{Z}(\mathcal{S}) = \mathfrak{Z}. \tag{138}$$

The Hasse diagram is constructed as follows:

- The nodes of the Hasse diagram are condensable algebras (not necessarily Lagrangian) in the MTC $\mathcal{Z}$ formed by anyons of $\mathfrak{Z}$. We arrange these algebras in layers according to their quantum dimensions, with the lowest quantum dimension on the top-most layer and the highest quantum dimension (i.e. Lagrangian algebras) at the bottom-most layer. The top-most layer is always occupied by a single node corresponding to a trivial algebra $\mathcal{A} = 1$.

- Given two condensable algebras $\mathcal{A}_1$ and $\mathcal{A}_2$ with quantum dimensions

$$\dim(\mathcal{A}_1) < \dim(\mathcal{A}_2),\qquad(139)$$

  we draw an edge connecting $\mathcal{A}_1$ and $\mathcal{A}_2$ if $\mathcal{A}_1$ is a subalgebra of $\mathcal{A}_2$. This establishes a partial ordering on the condensable algebras. Paths in the Hasse diagram are monotonous, from top to bottom, thus respecting the partial order.

The Hasse diagram obtained in this way is a priori blind to specific symmetry and depends only on the SymTFT. See the top half of figure 1 for an example where we have taken $\mathfrak{Z}$ to be the $\mathbb{Z}_4$ Dijkgraaf-Witten gauge theory (without twist).

**Hasse diagram for $\mathcal{S}$-symmetric phases.**  In order to convert this into a Hasse diagram for phases we choose a symmetry $\mathcal{S}$ whose SymTFT is $\mathfrak{Z}$. Concretely, this means that we choose a Lagrangian algebra $\mathcal{L}_{\mathcal{S}}^{\mathrm{sym}}$ in $\mathcal{Z}$ corresponding to a topological boundary condition hosting symmetry $\mathcal{S}$ along it. There may be multiple Lagrangian algebras leading to $\mathcal{S}$ symmetry, which would give equivalent but different translations of the Hasse diagram into a phase diagram. After choosing $\mathcal{L}_{\mathcal{S}}^{\mathrm{sym}}$, each condensable algebra in $\mathcal{Z}$ defines a gapped or gapless SPT or SSB phase with $\mathcal{S}$ symmetry. The nodes of the Hasse diagram are now identified with these phases. The edges describe possible deformations between the phases. That is, if $\mathcal{A}_1$ is connected to $\mathcal{A}_2$ with $\dim(\mathcal{A}_1) < \dim(\mathcal{A}_2)$, then there are no obstructions to deforming the phase corresponding to $\mathcal{A}_1$ to the phase corresponding to $\mathcal{A}_2$ on symmetry grounds (though the deformation may not be possible dynamically) without closing the symmetry gap $\Delta_{\mathcal{S}}$ and without breaking the symmetry $\mathcal{S}$ along the deformation. See the bottom half of figure 1 for an example where we have taken $\mathcal{S}$ to be a non-anomalous $\mathbb{Z}_4$ symmetry and $\mathcal{L}_{\mathcal{S}}^{\mathrm{sym}}$ to be the collection of all electric anyons.

In this phase diagram of $\mathcal{S}$-symmetric phases, the top-most layer is the canonical gSPT phase, and the bottom-most layer contains all the gapped SPT and SSB phases. In between these layers, we have other layers containing gSPT and gSSB phases, which may or may not be intrinsic.

To characterize these phases in more detail, we need to assign a positive integer $n$ to each node, which is computed using the formula (121). $n$ counts the number of interval compactifications of anyons between the symmetry boundary $\mathfrak{B}_{\mathcal{S}}^{\mathrm{sym}}$ and the interface $\mathcal{I}_\phi$ defined by the condensable algebra, by using various possible ends of anyons along $\mathfrak{B}_{\mathcal{S}}^{\mathrm{sym}}$ and $\mathcal{I}_\phi$. Physically this is the number of universes in each phase. Note that $n$ monotonically increases as one goes to lower levels in the Hasse diagram. We can now characterize the different phases as:

1. SPT (gapped): A condensable algebra in the lowest level (i.e. Lagrangian) for which we have $n = 1$.

2. gSPT (gapless): A condensable algebra not necessarily at the lowest level for which we have $n = 1$. In the phase diagrams we will only label the phases with $n = 1$ above the lowest level as gSPT phases and the phases with $n = 1$ at lowest level as SPT phases to avoid multiple labelings of the same node, but it should be kept in mind that a Lagrangian algebra with $n = 1$ can describe both gapped and gapless SPT phases.

3. igSPT: A condensable algebra not at the lowest level for which we have $n = 1$ and which cannot be joined by an oriented path in the Hasse diagram to a condensable algebra in the lowest level with $n = 1$.

4. SSB (gapped): A condensable algebra in the lowest level for which we have $n > 1$.

5. gSSB (gapless): A condensable algebra not necessarily at the lowest level for which we have $n > 1$. In the phase diagrams we will only label the phases with $n > 1$ at non-lowest levels as gSSB phases and the phases with $n > 1$ at lowest level as SSB phases to avoid multiple labelings of the same node, but it should be kept in mind that a Lagrangian algebra with $n > 1$ can describe both gapped and gapless SSB phases.

6. igSSB: A condensable algebra not at the lowest level for which we have $n > 1$ and which cannot be joined by an oriented path of edges to a condensable algebra in the lowest level with the same value of $n$.

Note that if we choose a different Lagrangian algebra $\mathcal{L}_{\tilde{\mathcal{S}}}^{\text{sym}}$ leading to a choice of a symmetry $\tilde{\mathcal{S}} \neq \mathcal{S}$ with

$$\mathfrak{Z}(\tilde{\mathcal{S}}) = \mathfrak{Z}(\mathcal{S}) = \mathfrak{Z}, \tag{140}$$

then the phase diagram obtained from the same Hasse diagram of condensable algebras is different, with different values of $n$ and different distributions of SPT, gSPT, igSPT, SSB, gSSB and igSSB phases. An example is discussed in figure 2 where we choose two different symmetries $S_3$ and $\text{Rep}(S_3)$ that have the same SymTFT.

**Phase transitions.** Consider a node $X$ in the Hasse diagram not in the bottom-most layer that is directly connected to two nodes $Y_1$ and $Y_2$ in the bottom-most layer. By a direct connection, we mean that there is an oriented path from $X$ to $Y_1$ or $Y_2$ that does not pass through any other node. In particular the gapless phase $\mathcal{P}_X$ corresponding to the node $X$ has no obstruction to being deformed to gapped phases $\mathcal{P}_{Y_1}$ and $\mathcal{P}_{Y_2}$ corresponding to the nodes $Y_1$ and $Y_2$. If we have a gapless theory $\mathfrak{T}_X$ in gapless phase $\mathcal{P}_X$ which dynamically admits relevant deformations to gapped theories in phases $\mathcal{P}_{Y_1}$ and $\mathcal{P}_{Y_2}$, then we say that $\mathfrak{T}_X$ is a phase transition between phases $\mathcal{P}_{Y_1}$ and $\mathcal{P}_{Y_2}$. In this way, the Hasse diagram encodes various phase transitions. Similarly, it also encodes phase transitions between two gapless phases, or phase transitions between more than two gapped/gapless phases (parametrized by other types of nodes in non-bottom layers). This strategy was used in [10] to realize phase transitions between gapped phases with non-invertible symmetries in (1+1)d. In the language of this paper, the gapless theories exhibiting these phase transitions lie in gSPT and gSSB phases.

The key tool to characterize the gapless phases describing the transitions is the club sandwich [10]

$$\begin{array}{ccc} \mathfrak{B}_{\mathcal{S}}^{\text{sym}} & \mathcal{I}_\phi & \mathfrak{B}_{\mathfrak{T}^{\text{IR}}}^{\text{phys}} \\ & & \\ \mathfrak{Z}(\mathcal{S}) & \mathfrak{Z}(\mathcal{S}') & \end{array} \tag{141}$$

where $\mathcal{I}_\phi$ is an interface between $\mathfrak{Z}(\mathcal{S})$ and the reduced topological order $\mathfrak{Z}(\mathcal{S}')$, and defines a KT transformation

$$\mathcal{K}_{\mathcal{I}_\phi}^{\mathcal{S},\mathcal{S}'} : \{\mathcal{S}'\text{-symmetric QFTs}\} \rightarrow \{\mathcal{S}\text{-symmetric QFTs}\}. \tag{142}$$

This enables one to start with a phase transition for $\mathcal{S}'$ symmetry, characterized by an $\mathcal{S}'$-symmetric CFT $\mathfrak{T}_{CFT}^{\mathcal{S}'}$ between the gapped $\mathcal{S}'$-symmetric phases $\mathfrak{T}_A^{\mathcal{S}'}$ and $\mathfrak{T}_B^{\mathcal{S}'}$ (which can be obtained from $\mathfrak{T}_{CFT}^{\mathcal{S}'}$ by a deformation with the operator $\mathcal{O}'$ uncharged under $\mathcal{S}'$):

$$\mathfrak{T}_B^{\mathcal{S}'} \xleftarrow{\ -\mathcal{O}'\ } \mathfrak{T}_{CFT}^{\mathcal{S}'} \xrightarrow{\ +\mathcal{O}'\ } \mathfrak{T}_A^{\mathcal{S}'}, \tag{143}$$

and then construct $\mathcal{S}$-symmetric phase transitions using the KT transformation, resulting in an $\mathcal{S}$-symmetric CFT $\mathfrak{T}^{\mathcal{S}}_{CFT}$ acting as a phase transition between two $\mathcal{S}$-symmetric gapped phases $\mathfrak{T}^{\mathcal{S}}_A$ and $\mathfrak{T}^{\mathcal{S}}_B$:

$$\mathfrak{T}^{\mathcal{S}}_B \xleftarrow{\quad -\mathcal{O} \quad} \mathfrak{T}^{\mathcal{S}}_{CFT} \xrightarrow{\quad +\mathcal{O} \quad} \mathfrak{T}^{\mathcal{S}}_A \ . \tag{144}$$

The operator $\mathcal{O}$ is uncharged under $\mathcal{S}$ and is obtained from $\mathcal{O}'$ by means of a KT transformation and the club sandwich compactification, as explained in [10].

## 5.2 Example: $\mathrm{Rep}(D_8)$ gapped and gapless phases

We now turn to the Hasse diagram for $\mathrm{Rep}(D_8)$. The condensable algebras are listed in table 3 and the Hasse diagram is shown in figure 3. A detailed discussion of all the condensable algebras is given in appendix B, and the corresponding phases are described in appendix C. We note that there is an igSPT for $\mathrm{Rep}(D_8)$, which we discussed in section 3.2 and three Ising igSSB phases, described in Appendix C.2.6.

Again we can reinterpret the Hasse diagram for any symmetry $\mathcal{S}$ which has the same center $\mathcal{Z}(\mathrm{Rep}(D_8))$, by specifying the associated Lagrangian to be the symmetry boundary: there are 11 distinct such symmetries.

As can be seen from the Hasse diagram in figure 3, condensable algebras in $\mathcal{Z}(\mathrm{Rep}(D_8))$ with $\mathcal{S}' = \mathbb{Z}_2, \mathbb{Z}_4, \mathbb{Z}_2 \times \mathbb{Z}_2$ correspond to gapless phases describing transitions respectively between two, three or five gapped phases. We derive this in detail in appendix D. The results are summarized in tables 4,5,6.

# 6 Gauging trivially acting symmetries and decomposition

At the end of the previous section, we saw that gauging a (gapped or gapless) SPT phase can lead to a (gapped or gapless) SSB phase. In the IR of a system in a SPT phase, the full symmetry $\mathcal{S}$ is only acting as a smaller symmetry $\mathcal{S}'$ via a map

$$\phi : \ \mathcal{S} \to \mathcal{S}', \tag{145}$$

and the symmetries lying in the kernel of $\phi$ act trivially. On the other hand, the IR of a system in a SSB phase is decomposed into multiple universes, with the symmetry $\mathcal{S}$ mixing the universes into each other. This is a special example of a general phenomenon known as **decomposition** [47–54], which occurs when trivially acting symmetries are gauged.

Thus, in the process of studying these phases, what we have discovered is a general formalism for understanding decomposition in the presence of non-invertible symmetries using the SymTFT. Let us describe this formalism in more detail. Consider a 2d QFT $\mathfrak{T}$ and a symmetry $\mathcal{S}$ that does not act faithfully on $\mathfrak{T}$. Let $\mathcal{S}$ act on $\mathfrak{T}$ via a map $\phi$ as displayed above. The symmetry $\mathcal{S}'$ involved in this map is a faithfully acting symmetry of $\mathfrak{T}$, which means that $\mathfrak{T}$ admits (possibly non-topological) local operators transforming in all possible generalized charges (which are valued in the Drinfeld center $\mathcal{Z}(\mathcal{S}')$) of $\mathcal{S}'$. The charges of $\mathcal{S}$ realized by the operators of $\mathfrak{T}$ are the ones lying in the image of the map

$$\mathcal{Z}(\phi) : \ \mathcal{Z}(\mathcal{S}') \to \mathcal{Z}(\mathcal{S}), \tag{146}$$

discussed in section 3.

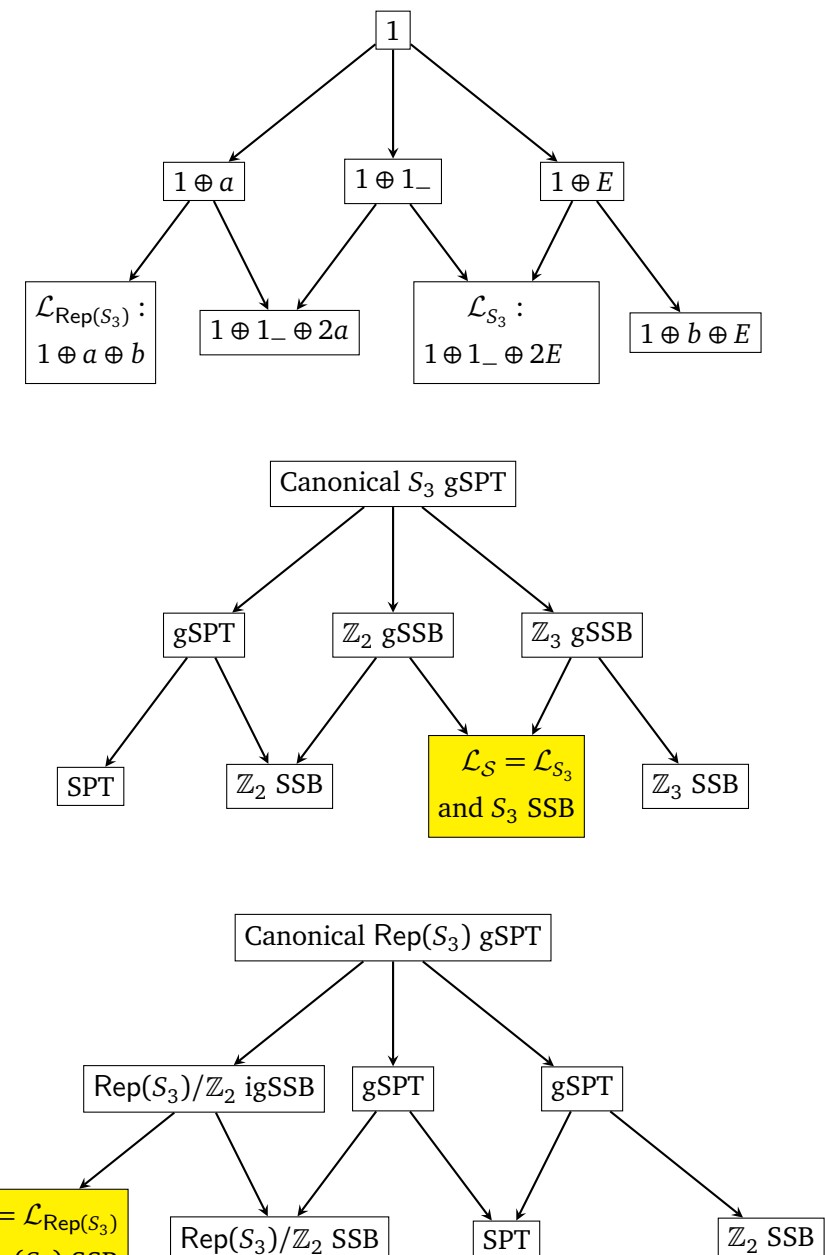

Figure 2: Hasse diagram for $\mathcal{Z}(\mathsf{Vec}_{S_3})$: The top figure shows the Hasse diagram of the condensable algebras. The lowest level are the maximal, i.e. Lagrangian, algebras. Picking one of these as the symmetry Lagrangian algebra that fixes the symmetry $\mathcal{S}$ allows classification of all phases: this is done for $\mathsf{Vec}_{S_3}$ in the middle figure and $\mathsf{Rep}(S_3)$ in the bottom figure. There are no igSPTs for these symmetries.

Table 3: Condensable Algebras of $\mathcal{Z}(\mathrm{Rep}(D_8))$ with their quantum dimension, label and $\mathcal{S}'$. The associated Reduced Topological Order is $\mathfrak{Z}(\mathcal{S}')$. The last two columns describe the phases once the symmetry $\mathcal{S} = \mathrm{Rep}(D_8)$ is fixed and the number of universes $n$, defined in equation (121).

| Dim | Label | Condensable Algebra of $\mathcal{Z}(\mathrm{Rep}(D_8))$ | $\mathcal{S}'$ | Phase for $\mathcal{S} = \mathrm{Rep}(D_8)$ | $n$ |
|---|---|---|---|---|---|
| 1 | $\mathcal{A}_0$ | $1$ | $\mathcal{S}$ | Canonical $\mathrm{Rep}(D_8)$ gSPT | 1 |
| 2 | $\mathcal{A}_1$ | $1 \oplus e_{RG}$ | $\mathbb{Z}_4$ | gSPT | 1 |
| 2 | $\mathcal{A}_2$ | $1 \oplus e_{GB}$ | $\mathbb{Z}_4$ | gSPT | 1 |
| 2 | $\mathcal{A}_3$ | $1 \oplus e_{RB}$ | $\mathbb{Z}_4$ | gSPT | 1 |
| 2 | $\mathcal{A}_4$ | $1 \oplus e_R$ | $\mathbb{Z}_2 \times \mathbb{Z}_2$ | gSPT | 1 |
| 2 | $\mathcal{A}_5$ | $1 \oplus e_G$ | $\mathbb{Z}_2 \times \mathbb{Z}_2$ | gSPT | 1 |
| 2 | $\mathcal{A}_6$ | $1 \oplus e_B$ | $\mathbb{Z}_2 \times \mathbb{Z}_2$ | gSPT | 1 |
| 2 | $\mathcal{A}_7$ | $1 \oplus e_{RGB}$ | $\mathbb{Z}_2 \times \mathbb{Z}_2$ | $\mathrm{Rep}(D_8)/(\mathbb{Z}_2 \times \mathbb{Z}_2)$ gSSB | 2 |
| 4 | $\mathcal{A}_8$ | $1 \oplus e_{GB} \oplus e_{RB} \oplus e_{RG}$ | $\mathbb{Z}_2^\omega$ | igSPT | 1 |
| 4 | $\mathcal{A}_9$ | $1 \oplus e_R \oplus m_{GB}$ | $\mathbb{Z}_2$ | $\mathbb{Z}_2$ gSSB | 2 |
| 4 | $\mathcal{A}_{10}$ | $1 \oplus e_R \oplus m_G$ | $\mathbb{Z}_2$ | gSPT | 1 |
| 4 | $\mathcal{A}_{11}$ | $1 \oplus e_R \oplus m_B$ | $\mathbb{Z}_2$ | gSPT | 1 |
| 4 | $\mathcal{A}_{12}$ | $1 \oplus e_G \oplus m_{RB}$ | $\mathbb{Z}_2$ | $\mathbb{Z}_2$ gSSB | 2 |
| 4 | $\mathcal{A}_{13}$ | $1 \oplus e_G \oplus m_R$ | $\mathbb{Z}_2$ | gSPT | 1 |
| 4 | $\mathcal{A}_{14}$ | $1 \oplus e_G \oplus m_B$ | $\mathbb{Z}_2$ | gSPT | 1 |
| 4 | $\mathcal{A}_{15}$ | $1 \oplus e_B \oplus m_{RG}$ | $\mathbb{Z}_2$ | $\mathbb{Z}_2$ gSSB | 2 |
| 4 | $\mathcal{A}_{16}$ | $1 \oplus e_B \oplus m_R$ | $\mathbb{Z}_2$ | gSPT | 1 |
| 4 | $\mathcal{A}_{17}$ | $1 \oplus e_B \oplus m_G$ | $\mathbb{Z}_2$ | gSPT | 1 |
| 4 | $\mathcal{A}_{18}$ | $1 \oplus e_{RGB} \oplus m_{RG}$ | $\mathbb{Z}_2$ | Ising igSSB | 3 |
| 4 | $\mathcal{A}_{19}$ | $1 \oplus e_{RGB} \oplus m_{GB}$ | $\mathbb{Z}_2$ | Ising igSSB | 3 |
| 4 | $\mathcal{A}_{20}$ | $1 \oplus e_{RGB} \oplus m_{RB}$ | $\mathbb{Z}_2$ | Ising igSSB | 3 |
| 4 | $\mathcal{A}_{21}$ | $1 \oplus e_G \oplus e_R \oplus e_{RG}$ | $\mathbb{Z}_2$ | gSPT | 1 |
| 4 | $\mathcal{A}_{22}$ | $1 \oplus e_B \oplus e_G \oplus e_{GB}$ | $\mathbb{Z}_2$ | gSPT | 1 |
| 4 | $\mathcal{A}_{23}$ | $1 \oplus e_B \oplus e_R \oplus e_{RB}$ | $\mathbb{Z}_2$ | gSPT | 1 |
| 4 | $\mathcal{A}_{24}$ | $1 \oplus e_{GB} \oplus e_R \oplus e_{RGB}$ | $\mathbb{Z}_2$ | $\mathrm{Rep}(D_8)/(\mathbb{Z}_2 \times \mathbb{Z}_2)$ gSSB | 2 |
| 4 | $\mathcal{A}_{25}$ | $1 \oplus e_G \oplus e_{RB} \oplus e_{RGB}$ | $\mathbb{Z}_2$ | $\mathrm{Rep}(D_8)/(\mathbb{Z}_2 \times \mathbb{Z}_2)$ gSSB | 2 |
| 4 | $\mathcal{A}_{26}$ | $1 \oplus e_B \oplus e_{RG} \oplus e_{RGB}$ | $\mathbb{Z}_2$ | $\mathrm{Rep}(D_8)/(\mathbb{Z}_2 \times \mathbb{Z}_2)$ gSSB | 2 |
| 8 | $\mathcal{A}_{27}$ | $1 \oplus e_G \oplus e_R \oplus e_{RG} \oplus 2m_B$ | trivial | SPT | 1 |
| 8 | $\mathcal{A}_{28}$ | $1 \oplus e_B \oplus e_{RG} \oplus e_{RGB} \oplus 2m_{RG}$ | trivial | $\mathbb{Z}_2 \times \mathbb{Z}_2$ SSB | 4 |
| 8 | $\mathcal{A}_{29}$ | $1 \oplus e_{GB} \oplus e_R \oplus e_{RGB} \oplus 2m_{GB}$ | trivial | $\mathbb{Z}_2 \times \mathbb{Z}_2$ SSB | 4 |
| 8 | $\mathcal{A}_{30}$ | $1 \oplus e_B \oplus e_R \oplus e_{RB} \oplus 2m_G$ | trivial | SPT | 1 |
| 8 | $\mathcal{A}_{31}$ | $1 \oplus e_G \oplus e_{RB} \oplus e_{RGB} \oplus 2m_{RB}$ | trivial | $\mathbb{Z}_2 \times \mathbb{Z}_2$ SSB | 4 |
| 8 | $\mathcal{A}_{32}$ | $1 \oplus e_B \oplus e_G \oplus e_{GB} \oplus 2m_R$ | trivial | SPT | 1 |
| 8 | $\mathcal{A}_{33}$ | $1 \oplus e_{RGB} \oplus m_{GB} \oplus m_{RB} \oplus m_{RG}$ | trivial | $\mathcal{L}_{\mathcal{S}}$ and $\mathrm{Rep}(D_8)$ SSB | 5 |
| 8 | $\mathcal{A}_{34}$ | $1 \oplus e_B \oplus m_G \oplus m_R \oplus m_{RG}$ | trivial | $\mathbb{Z}_2$ SSB | 2 |
| 8 | $\mathcal{A}_{35}$ | $1 \oplus e_R \oplus m_B \oplus m_G \oplus m_{GB}$ | trivial | $\mathbb{Z}_2$ SSB | 2 |
| 8 | $\mathcal{A}_{36}$ | $1 \oplus e_G \oplus m_B \oplus m_R \oplus m_{RB}$ | trivial | $\mathbb{Z}_2$ SSB | 2 |
| 8 | $\mathcal{A}_{37}$ | $1 \oplus e_B \oplus e_G \oplus e_{GB} \oplus e_R \oplus e_{RB} \oplus e_{RG} \oplus e_{RGB}$ | trivial | $\mathrm{Rep}(D_8)/(\mathbb{Z}_2 \times \mathbb{Z}_2)$ SSB | 2 |

Table 4: Non-maximal condensable algebras in $\mathcal{Z}(\mathrm{Rep}(D_8))$ of dimension 4 (first column) have $\mathcal{S}' = \mathbb{Z}_2^\omega$ for the igSPT, (first row) which can only flow to one gapped phase (an SSB) or $\mathcal{S}' = \mathbb{Z}_2$, (following rows) in which case they describe phase transitions between two $\mathrm{Rep}(D_8)$-symmetric gapped phases (columns 2 and 3). The corresponding $\mathrm{Rep}(D_8)$-symmetric CFT is shown in the last column. We represent the functor for the example described in the section linked to in the first column and denote the $\mathrm{Rep}(D_8)$ symmetry generators as follows: $R$ in red, $G$ in green, $B$ in blue. Gray denotes the $\mathbb{Z}_2 \times \mathbb{Z}_2$ subsymmetry of $\mathrm{Rep}(D_8)$ (whose generators are $R$ and $G$) and black the full $\mathrm{Rep}(D_8)$.

| Gapless phase | Gapped phase $A$ | Gapped phase $B$ | $\mathrm{Rep}(D_8)$-symmetric CFT |
|---|---|---|---|
| **igSPT** $\mathcal{A}_8$ sec. 3.2 | $\mathbf{Rep}(D_8)/(\mathbb{Z}_2 \times \mathbb{Z}_2)$ **SSB** $\mathcal{A}_{37}$ sec. C.3.3 | - | $SU(2)_1$ ↺ sec. D.1.3 |
| **gSPT** $\mathcal{A}_{10}$ $\mathcal{A}_{11}$ $\mathcal{A}_{13}$ $\mathcal{A}_{14}$ $\mathcal{A}_{16}$ $\mathcal{A}_{17}$ sec. C.2.1 | **SPT** $\mathcal{A}_{30}$ $\mathcal{A}_{27}$ $\mathcal{A}_{32}$ $\mathcal{A}_{27}$ $\mathcal{A}_{32}$ $\mathcal{A}_{30}$ sec. C.3.1 | $\mathbb{Z}_2$ **SSB** $\mathcal{A}_{35}$ $\mathcal{A}_{35}$ $\mathcal{A}_{36}$ $\mathcal{A}_{36}$ $\mathcal{A}_{34}$ $\mathcal{A}_{34}$ sec. C.3.2 | Ising ↺ sec. D.1.1 |
| **gSPT** $\mathcal{A}_{21}$ $\mathcal{A}_{22}$ $\mathcal{A}_{23}$ sec. C.2.2 | **SPT** $\mathcal{A}_{27}$ $\mathcal{A}_{32}$ $\mathcal{A}_{30}$ sec. C.3.1 | $\mathbf{Rep}(D_8)/(\mathbb{Z}_2 \times \mathbb{Z}_2)$ **SSB** $\mathcal{A}_{37}$ $\mathcal{A}_{37}$ $\mathcal{A}_{37}$ sec. C.3.3 | Ising ↺ |
| $\mathbb{Z}_2$ **gSSB** $\mathcal{A}_9$ $\mathcal{A}_{12}$ $\mathcal{A}_{15}$ sec. C.2.4 | $\mathbb{Z}_2$ **SSB** $\mathcal{A}_{35}$ $\mathcal{A}_{36}$ $\mathcal{A}_{34}$ sec. C.3.2 | $\mathbb{Z}_2 \times \mathbb{Z}_2$ **SSB** $\mathcal{A}_{29}$ $\mathcal{A}_{31}$ $\mathcal{A}_{28}$ sec. C.3.4 | $\mathrm{Ising}_0 \oplus \mathrm{Ising}_1$ |
| $\mathbf{Rep}(D_8)/(\mathbb{Z}_2 \times \mathbb{Z}_2)$ **gSSB** $\mathcal{A}_{24}$ $\mathcal{A}_{25}$ $\mathcal{A}_{26}$ sec. C.2.5 | $\mathbf{Rep}(D_8)/(\mathbb{Z}_2 \times \mathbb{Z}_2)$ **SSB** $\mathcal{A}_{37}$ $\mathcal{A}_{37}$ $\mathcal{A}_{37}$ sec. C.3.3 | $\mathbb{Z}_2 \times \mathbb{Z}_2$ **SSB** $\mathcal{A}_{29}$ $\mathcal{A}_{31}$ $\mathcal{A}_{28}$ sec. C.3.4 | $\mathrm{Ising}_0 \oplus \mathrm{Ising}_1$ |
| Ising **igSSB** $\mathcal{A}_{18}$ $\mathcal{A}_{19}$ $\mathcal{A}_{20}$ sec. C.2.6 | $\mathbb{Z}_2 \times \mathbb{Z}_2$ **SSB** $\mathcal{A}_{28}$ $\mathcal{A}_{29}$ $\mathcal{A}_{31}$ sec. C.3.4 | $\mathbf{Rep}(D_8)$ **SSB** $\mathcal{A}_{33}$ $\mathcal{A}_{33}$ $\mathcal{A}_{33}$ sec. C.3.5 | $\mathrm{Ising}_0^m \oplus (\mathrm{Ising}_1^e)_{\sqrt{2}} \oplus (\mathrm{Ising}_2^e)_{\sqrt{2}}$ |

In terms of the SymTFT, we can express $\mathfrak{T}$ as a club sandwich of the form

$$
\mathfrak{T} \quad = \quad \underbrace{\boxed{\mathfrak{Z}(\mathcal{S})}}_{\mathfrak{B}_{\mathcal{S}}^{\mathrm{sym}}} \overbrace{\phantom{\Big|}}^{\mathcal{I}_\phi} \underbrace{\boxed{\mathfrak{Z}(\mathcal{S}')}}_{\mathfrak{B}_{\mathfrak{T}}^{\mathrm{phys}}} \quad , \tag{147}
$$

with a physical boundary $\mathfrak{B}_{\mathfrak{T}}^{\mathrm{phys}}$ of $\mathfrak{Z}(\mathcal{S}')$, a topological interface $\mathcal{I}_\phi$ from $\mathfrak{Z}(\mathcal{S})$ to $\mathfrak{Z}(\mathcal{S}')$ as-

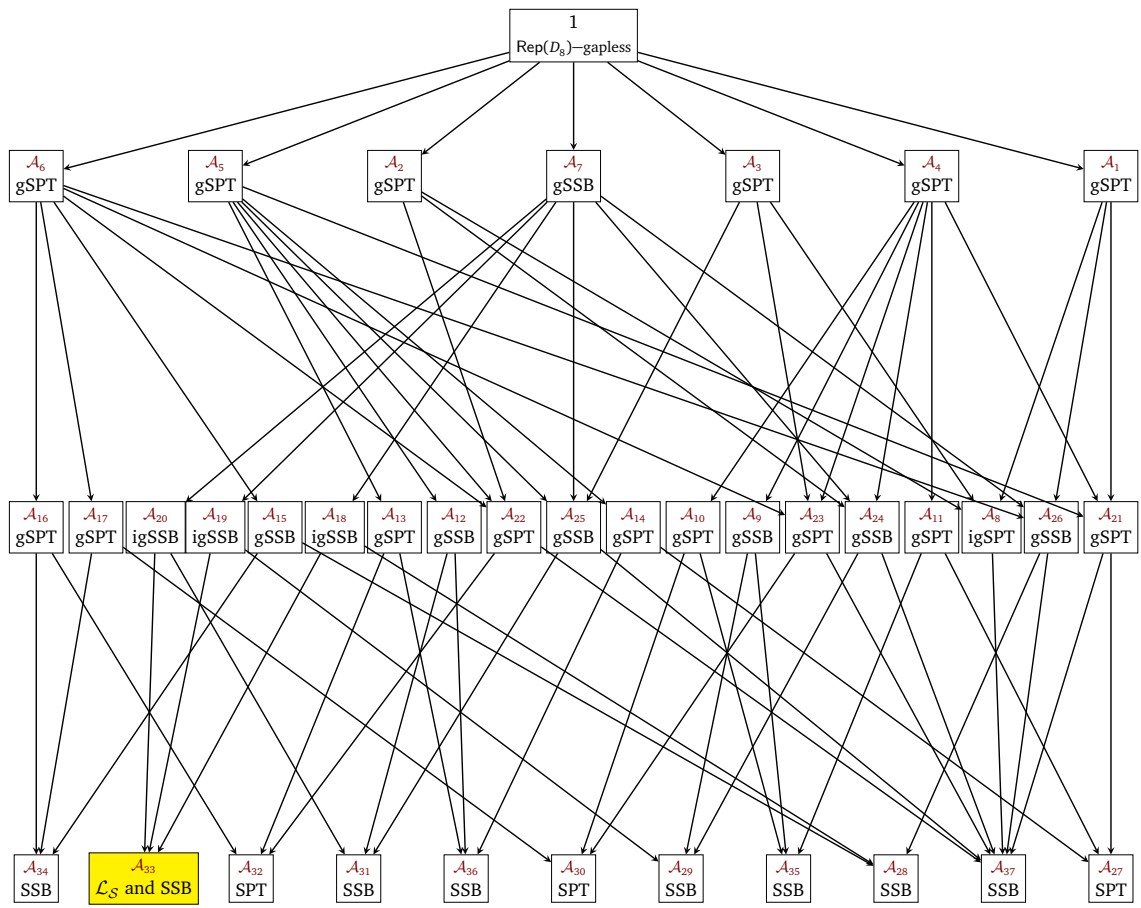

Figure 3: Hasse diagram for $\mathcal{Z}(\mathrm{Rep}(D_8))$ Phases. In each box we link to the condensable algebras, listed in table 3. The lowest level contains the maximal, i.e. Lagrangian, algebras. We pick as symmetry Lagrangian algebra $\mathcal{L}_{\mathrm{Rep}(D_8)}^{\mathrm{sym}}$ and classify all $\mathrm{Rep}(D_8)$-symmetric phases.

sociated to $\phi$, and a symmetry topological boundary $\mathfrak{B}_{\mathcal{S}}^{\mathrm{sym}}$ of $\mathfrak{Z}(\mathcal{S})$. The boundary $\mathfrak{B}_{\mathcal{S}}^{\mathrm{sym}}$ is associated to a Lagrangian algebra $\mathcal{L}_{\mathcal{S}}^{\mathrm{sym}} \in \mathcal{Z}(\mathcal{S})$ and the interface $\mathcal{I}_\phi$ is associated to a Lagrangian algebra of the folded model $\mathcal{L}_\phi \in \mathcal{Z}(\mathcal{S}) \boxtimes \overline{\mathcal{Z}(\mathcal{S}')}$. The part of $\mathcal{L}_\phi$ which does not involve non-trivial anyons of $\overline{\mathcal{Z}(\mathcal{S}')}$ describes a condensable algebra $\mathcal{A}_\phi \in \mathcal{Z}(\mathcal{S})$ satisfying the condition (52).

A gauging of a symmetry $\mathcal{S}$ is specified by a module category $\mathcal{M}$ of $\mathcal{S}$ which determines a topological boundary condition $\mathfrak{B}_{\mathcal{S}/\mathcal{M}}^{\mathrm{sym}}$ of $\mathfrak{Z}(\mathcal{S})$ having the property that the topological interfaces from $\mathfrak{B}_{\mathcal{S}}^{\mathrm{sym}}$ to $\mathfrak{B}_{\mathcal{S}/\mathcal{M}}^{\mathrm{sym}}$ are described by the module category $\mathcal{M}$. The boundary $\mathfrak{B}_{\mathcal{S}/\mathcal{M}}^{\mathrm{sym}}$ is associated to a Lagrangian algebra $\mathcal{L}_{\mathcal{S}/\mathcal{M}}^{\mathrm{sym}} \in \mathcal{Z}(\mathcal{S})$. The 2d QFT $\mathfrak{T}/\mathcal{M}$ obtained after performing the gauging $\mathcal{M}$ of the symmetry $\mathcal{S}$ is obtained by constructing the club sandwich with the boundary $\mathfrak{B}_{\mathcal{S}}^{\mathrm{sym}}$ replaced by $\mathfrak{B}_{\mathcal{S}/\mathcal{M}}^{\mathrm{sym}}$

$$
\left.\mathfrak{T}/\mathcal{M} \quad\right|\quad = \quad \boxed{\begin{array}{cc} \mathfrak{B}_{\mathcal{S}/\mathcal{M}}^{\mathrm{sym}} & \mathcal{I}_\phi \quad \mathfrak{B}_{\mathfrak{T}}^{\mathrm{phys}} \\ \mathfrak{Z}(\mathcal{S}) & \mathfrak{Z}(\mathcal{S}') \end{array}} \quad . \tag{148}
$$

Table 5: Non-maximal condensable algebras in $\mathcal{Z}(\mathrm{Rep}(D_8))$ with $\mathcal{S}' = \mathbb{Z}_4$ (first column) describe phase transitions between three $\mathrm{Rep}(D_8)$-symmetric gapped phases (columns 2,3 and 4). The CFT responsible for the transition, shown in the last column, is the 4-state Potts model regarded as $\mathrm{Rep}(D_8)$-symmetric.

| Gapless phase | Gapped phase $A$ | Gapped phase $B$ | Gapped phase $C$ | $\mathrm{Rep}(D_8)$-symmetric CFT |
|---|---|---|---|---|
| **gSPT** | **SPT** | $\mathrm{Rep}(D_8)/(\mathbb{Z}_2 \times \mathbb{Z}_2)$ SSB | $\mathbb{Z}_2 \times \mathbb{Z}_2$ SSB | |
| $\mathcal{A}_1$ | $\mathcal{A}_{27}$ | $\mathcal{A}_{37}$ | $\mathcal{A}_{28}$ | |
| $\mathcal{A}_2$ | $\mathcal{A}_{32}$ | $\mathcal{A}_{37}$ | $\mathcal{A}_{29}$ | 4-Potts ⟲ |
| $\mathcal{A}_3$ | $\mathcal{A}_{30}$ | $\mathcal{A}_{37}$ | $\mathcal{A}_{31}$ | |
| sec. C.1.1 | sec. C.3.1 | sec. C.3.3 | sec. C.3.4 | sec. D.2.1 |

Table 6: Non-maximal condensable algebras in $\mathcal{Z}(\mathrm{Rep}(D_8))$ with $\mathcal{S}' = \mathbb{Z}_2 \times \mathbb{Z}_2$ (first column) describe phase transitions between five $\mathrm{Rep}(D_8)$-symmetric gapped phases (columns 2 to 6). The $\mathrm{Rep}(D_8)$-symmetric CFT responsible for the transition is shown in the last column. $\mathsf{Ising}^2$ denotes the $\mathbb{Z}_2 \times \mathbb{Z}_2$-symmetric CFT $\mathsf{Ising} \times \mathsf{Ising}$. In the first row it is regarded as $\mathrm{Rep}(D_8)$-symmetric, whereas in the second there are two copies $\mathsf{Ising}_i^2$, for $i = 0, 1$, exchanged by the $B$ symmetry of $\mathrm{Rep}(D_8)$ and invariant under the $\mathbb{Z}_2 \times \mathbb{Z}_2$ subsymmetry of $\mathrm{Rep}(D_8)$.

| Gapless phase | Gapped phase $A$ | Gapped phase $B$ | Gapped phase $C$ | Gapped phase $D$ | Gapped phase $E$ | $\mathrm{Rep}(D_8)$-symmetric CFT |
|---|---|---|---|---|---|---|
| **gSPT** | **SPT** | **SPT** | $\frac{\mathrm{Rep}(D_8)}{(\mathbb{Z}_2 \times \mathbb{Z}_2)}$ SSB | $\mathbb{Z}_2$ SSB | $\mathbb{Z}_2 \times \mathbb{Z}_2$ SSB | |
| $\mathcal{A}_4$ | $\mathcal{A}_{27}$ | $\mathcal{A}_{30}$ | $\mathcal{A}_{37}$ | $\mathcal{A}_{35}$ | $\mathcal{A}_{29}$ | |
| $\mathcal{A}_5$ | $\mathcal{A}_{27}$ | $\mathcal{A}_{32}$ | $\mathcal{A}_{37}$ | $\mathcal{A}_{36}$ | $\mathcal{A}_{31}$ | $\mathsf{Ising}^2$ ⟲ |
| $\mathcal{A}_6$ | $\mathcal{A}_{30}$ | $\mathcal{A}_{32}$ | $\mathcal{A}_{37}$ | $\mathcal{A}_{34}$ | $\mathcal{A}_{28}$ | |
| sec. C.1.2 | sec. C.3.1 | sec. C.3.1 | sec. C.3.3 | sec. C.3.2 | sec. C.3.4 | sec. D.2.2 |
| $\frac{\mathrm{Rep}(D_8)}{(\mathbb{Z}_2 \times \mathbb{Z}_2)}$ **gSSB** | $\frac{\mathrm{Rep}(D_8)}{(\mathbb{Z}_2 \times \mathbb{Z}_2)}$ SSB | $\mathbb{Z}_2 \times \mathbb{Z}_2$ SSB | $\mathbb{Z}_2 \times \mathbb{Z}_2$ SSB | $\mathbb{Z}_2 \times \mathbb{Z}_2$ SSB | $\mathrm{Rep}(D_8)$ SSB | |
| $\mathcal{A}_7$ | $\mathcal{A}_{37}$ | $\mathcal{A}_{28}$ | $\mathcal{A}_{29}$ | $\mathcal{A}_{31}$ | $\mathcal{A}_{33}$ | ⟲ $\mathsf{Ising}_0^2 \oplus \mathsf{Ising}_1^2$ ⟲ |
| sec. C.1.3 | sec. C.3.3 | sec. C.3.4 | sec. C.3.4 | sec. C.3.4 | sec. C.3.5 | sec. D.2.3 |

Now, the condensable algebra $\mathcal{A}_\phi$ may not satisfy the condition (52) with respect to the symmetry Lagrangian algebra $\mathcal{L}_{\mathcal{S}/\mathcal{M}}^{\mathrm{sym}}$, i.e. we may have

$$\mathcal{A}_\phi \cap \mathcal{L}_{\mathcal{S}/\mathcal{M}}^{\mathrm{sym}} \supsetneq 1. \tag{149}$$

In fact the number of universes in the gauged theory $\mathfrak{T}/\mathcal{M}$ is

$$n_\mathcal{M} = \sum_a n_{a^*}^\phi n_{a,\mathcal{M}}^{\mathrm{sym}}, \tag{150}$$

cf. (121), where $n_a^\phi$ are the coefficients of the simple anyons $\mathbf{Q}_a$ in $\mathcal{A}_\phi$ and $n_{a,\mathcal{M}}^{\mathrm{sym}}$ are the coefficients of $\mathbf{Q}_a$ in $\mathcal{L}_{\mathcal{S}/\mathcal{M}}^{\mathrm{sym}}$

$$\mathcal{L}_{\mathcal{S}/\mathcal{M}}^{\mathrm{sym}} = \bigoplus_a n_{a,\mathcal{M}}^{\mathrm{sym}} \mathbf{Q}_a. \tag{151}$$

Note that $n_\mathcal{M}$ may be equal to 1, but is generically greater than 1. Thus, it may happen that gauging trivially acting symmetries does not lead to decomposition. These $n_\mathcal{M}$ number of universes are permuted into each other by the action of the dual symmetry $\mathcal{S}/\mathcal{M}$ obtained after gauging. Note that the $\mathcal{S}/\mathcal{M}$ symmetry is also not realized faithfully on $\mathfrak{T}/\mathcal{M}$ if the original $\mathcal{S}$ symmetry is not realized faithfully on $\mathfrak{T}$. This is because the confined charges are a property only of the interface $\mathcal{I}_\phi$ or equivalently of the functor (146), which is an invariant of the gauging procedure, as the gauging procedure only manipulates the symmetry boundary while leaving $\mathcal{I}_\phi$ invariant.

The properties of $\mathfrak{T}/\mathcal{M}$ including the number of universes and the relative Euler terms (also known as dilaton shifts) between the different universes, along with the precise realization of the dual symmetry $\mathcal{S}/\mathcal{M}$ on $\mathfrak{T}/\mathcal{M}$ can be computed by studying the club sandwich as described in detail in [10]. In the language of this reference, the gauged theory $\mathfrak{T}/\mathcal{M}$ is obtained by applying a KT transformation associated to the topological interface $\mathcal{I}_\phi$ on the original theory $\mathfrak{T}$. The reference also discusses how the (possibly non-topological) local operators of $\mathfrak{T}$ are transformed under the gauging, and how their charges are modified realizing the functor (146).

One can equally well begin with a symmetry $\mathcal{S}$ that is realized in a decomposed fashion on a 2d QFT $\mathfrak{T}$ having multiple universes. In such a situation, $\mathcal{S}$ is never realized faithfully, i.e. the sandwich construction of $\mathfrak{T}$ can always be decomposed as in (147) with the condensable algebra $\mathcal{A}_\phi$ associated to the interface $\mathcal{I}_\phi$ satisfying

$$\mathcal{A}_\phi \cap \mathcal{L}_\mathcal{S}^{\mathrm{sym}} \supsetneq 1 \,. \tag{152}$$

The $\mathcal{M}$ gauging of $\mathcal{S}$ is again implemented simply by changing the symmetry boundary condition and evaluating the club sandwich (148).

**Decomposition interpretations of the Hasse diagram**  Finally, all possible (non-faithful and possibly decomposed) realizations of a symmetry $\mathcal{S}$ are in one-to-one correspondence with $\mathcal{S}$-symmetric gapless and gapped phases, and hence are described by the same Hasse diagram. In fact, the IR theories for systems in these $\mathcal{S}$-symmetric phases are examples of theories manifesting the corresponding realizations of the symmetry $\mathcal{S}$. We thus have the following one-to-one correspondence:

1. Canonical gSPT: The top of the Hasse diagram corresponds to the symmetry $\mathcal{S}$ being realized faithfully.

2. gSPT: As we move down in the Hasse diagram, the $\mathcal{S}$ symmetry is realized more and more non-faithfully. A gSPT phase corresponds to a non-faithful realization of symmetry $\mathcal{S}$ such that there is no decomposition.

3. gSSB: Corresponds to a non-faithful realization of symmetry $\mathcal{S}$ with the symmetry $\mathcal{S}$ realized in a decomposed fashion on multiple universes.

4. igSPT: The edges of the Hasse diagram describe paths in which new confined charges are added. In other words, they describe how a realization of $\mathcal{S}$ can be made more non-faithful. An igSPT phase corresponds to a non-decomposed non-faithful realization of $\mathcal{S}$, which cannot be made more non-faithful without making $\mathcal{S}$ act in a decomposed way.

5. igSSB: Corresponds to a decomposed non-faithful realization of $\mathcal{S}$, which cannot be made more non-faithful without increasing the amount of decomposition.

The result of gauging $\mathcal{S}$ to $\mathcal{S}/\mathcal{M}$ for a (non-faithful, possibly decomposed) realization of $\mathcal{S}$, corresponding to a node $X$ of the $\mathcal{S}$-Hasse diagram, is encoded in the same node of the $\mathcal{S}/\mathcal{M}$-Hasse diagram.

**Example: $S_3 \to \mathrm{Rep}(S_3)$.**  Let us consider $\mathcal{S} = \mathrm{Vec}_{S_3}$, i.e. non-anomalous $S_3$ symmetry, whose Hasse diagram appears in figure 2. The various possible realizations of $S_3$ symmetry are:

1. $\mathcal{A} = (1,1)$: Faithfully realized $S_3$ symmetry.

2. $\mathcal{A} = (1,1) \oplus (1,1_-)$: Two universes exchanged by $\mathbb{Z}_2$ projection of $S_3$ symmetry, with the $\mathbb{Z}_3$ subgroup acting faithfully within each universe.

3. $\mathcal{A} = (1,1) \oplus (1,E)$: Three universes permuted by $S_3$ such that a $\mathbb{Z}_2$ subgroup of $S_3$ acts faithfully in each universe.

4. $\mathcal{A} = (1,1) \oplus (a,1)$: $\mathbb{Z}_2$ projection of $S_3$ symmetry is realized faithfully on a single universe.

5. $\mathcal{A} = (1,1) \oplus (a,1) \oplus (b,+)$: All of $S_3$ realized non-faithfully on a single universe.

6. $\mathcal{A} = (1,1) \oplus (1,1_-) \oplus 2(a,1)$: Two universes exchanged by $\mathbb{Z}_2$ projection of $S_3$ symmetry, with the $\mathbb{Z}_3$ subgroup acting non-faithfully within each universe.

7. $\mathcal{A} = (1,1) \oplus (1,1_-) \oplus 2(1,E)$: Six universes permuted by $S_3$.

8. $\mathcal{A} = (1,1) \oplus (1,E) \oplus (b,+)$: Three universes permuted by $S_3$ such that a $\mathbb{Z}_2$ subgroup of $S_3$ preserves each universe but acts non-faithfully within it.

We now perform full gauging of $S_3$ symmetry corresponding to $\mathcal{M} = \mathsf{Vec}$, for which the symmetry obtained after gauging is the non-invertible symmetry $\mathcal{S}/\mathcal{M} = \mathsf{Rep}(S_3)$. Its Hasse diagram also appears in figure 2. The results of $S_3$ gauging and the realizations of $\mathsf{Rep}(S_3)$ symmetry in them are:

1. $\mathcal{A} = (1,1)$: Faithfully realized $\mathsf{Rep}(S_3)$ symmetry.

2. $\mathcal{A} = (1,1) \oplus (1,1_-)$: Single universe on which the $\mathbb{Z}_2$ subgroup of $\mathsf{Rep}(S_3)$ generated by $1_-$ acts non-faithfully, while the non-invertible element $E$ acts faithfully via a $\mathbb{Z}_3$ symmetry according to the relation $E = P \oplus P^2$, where $P$ is the generator of the $\mathbb{Z}_3$ symmetry.

3. $\mathcal{A} = (1,1) \oplus (1,E)$: Single universe on which the $\mathbb{Z}_2$ subgroup of $\mathsf{Rep}(S_3)$ generated by $1_-$ acts faithfully and $E$ acts non-faithfully as $E = 1 \oplus 1_-$.

4. $\mathcal{A} = (1,1) \oplus (a,1)$: Two universes related to each other by gauging of a faithfully acting $\mathbb{Z}_2$ symmetry along with a non-trivial dilaton shift (relative Euler term). The $\mathbb{Z}_2$ subgroup of $\mathsf{Rep}(S_3)$ generated by $1_-$ acts non-faithfully in the first universe and faithfully via the $\mathbb{Z}_2$ symmetry of the second universe. The non-invertible element $E$ exchanges the two universes and also acts within the first universe via its $\mathbb{Z}_2$ symmetry.

5. $\mathcal{A} = (1,1) \oplus (a,1) \oplus (b,+)$: Three universes $i \in \{0,1,2\}$ with relative Euler terms between universes 0 and 1, and universes 0 and 2, but no relative Euler terms between universes 1 and 2. The $1_-$ symmetry acts by exchanging universes 1 and 2, while preserving universe 0, inside which it acts non-faithfully. The $E$ symmetry acts within universe 0, and also sends universe 0 to universes 1 and 2, while also sending universes 1 and 2 to universe 0.

6. $\mathcal{A} = (1,1) \oplus (1,1_-) \oplus 2(a,1)$: Three universes with no relative Euler terms which are all preserved by $1_-$ that is realized non-faithfully in each universe. The symmetry $E$ acts by sending each universe into the sum of the other two.

7. $\mathcal{A} = (1,1) \oplus (1,1_-) \oplus 2(1,E)$: Single universe in which all of $\mathsf{Rep}(S_3)$ is realized non-faithfully.

8. $\mathcal{A} = (1,1) \oplus (1,E) \oplus (b,+)$: Two universes exchanged by $1_-$ and $E$ acting non-faithfully as $E = 1 \oplus 1_-$.

The examples of decomposition of $\mathsf{Rep}(S_3)$ symmetry resulting from gauging $S_3$ symmetry acting (partially) trivially (i.e. non-faithfully) on a single universe are provided by $\mathcal{A} = (1, 1) \oplus (a, 1)$ and $\mathcal{A} = (1, 1) \oplus (a, 1) \oplus (b, +)$.

# 7 Conclusions

The main focus of this paper is the systematic exploration of gapped and gapless phases in (1+1)d theories. In the realm of these theories, fusion category symmetries and their SymTFTs, as well as Drinfeld centers can be applied to achieve a comprehensive picture of all phases, and their interconnections – depicted by the Hasse diagram of phases.

The approach to study gapped phases and gapless phases using the SymTFT for a categorical symmetry extends beyond the (1+1)d realm, and will give new insights into QFTs and their IR phases in higher dimensions, in the presence of categorical symmetries. There has been enormous activity in particular in high energy physics in the study of generalized, non-invertible symmetries, in (2+1)d and (3+1)d, for reviews on this topic see [35,36,55–58], and a list of recent papers to illustrate the sheer wealth of activity and ubiquity of such symmetries see [1,2,9,10,17,25,30–34,52,59–157]. Some of the challenges will be in the analysis of all gapped boundary conditions, as well as the mathematically precise definition of condensable "algebra" in the context of higher fusion-categories.

# Acknowledgments

We thank Lea Bottini, Christian Copetti, Po-Shen Hsin, Nick Jones, Heidar Moradi, Sanjay Moudgalya, Shu-Heng Shao, Apoorv Tiwari and Yunqin Zheng for discussions at various stages of this work. LB thanks Niels Bohr International Academy for hospitality during some of this work.

**Funding information**   LB is funded as a Royal Society University Research Fellow through grant URF\R1\231467. SSN thanks King's College London for hospitality during some of this work. The work of SSN and AW is supported by the UKRI Frontier Research Grant, underwriting the ERC Advanced Grant "Generalized Symmetries in Quantum Field Theory and Quantum Gravity".

# A  Fusion rules of $\mathcal{Z}(\mathsf{Rep}(D_8))$

In Table 7 we summarize the fusion rules for $\mathcal{Z}(\mathsf{Rep}(D_8))$, which we computed from the formulae in [45]. These fusion rules have already appeared in reference [43].

# B  Condensable algebras of $\mathcal{Z}(\mathsf{Rep}(D_8))$

In this appendix we provide some details on the condensable algebras and reduced topological orders for $\mathcal{Z}(\mathsf{Rep}(D_8))$. These are already listed and their partial ordering shown in a Hasse diagram in section 5.

Table 7: Fusion rules (up to permutation of colors) of the anyons in $\mathcal{Z}(\mathrm{Rep}(D_8))$.

| $\otimes$ | $e_R$ | $e_{RG}$ | $m_R$ | $m_{RG}$ | $m_{RB}$ | $s_{RGB}$ |
|---|---|---|---|---|---|---|
| $e_R$ | 1 | $e_G$ | $f_R$ | $f_{RG}$ | $f_{RB}$ | $\bar{s}_{RGB}$ |
| $e_G$ | $e_{RG}$ | $e_R$ | $m_R$ | $f_{RG}$ | $m_{RB}$ | $\bar{s}_{RGB}$ |
| $e_B$ | $e_{RB}$ | $e_{RGB}$ | $m_R$ | $m_{RG}$ | $f_{RB}$ | $\bar{s}_{RGB}$ |
| $e_{RG}$ | $e_G$ | 1 | $f_R$ | $m_{RG}$ | $f_{RB}$ | $s_{RGB}$ |
| $e_{RGB}$ | $e_{GB}$ | $e_B$ | $f_R$ | $m_{RG}$ | $m_{RB}$ | $\bar{s}_{RGB}$ |
| $m_R$ | $f_R$ | $f_R$ | $1 \oplus e_G \oplus e_B \oplus e_{GB}$ | $m_G \oplus f_G$ | $m_B \oplus f_B$ | $m_{GB} \oplus f_{GB}$ |
| $m_G$ | $m_G$ | $f_G$ | $m_{RG} \oplus f_{RG}$ | $m_R \oplus f_R$ | $s_{RGB} \oplus \bar{s}_{RGB}$ | $m_{RB} \oplus f_{RB}$ |
| $m_B$ | $m_B$ | $m_B$ | $m_{RB} \oplus f_{RB}$ | $s_{RGB} \oplus \bar{s}_{RGB}$ | $m_R \oplus f_R$ | $m_{RG} \oplus f_{RG}$ |
| $m_{RG}$ | $f_{RG}$ | $m_{RG}$ | $m_G \oplus f_G$ | $1 \oplus e_{RG} \oplus e_{RGB} \oplus e_B$ | $m_{GB} \oplus f_{GB}$ | $m_B \oplus f_B$ |
| $s_{RGB}$ | $\bar{s}_{RGB}$ | $s_{RGB}$ | $m_{GB} \oplus f_{GB}$ | $m_B \oplus f_B$ | $m_G \oplus f_G$ | $1 \oplus e_{RG} \oplus e_{GB} \oplus e_{RB}$ |

## B.1 Condensable algebras of dimension 2

The Drinfeld center of $\mathrm{Rep}(D_8)$ has 7 condensable algebras of dimension 2. However, only 6 of these have a trivial overlap with the Lagrangian algebra for the $\mathrm{Rep}(D_8)$ symmetry

$$\mathcal{L}^{\mathrm{sym}}_{\mathrm{Rep}(D_8)} = 1 \oplus e_{RGB} \oplus m_{GB} \oplus m_{RB} \oplus m_{RG}\,. \tag{B.1}$$

We can further divide these 6 condensable algebras into two groups, those that reduce the center $\mathcal{Z}(\mathrm{Rep}(D_8))$ to $\mathcal{Z}(\mathrm{Vec}_{\mathbb{Z}_4})$ and those that reduce it to $\mathcal{Z}(\mathrm{Vec}_{\mathbb{Z}_2 \times \mathbb{Z}_2})$.

The two conditions that can help us realize which 2d condensable algebra reduces to which center are:

- The anyons $s_{RGB}, \bar{s}_{RGB} \in \mathcal{Z}(\mathrm{Rep}(D_8))$ braid trivially with the given algebra.

- Anyon spins match on both sides of the condensable interface.

All elements of $\mathcal{Z}(\mathrm{Rep}(D_8))$ have either spin 1 or $-1$, except for $s_{RGB}, \bar{s}_{RGB}$ which have spins $\pm i$. Thus if the first condition is *not* satisfied then the reduced center must be $\mathcal{Z}(\mathrm{Vec}_{\mathbb{Z}_2 \times \mathbb{Z}_2})$. On the other hand, if the first condition is satisfied then the second condition ensures the spins match those of $\mathcal{Z}(\mathrm{Vec}_{\mathbb{Z}_4})$.

### B.1.1 $\mathcal{Z}(\mathrm{Vec}_{\mathbb{Z}_4})$ reduced center

Non-trivial elements of $\mathcal{Z}(\mathrm{Rep}(D_8))$ that braid trivially with $s_{RGB}, \bar{s}_{RGB}$ are $e_{RG}$, $e_{GB}$, and $e_{RB}$. Out of these we can build the following dimension 2 condensable algebras:

- $\mathcal{A}_1 = 1 \oplus e_{RG}$,

- $\mathcal{A}_2 = 1 \oplus e_{GB}$,

- $\mathcal{A}_3 = 1 \oplus e_{RB}$.

We will now go on to show these reduce $\mathcal{Z}(\mathrm{Rep}(D_8))$ to $\mathcal{Z}(\mathrm{Vec}_{\mathbb{Z}_4})$, i.e.

$$\mathcal{Z}(\mathrm{Vec}_{\mathbb{Z}_4}) = \mathcal{Z}(\mathrm{Rep}(D_8))/\mathcal{A}_i\,, \qquad i \in \{1, 2, 3\}\,. \tag{B.2}$$

To show this, we have to show that the topological interface $\mathcal{I}_i$ corresponding to $\mathcal{A}_i$ is indeed an interface between $\mathcal{Z}(\mathrm{Rep}(D_8))$ and $\mathcal{Z}(\mathrm{Vec}_{\mathbb{Z}_4})$. This can be shown by exhibiting a (folded) Lagrangian algebra $\mathcal{L}_i$ in $\mathcal{Z}(\mathrm{Vec}_{D_8}) \boxtimes \overline{\mathcal{Z}(\mathrm{Vec}_{\mathbb{Z}_4})}$ of which $\mathcal{A}_i$ is a subalgebra. Details on how to determine the reduced topological order by condensing a non-Lagrangian condensable algebra can be found in Appendix B of [10] or in [158, 159].

In the present case, the Lagrangian algebra $\mathcal{L}_1$ can then be found as

$$
\begin{aligned}
\mathcal{L}_1 = 1 &\oplus e_{RG} \oplus [e_{RGB} \oplus e_B]\overline{e}^2 \oplus [e_R \oplus e_G]\overline{m}^2 \\
&\oplus [e_{GB} \oplus e_{RB}]\overline{e}^2\overline{m}^2 \oplus m_B\overline{m} \oplus m_B\overline{m}^3 \\
&\oplus m_{RG}\overline{e} \oplus m_{RG}\overline{e}^3 \oplus f_B\overline{e}^2\overline{m} \oplus f_B\overline{e}^2\overline{m}^3 \\
&\oplus f_{RG}\overline{em}^2 \oplus f_{RG}\overline{e}^3\overline{m}^2 \oplus s_{RGB}\overline{em} \\
&\oplus s_{RGB}\overline{e}^3\overline{m}^3 \oplus \overline{s}_{RGB}\overline{e}^3\overline{m} \oplus \overline{s}_{RGB}\overline{em}^3 .
\end{aligned}
\tag{B.3}
$$

One can check that all elements in $\mathcal{L}_1$ are indeed bosons and have trivial mutual braidings with each other, among the other conditions for the algebra to be Lagrangian (maximal). One can thus see that $\mathcal{A}_1$ reduces $\mathcal{Z}(\mathrm{Rep}(D_8))$ to $\mathcal{Z}(\mathrm{Vec}_{\mathbb{Z}_4})$.

As a side note, this result was expected as $\mathcal{Z}(\mathrm{Rep}(D_8)) = \mathcal{Z}(\mathrm{Vec}_{D_8})$ which one can obtain by gauging the $\mathbb{Z}_2$ outer automorphism symmetry of $\mathcal{Z}(\mathrm{Vec}_{\mathbb{Z}_4})$ which exchanges $e^i m^j \leftrightarrow e^{4-i} m^{4-j}$ for $i, j \in \{0, 1, 2, 3\}$. Gauging this $\mathbb{Z}_2$ symmetry of $\mathcal{Z}(\mathrm{Vec}_{\mathbb{Z}_4})$ introduces a new line in the theory which is precisely $e_{RG}$. Thus it is not surprising that by starting with $\mathcal{Z}(\mathrm{Vec}_{D_8})$ and condensing $e_{RG}$, one ends up back in $\mathcal{Z}(\mathrm{Vec}_{\mathbb{Z}_4})$.

Analogously, for $\mathcal{A}_2$ one finds the folded Lagrangian algebra to be

$$
\begin{aligned}
\mathcal{L}_2 = 1 &\oplus e_{GB} \oplus [e_R \oplus e_{RGB}]\overline{e}^2 \oplus [e_G \oplus e_B]\overline{m}^2 \\
&\oplus [e_{RG} \oplus e_{RB}]\overline{e}^2\overline{m}^2 \oplus m_R\overline{m} \oplus m_R\overline{m}^3 \\
&\oplus m_{GB}\overline{e} \oplus m_{GB}\overline{e}^3 \oplus f_R\overline{e}^2\overline{m} \oplus f_R\overline{e}^2\overline{m}^3 \\
&\oplus f_{GB}\overline{em}^2 \oplus f_{GB}\overline{e}^3\overline{m}^2 \oplus s_{RGB}\overline{em} \\
&\oplus s_{RGB}\overline{e}^3\overline{m}^3 \oplus \overline{s}_{RGB}\overline{e}^3\overline{m} \oplus \overline{s}_{RGB}\overline{em}^3 ,
\end{aligned}
\tag{B.4}
$$

and for $\mathcal{A}_3$ similarly

$$
\begin{aligned}
\mathcal{L}_3 = 1 &\oplus e_{RB} \oplus [e_G \oplus e_{RGB}]\overline{e}^2 \oplus [e_R \oplus e_B]\overline{m}^2 \\
&\oplus [e_{RG} \oplus e_{GB}]\overline{e}^2\overline{m}^2 \oplus m_G\overline{m} \oplus m_G\overline{m}^3 \\
&\oplus m_{RB}\overline{e} \oplus m_{RB}\overline{e}^3 \oplus f_G\overline{e}^2\overline{m} \oplus f_G\overline{e}^2\overline{m}^3 \\
&\oplus f_{RB}\overline{em}^2 \oplus f_{RB}\overline{e}^3\overline{m}^2 \oplus s_{RGB}\overline{em} \\
&\oplus s_{RGB}\overline{e}^3\overline{m}^3 \oplus \overline{s}_{RGB}\overline{e}^3\overline{m} \oplus \overline{s}_{RGB}\overline{em}^3 .
\end{aligned}
\tag{B.5}
$$

One can then see that (B.2) is indeed satisfied. Equivalent Lagrangian algebras can be determined by relabeling $e \leftrightarrow m$ in the above expressions.

Furthermore, by condensing $1 \oplus \overline{e}^2\overline{m}^2$ in any of the three folded algebras above, one precisely obtains the folded algebra $\mathcal{L}_\phi$ in (72) which corresponds to the reduced center $\mathcal{Z}(\mathrm{Vec}_{\mathbb{Z}_2}^\omega)$. However, since $\mathrm{Vec}_{\mathbb{Z}_4}$ admits a fiber functor, taking the symmetry boundary to be (B.1) and condensing $\mathcal{A}_i$ for any $i \in \{1, 2, 3\}$ leads to a non-intrinsic gSPT. This is contrary to the case of condensing (71), which leads to an intrinsic gSPT as $\mathrm{Vec}_{\mathbb{Z}_2}^\omega$ does not admit a fiber functor.

### B.1.2 $\mathcal{Z}(\mathrm{Vec}_{\mathbb{Z}_2 \times \mathbb{Z}_2})$ reduced center

The 3 remaining non-Lagrangian algebras of dimension 2 that do not braid trivially with $s_{RGB}, \overline{s}_{RGB}$ and have a *trivial* intersection with $\mathcal{L}_{\mathrm{Rep}(D_8)}^{\mathrm{sym}}$ are

- $\mathcal{A}_4 = 1 \oplus e_R$,

- $\mathcal{A}_5 = 1 \oplus e_G$,

- $\mathcal{A}_6 = 1 \oplus e_B$,

together with the 1 remaining 2d condensable algebra that does not braid trivially with $s_{RGB}, \bar{s}_{RGB}$ but has a *non-trivial* intersection with $\mathcal{L}^{\mathrm{sym}}_{\mathrm{Rep}(D_8)}$

- $\mathcal{A}_7 = 1 \oplus e_{RGB}$.

We will now go on to show these reduce $\mathcal{Z}(\mathrm{Rep}(D_8))$ to $\mathcal{Z}(\mathrm{Vec}_{\mathbb{Z}_2 \times \mathbb{Z}_2})$, i.e.

$$\mathcal{Z}(\mathrm{Vec}_{\mathbb{Z}_2 \times \mathbb{Z}_2}) = \mathcal{Z}(\mathrm{Rep}(D_8))/\mathcal{A}_j, \qquad j \in \{4, 5, 6, 7\}. \tag{B.6}$$

First one can notice that in the absence of $s_{RGB}, \bar{s}_{RGB}$ terms in the folded Lagrangian algebra due to non-trivial braiding with $\mathcal{A}_j$, the reduced center will only include anyons of spin $\pm 1$. By also considering the dimension of this reduced center, $\mathcal{Z}(\mathrm{Vec}_{\mathbb{Z}_2 \times \mathbb{Z}_2})$ becomes the obvious candidate.

Following similar steps as outlined previously, one finds a folded Lagrangian algebra for $\mathcal{A}_4$ to be

$$\begin{aligned}
\mathcal{L}_4 = {}& 1 \oplus e_R \oplus [e_{RGB} \oplus e_{GB}]\bar{e}_1\bar{e}_2 \\
& \oplus [e_{RG} \oplus e_G]\overline{m}_1\overline{m}_2 \oplus [e_B \oplus e_{RB}]\bar{e}_1\bar{e}_2\overline{m}_1\overline{m}_2 \\
& \oplus m_B\overline{m}_2 \oplus m_B\overline{m}_1 \oplus m_{GB}\bar{e}_1 \\
& \oplus m_{GB}\bar{e}_2 \oplus m_G\bar{e}_1\overline{m}_2 \oplus m_G\bar{e}_2\overline{m}_1 \\
& \oplus f_B\bar{e}_1\bar{e}_2\overline{m}_2 \oplus f_B\bar{e}_1\bar{e}_2\overline{m}_1 \oplus f_{GB}\bar{e}_1\overline{m}_1\overline{m}_2 \\
& \oplus f_{GB}\bar{e}_2\overline{m}_1\overline{m}_2 \oplus f_G\bar{e}_2\overline{m}_2 \oplus f_G\bar{e}_1\overline{m}_1,
\end{aligned} \tag{B.7}$$

where $e_1^i e_2^j m_1^i m_2^j \in \mathcal{Z}(\mathrm{Vec}_{\mathbb{Z}_2 \times \mathbb{Z}_2})$ for $i, j \in \{0, 1\}$. One can also relabel $e_1 \leftrightarrow m_1$ and $e_2 \leftrightarrow m_2$, or $e_1 \leftrightarrow e_2$ and $m_1 \leftrightarrow m_2$. From the folded algebra it is also apparent that it is possible to obtain the center $\mathcal{Z}(\mathrm{Rep}(D_8))$ from $\mathcal{Z}(\mathrm{Vec}_{\mathbb{Z}_2 \times \mathbb{Z}_2})$ by gauging the outer automorphism $m_1 \leftrightarrow m_2$, $e_1 \leftrightarrow e_2$, with $e_1 e_2$, $m_1 m_2$ left invariant.

Analogously, for $\mathcal{A}_5$ one finds a folded Lagrangian algebra to be

$$\begin{aligned}
\mathcal{L}_5 = {}& 1 \oplus e_G \oplus [e_{RGB} \oplus e_{RB}]\bar{e}_1\bar{e}_2 \\
& \oplus [e_B \oplus e_{GB}]\overline{m}_1\overline{m}_2 \oplus [e_{RG} \oplus e_R]\bar{e}_1\bar{e}_2\overline{m}_1\overline{m}_2 \\
& \oplus m_R\overline{m}_2 \oplus m_R\overline{m}_1 \oplus m_{RB}\bar{e}_1 \\
& \oplus m_{RB}\bar{e}_2 \oplus m_B\bar{e}_1\overline{m}_2 \oplus m_B\bar{e}_2\overline{m}_1 \\
& \oplus f_R\bar{e}_1\bar{e}_2\overline{m}_2 \oplus f_R\bar{e}_1\bar{e}_2\overline{m}_1 \oplus f_{RB}\bar{e}_1\overline{m}_1\overline{m}_2 \\
& \oplus f_{RB}\bar{e}_2\overline{m}_1\overline{m}_2 \oplus f_B\bar{e}_2\overline{m}_2 \oplus f_B\bar{e}_1\overline{m}_1,
\end{aligned} \tag{B.8}$$

and for $\mathcal{A}_6$ similarly

$$\begin{aligned}
\mathcal{L}_6 = {}& 1 \oplus e_B \oplus [e_R \oplus e_{RB}]\bar{e}_1\bar{e}_2 \\
& \oplus [e_G \oplus e_{GB}]\overline{m}_1\overline{m}_2 \oplus [e_{RG} \oplus e_{RGB}]\bar{e}_1\bar{e}_2\overline{m}_1\overline{m}_2 \\
& \oplus m_R\overline{m}_2 \oplus m_R\overline{m}_1 \oplus m_G\bar{e}_1 \\
& \oplus m_G\bar{e}_2 \oplus m_{RG}\bar{e}_1\overline{m}_2 \oplus m_{RG}\bar{e}_2\overline{m}_1 \\
& \oplus f_R\bar{e}_1\bar{e}_2\overline{m}_2 \oplus f_R\bar{e}_1\bar{e}_2\overline{m}_1 \oplus f_G\bar{e}_1\overline{m}_1\overline{m}_2 \\
& \oplus f_G\bar{e}_2\overline{m}_1\overline{m}_2 \oplus f_{RG}\bar{e}_2\overline{m}_2 \oplus f_{RG}\bar{e}_1\overline{m}_1.
\end{aligned} \tag{B.9}$$

Finally for $\mathcal{A}_7$ one finds

$$\begin{aligned}
\mathcal{L}_7 = {}& 1 \oplus e_{RGB} \oplus [e_R \oplus e_{GB}]\bar{e}_2\overline{m}_1 \\
& \oplus [e_G \oplus e_{RB}]\bar{e}_1\overline{m}_2 \oplus [e_{RG} \oplus e_B]\bar{e}_1\bar{e}_2\overline{m}_1\overline{m}_2 \\
& \oplus m_{RB}\overline{m}_2 \oplus m_{RB}\bar{e}_1 \oplus m_{GB}\overline{m}_1 \oplus m_{GB}\bar{e}_2 \\
& \oplus m_{RG}\overline{m}_1\overline{m}_2 \oplus m_{RG}\bar{e}_1\bar{e}_2 \\
& \oplus f_{RB}\bar{e}_2\overline{m}_1\overline{m}_2 \oplus f_{RB}\bar{e}_1\bar{e}_2\overline{m}_1 \oplus f_{GB}\bar{e}_1\overline{m}_1\overline{m}_2 \\
& \oplus f_{GB}\bar{e}_1\bar{e}_2\overline{m}_2 \oplus f_{RG}\bar{e}_2\overline{m}_2 \oplus f_{RG}\bar{e}_1\overline{m}_1,
\end{aligned} \tag{B.10}$$

where it is apparent from the folded algebra that it is possible to obtain the center $\mathcal{Z}(\text{Rep}(D_8))$ from $\mathcal{Z}(\text{Vec}_{\mathbb{Z}_2 \times \mathbb{Z}_2})$ in this case by gauging the outer automorphism $e_1 \leftrightarrow m_2$, $e_2 \leftrightarrow m_1$, with $e_1 m_2$, $e_2 m_1$ left invariant.

## B.2 Condensable algebras of dimension 4

Out of the 19 condensable algebras of dimension 4, there are 10 that have a trivial intersection with $\mathcal{L}^{\text{sym}}_{\text{Rep}(D_8)}$, out of which only 1 reduces $\mathcal{Z}(\text{Rep}(D_8))$ to $\mathcal{Z}(\text{Vec}^{\omega}_{\mathbb{Z}_2})$ and the other 9 to $\mathcal{Z}(\text{Vec}_{\mathbb{Z}_2})$. Hence there is only one intrinsic gSPT for $\mathcal{L}^{\text{sym}}_{\text{Rep}(D_8)}$ and the others are non-instrinsic.

Only the algebra $\mathcal{A}_8$ gives rise to $\mathcal{Z}(\text{Vec}^{\omega}_{\mathbb{Z}_2})$ reduced topological order: the corresponding folded Lagrangian algebra can be found in (72)

$$
\begin{aligned}
\mathcal{L}_8 = {} & 1 \oplus e_{RG} \oplus e_{GB} \oplus e_{RB} \\
& \oplus s\bar{s}\left[e_R \oplus e_G \oplus e_{RGB} \oplus e_B\right] \\
& \oplus 2s\bar{s}_{RGB} \oplus 2\bar{s}s_{RGB} \\
& \in \mathcal{Z}(\text{Rep}(D_8)) \boxtimes \overline{\mathcal{Z}(\text{Vec}^{\omega}_{\mathbb{Z}_2})} ,
\end{aligned}
\tag{B.11}
$$

which one can also obtain by condensing $e^2 m^2$ in (B.3), (B.4) or (B.5). For the remaining 18 condensable algebras, the reduced topological order will be $\mathcal{Z}(\text{Vec}_{\mathbb{Z}_2})$. One can read off the possible 4d condensable algebras by condensing any bosonic anyon in (B.7), (B.8), (B.9) or (B.10).

An example of a reduction to $\mathcal{Z}(\text{Vec}_{\mathbb{Z}_2})$ can be seen by taking (B.7) and condensing $e_1 m_2$, in that case one finds the following folded algebra

$$
\begin{aligned}
\mathcal{L}_{10} = {} & 1 \oplus e_R \oplus m_G \oplus [m_B \oplus m_{GB}]\bar{e} \\
& \oplus [e_B \oplus e_{RB} \oplus m_G]\bar{m} \oplus [f_B \oplus f_{GB}]\overline{em} ,
\end{aligned}
\tag{B.12}
$$

from the algebra

$$
\begin{aligned}
\mathcal{L}_4 = {} & 1 \oplus e_R \oplus [e_{RGB} \oplus e_{GB}]\bar{e}_1\bar{e}_2 \\
& \oplus [e_{RG} \oplus e_G]\overline{m}_1\overline{m}_2 \oplus [e_B \oplus e_{RB}]\bar{e}_1\bar{e}_2\overline{m}_1\overline{m}_2 \\
& \oplus m_B\overline{m}_2 \oplus m_B\overline{m}_1 \oplus m_{GB}\bar{e}_1 \\
& \oplus m_{GB}\bar{e}_2 \oplus m_G\bar{e}_1\overline{m}_2 \oplus m_G\bar{e}_2\overline{m}_1 \\
& \oplus f_B\bar{e}_1\bar{e}_2\overline{m}_2 \oplus f_B\bar{e}_1\bar{e}_2\overline{m}_1 \oplus f_{GB}\bar{e}_1\overline{m}_1\overline{m}_2 \\
& \oplus f_{GB}\bar{e}_2\overline{m}_1\overline{m}_2 \oplus f_G\bar{e}_2\overline{m}_2 \oplus f_G\bar{e}_1\overline{m}_1 ,
\end{aligned}
\tag{B.13}
$$

where we have identified

$$
\begin{aligned}
1 &\leftrightarrow e_1 m_2 \to 1 , \\
e_2 m_1 &\leftrightarrow e_1 e_2 m_1 m_2 \to m , \\
e_1 &\leftrightarrow m_2 \to e , \\
e_1 e_2 m_1 &\leftrightarrow e_2 m_1 m_2 \to em .
\end{aligned}
\tag{B.14}
$$

Other elements do not braid trivially with $(1 \oplus e_1 m_2)$ and are thus discarded. One can then clearly see that condensing $e_1 m_2$ in (B.7) reduces to (B.12) and the corresponding 4-dimensional condensable algebra is $\mathcal{A}_{11}$, describing a $\mathcal{Z}(\text{Vec}_{\mathbb{Z}_2})$ topological order.

For completeness, we include here the folded algebras for all the 4d condensable algebras that reduce to $\mathcal{Z}(\text{Vec}_{\mathbb{Z}_2})$. The folded Lagrangian algebras for 4d condensable algebras that produce non-intrinsic gSPTs are

$$
\begin{aligned}
\mathcal{L}_{10} = {} & 1 \oplus e_R \oplus m_G \oplus [m_B \oplus m_{GB}]\bar{e} \\
& \oplus [e_B \oplus e_{RB} \oplus m_G]\bar{m} \oplus [f_B \oplus f_{GB}]\overline{em} ,
\end{aligned}
\tag{B.15}
$$

$$\mathcal{L}_{11} = 1 \oplus e_R \oplus m_B \oplus [m_G \oplus m_{GB}]\overline{e}$$
$$\oplus [e_G \oplus e_{RG} \oplus m_B]\overline{m} \oplus [f_G \oplus f_{GB}]\overline{em}, \tag{B.16}$$

$$\mathcal{L}_{13} = 1 \oplus e_G \oplus m_R \oplus [m_B \oplus m_{RB}]\overline{e}$$
$$\oplus [e_B \oplus e_{GB} \oplus m_R]\overline{m} \oplus [f_B \oplus f_{RB}]\overline{em}, \tag{B.17}$$

$$\mathcal{L}_{14} = 1 \oplus e_G \oplus m_B \oplus [m_R \oplus m_{RB}]\overline{e}$$
$$\oplus [e_R \oplus e_{RG} \oplus m_B]\overline{m} \oplus [f_R \oplus f_{RB}]\overline{em}, \tag{B.18}$$

$$\mathcal{L}_{16} = 1 \oplus e_B \oplus m_R \oplus [m_G \oplus m_{RG}]\overline{e}$$
$$\oplus [e_G \oplus e_{GB} \oplus m_R]\overline{m} \oplus [f_G \oplus f_{RG}]\overline{em}, \tag{B.19}$$

$$\mathcal{L}_{17} = 1 \oplus e_B \oplus m_G \oplus [m_R \oplus m_{RG}]\overline{e}$$
$$\oplus [e_R \oplus e_{RB} \oplus m_G]\overline{m} \oplus [f_R \oplus f_{RG}]\overline{em}, \tag{B.20}$$

$$\mathcal{L}_{21} = 1 \oplus e_G \oplus e_R \oplus e_{RG} \oplus 2m_B\overline{m}$$
$$\oplus [e_B \oplus e_{GB} \oplus e_{RB} \oplus e_{RGB}]\overline{e} \oplus 2f_B\overline{em}, \tag{B.21}$$

$$\mathcal{L}_{22} = 1 \oplus e_B \oplus e_G \oplus e_{GB} \oplus 2m_R\overline{m}$$
$$\oplus [e_R \oplus e_{RB} \oplus e_{RG} \oplus e_{RGB}]\overline{e} \oplus 2f_R\overline{em}, \tag{B.22}$$

$$\mathcal{L}_{23} = 1 \oplus e_B \oplus e_R \oplus e_{RB} \oplus 2m_G\overline{m}$$
$$\oplus [e_G \oplus e_{GB} \oplus e_{RG} \oplus e_{RGB}]\overline{e} \oplus 2f_G\overline{em}. \tag{B.23}$$

The folded Lagrangian algebras for 4d condensable algebras that produce $n = 2$ gSSBs are

$$\mathcal{L}_9 = 1 \oplus e_R \oplus m_{GB} \oplus [m_B \oplus m_G]\overline{m}$$
$$\oplus [e_{GB} \oplus e_{RGB} \oplus m_{GB}]\overline{e} \oplus [f_B \oplus f_G]\overline{em}, \tag{B.24}$$

$$\mathcal{L}_{12} = 1 \oplus e_G \oplus m_{RB} \oplus [m_B \oplus m_R]\overline{m}$$
$$\oplus [e_{RB} \oplus e_{RGB} \oplus m_{RB}]\overline{e} \oplus [f_B \oplus f_R]\overline{em}, \tag{B.25}$$

$$\mathcal{L}_{15} = 1 \oplus e_B \oplus m_{RG} \oplus [m_G \oplus m_R]\overline{m}$$
$$\oplus [e_{RG} \oplus e_{RGB} \oplus m_{RG}]\overline{e} \oplus [f_G \oplus f_R]\overline{em}, \tag{B.26}$$

$$\mathcal{L}_{24} = 1 \oplus e_{GB} \oplus e_R \oplus e_{RGB} \oplus 2m_{GB}\overline{e}$$
$$\oplus [e_B \oplus e_G \oplus e_{RB} \oplus e_{RG}]\overline{m} \oplus 2f_{GB}\overline{em}, \tag{B.27}$$

$$\mathcal{L}_{25} = 1 \oplus e_G \oplus e_{RB} \oplus e_{RGB} \oplus 2m_{RB}\overline{e}$$
$$\oplus [e_B \oplus e_{GB} \oplus e_R \oplus e_{RG}]\overline{m} \oplus 2f_{RB}\overline{em}, \tag{B.28}$$

$$\mathcal{L}_{26} = 1 \oplus e_B \oplus e_{RG} \oplus e_{RGB} \oplus 2m_{RG}\overline{e}$$
$$\oplus [e_G \oplus e_{GB} \oplus e_R \oplus e_{RB}]\overline{m} \oplus 2f_{RG}\overline{em}. \tag{B.29}$$

The folded Lagrangian algebras for 4d condensable algebras that produce $n = 3$ igSSBs are

$$\mathcal{L}_{18} = 1 \oplus e_{RGB} \oplus m_{RG} \oplus [m_{GB} \oplus m_{RB}]\overline{e}$$
$$\oplus [e_B \oplus e_{RG} \oplus m_{RG}]\overline{m} \oplus [f_{GB} \oplus f_{RB}]\overline{em}, \tag{B.30}$$

$$\mathcal{L}_{19} = 1 \oplus e_{RGB} \oplus m_{GB} \oplus [m_{RB} \oplus m_{RG}]\overline{e}$$
$$\oplus [e_{GB} \oplus e_R \oplus m_{GB}]\overline{m} \oplus [f_{RB} \oplus f_{RG}]\overline{em}, \tag{B.31}$$

$$\mathcal{L}_{20} = 1 \oplus e_{RGB} \oplus m_{RB} \oplus [m_{GB} \oplus m_{RG}]\overline{e}$$
$$\oplus [e_G \oplus e_{RB} \oplus m_{RB}]\overline{m} \oplus [f_{GB} \oplus f_{RG}]\overline{em}. \tag{B.32}$$

**Note on condensable algebras.** The algebras (B.33)-(B.38) listed below obey the necessary conditions for condensable algebras summarized in Appendix B of [10] (see also [8] and [160])

but do not give rise to reduced topological order. Those conditions are indeed necessary but not sufficient (as was remarked in [10]) and (B.33)-(B.38) are concrete examples of this,

$$1 \oplus e_{RB} \oplus m_{RB} \,, \tag{B.33}$$

$$1 \oplus e_{RB} \oplus m_{G} \,, \tag{B.34}$$

$$1 \oplus e_{GB} \oplus m_{R} \,, \tag{B.35}$$

$$1 \oplus e_{GB} \oplus m_{GB} \,, \tag{B.36}$$

$$1 \oplus e_{RG} \oplus m_{RG} \,, \tag{B.37}$$

$$1 \oplus e_{RG} \oplus m_{B} \,. \tag{B.38}$$

# C  Phases for $\mathsf{Rep}(D_8)$

By fixing the symmetry boundary to be $\mathcal{L}^{\text{sym}}_{\text{Rep}(D_8)}$ in (B.1) we can study all the various phases one may get by choosing different condensable algebras from the list we determined in Table 3. This process is analogous to studying different gapped phases by choosing different Lagrangian algebras for the physical boundary of the SymTFT sandwich. The major difference is these condensable algebras are in general not maximal like their Lagrangian counterparts and thus may lead to various gapless phases as well, as described in the main text. We will follow similar procedures and use related techniques to those described in [10].

In all of the cases below we shall use the notation for $\mathcal{S} = \mathsf{Rep}(D_8)$ adopted in the main text around (28), i.e. we label the symmetry generators as

$$\mathcal{S} = \mathsf{Rep}(D_8) = \{1, R, G, RG, B\} \,, \tag{C.1}$$

where $1$, $R$, $G$, $RG$ are the 1d irreducible representations, and $B$ is the 2d irreducible representation.

## C.1  Condensable algebras of dimension 2

### C.1.1  gSPT 1

By taking the condensable algebra $\mathcal{A}_1$, (and analogously $\mathcal{A}_2$ or $\mathcal{A}_3$), one finds the reduced topological order to be that of $\mathcal{Z}(\mathsf{Vec}_{\mathbb{Z}_4})$. Furthermore, as found in the previous section, these algebras define gSPT phases, and as such no new non-trivial local operators are introduced after compactification with $\mathcal{L}^{\text{sym}}_{\text{Rep}(D_8)}$. Thus colliding the interface with the symmetry boundary $\mathsf{B}^{\text{sym}}_{\text{Rep}(D_8)}$ produces an irreducible boundary $\mathfrak{B}'$. In fact, by looking at the completed algebras in (B.3), (B.4), and (B.5), one can identify $\mathfrak{B}'$ as an irreducible topological boundary condition lying either in the deformation class $[\mathfrak{B}](\mathcal{L}_e)$ (or $[\mathfrak{B}](\mathcal{L}_m)$) associated either to the Lagrangian algebra

$$\mathcal{L}_e = 1 \oplus e \oplus e^2 \oplus e^3 \in \mathcal{Z}(\mathsf{Vec}_{\mathbb{Z}_4}) \,, \qquad \text{or} \qquad \mathcal{L}_m = 1 \oplus m \oplus m^2 \oplus m^3 \in \mathcal{Z}(\mathsf{Vec}_{\mathbb{Z}_4}) \,. \tag{C.2}$$

The unbroken symmetry in this case can be described by the simple topological lines

$$\mathcal{S}' = \mathsf{Vec}_{\mathbb{Z}_4} = \{1, P, P^2, P^3\} \,, \tag{C.3}$$

where $1$ is the identity line on $\mathfrak{B}'$ and $P$ generates the $\mathbb{Z}_4$ symmetry of $\mathfrak{B}'$.

The club quiche picture in this case becomes

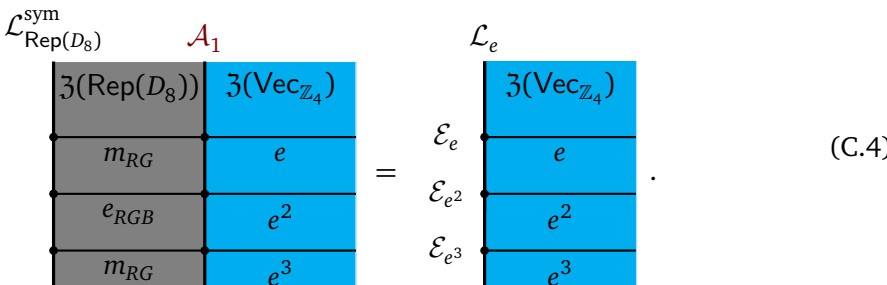

$$(C.4)$$

From the picture above it becomes evident that the symmetry generators of $\text{Rep}(D_8)$ will be projected down to $\text{Vec}_{\mathbb{Z}_4}$ on $\mathfrak{B}'$ as

$$
\begin{aligned}
\phi(1) &= 1, \\
\phi(R) &= P^2, \\
\phi(G) &= P^2, \\
\phi(RG) &= 1, \\
\phi(B) &= P \oplus P^3,
\end{aligned}
\tag{C.5}
$$

as both charges associated with $m_{RG}$ and $e_{RGB}$ are left invariant by the action of $RG$ but transform by sign $-1$ under $R$ and $G$.

This non-trivial homomorphism indeed ensures that the $\text{Rep}(D_8)$ fusion rules are still respected, especially

$$
\begin{aligned}
\phi(B)^2 &= 2(1 \oplus P^2) \\
&= \phi(1) \oplus \phi(R) \oplus \phi(G) \oplus \phi(RG) \\
&= \phi(B^2).
\end{aligned}
\tag{C.6}
$$

Mathematically, we have thus provided a pivotal tensor functor that describes a gSPT phase,

$$
\phi : \text{Rep}(D_8) \to \text{Vec}_{\mathbb{Z}_4},
\tag{C.7}
$$

where $\text{Vec}_{\mathbb{Z}_4}$ is the fusion category formed by topological lines living on the boundary $\mathfrak{B}'$.

### C.1.2 gSPT 2

By taking the condensable algebra $\mathcal{A}_4$, (and analogously $\mathcal{A}_5$ or $\mathcal{A}_6$), one finds the reduced topological order to be that of $\mathcal{Z}(\text{Vec}_{\mathbb{Z}_2 \times \mathbb{Z}_2})$. Furthermore, as determined in the previous section, these algebras also define gSPT phases, and as such no new non-trivial local operators are introduced after compactification with $\mathcal{L}^{\text{sym}}_{\text{Rep}(D_8)}$. Colliding the interface with the symmetry boundary $\text{B}^{\text{sym}}_{\text{Rep}(D_8)}$ produces an irreducible boundary $\mathfrak{B}'$ which is now different from the $\mathbb{Z}_4$ case above. From the completed algebras in (B.7), (B.8), and (B.9), one can identify $\mathfrak{B}'$ as an irreducible topological boundary condition lying in the deformation class $[\mathfrak{B}](\mathcal{L}_e)$ associated to the Lagrangian algebra

$$
\mathcal{L}_e = 1 \oplus e_1 \oplus e_2 \oplus e_1 e_2 \in \mathcal{Z}(\text{Vec}_{\mathbb{Z}_2 \times \mathbb{Z}_2}),
\tag{C.8}
$$

up to automorphisms - e.g. one can exchange $e_2 \to m_2$ or $e_2 \to e_1 m_2$ etc.

The unbroken symmetry in this case can be described by the simple topological lines

$$
\mathcal{S}' = \text{Vec}_{\mathbb{Z}_2 \times \mathbb{Z}_2} = \{1, P_1, P_2, P_1 P_2\},
\tag{C.9}
$$

where 1 is the identity line on $\mathfrak{B}'$ while $P_1$ and $P_2$ generate $\mathbb{Z}_2$ subsymmetries of $\mathbb{Z}_2 \times \mathbb{Z}_2$ on $\mathfrak{B}'$.

The club quiche can then be represented as

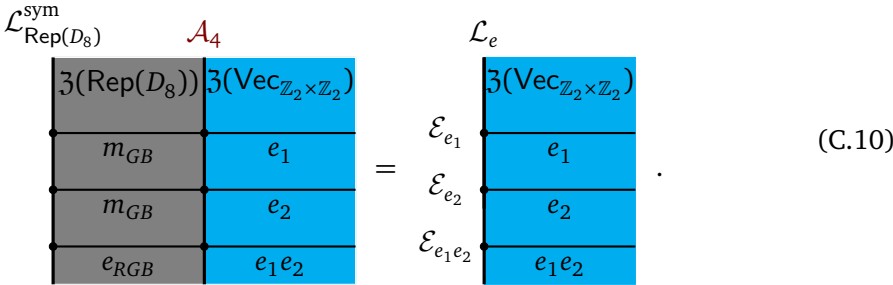

$$\tag{C.10}$$

It is again not hard to see that the symmetry generators of $\mathrm{Rep}(D_8)$ will be projected down to $\mathrm{Vec}_{\mathbb{Z}_2 \times \mathbb{Z}_2}$ on $\mathfrak{B}'$ as

$$\begin{aligned}
\phi(1) &= 1\,, \\
\phi(R) &= 1\,, \\
\phi(G) &= P_1 P_2\,, \\
\phi(RG) &= P_1 P_2\,, \\
\phi(B) &= P_1 \oplus P_2\,,
\end{aligned} \tag{C.11}$$

as both charges associated with $m_{GB}$ and $e_{RGB}$ are left invariant by the action of $R$ but transform by sign $-1$ under $G$ and $RG$.

This non-trivial homomorphism indeed ensures that the $\mathrm{Rep}(D_8)$ fusion rules are still respected, especially

$$\begin{aligned}
\phi(B)^2 &= 2(1 \oplus P_1 P_2) \\
&= \phi(1) \oplus \phi(R) \oplus \phi(G) \oplus \phi(RG) \\
&= \phi(B^2)\,.
\end{aligned} \tag{C.12}$$

Mathematically, we have thus provided another pivotal tensor functor that describes a gSPT phase, in this case

$$\phi : \ \mathrm{Rep}(D_8) \to \mathrm{Vec}_{\mathbb{Z}_2 \times \mathbb{Z}_2}\,, \tag{C.13}$$

where $\mathrm{Vec}_{\mathbb{Z}_2 \times \mathbb{Z}_2}$ is the fusion category formed by topological lines living on the boundary $\mathfrak{B}'$.

### C.1.3 $\mathrm{Rep}(D_8)/(\mathbb{Z}_2 \times \mathbb{Z}_2)$ gSSB

Unlike for all the other 2d condensable algebras, $\mathcal{A}_7$ has a non-trivial intersection with $\mathcal{L}^{\mathrm{sym}}_{\mathrm{Rep}(D_8)}$, specifically the anyon $e_{RGB}$. This in turn means that collapsing the interface $\mathcal{A}_7$ with the symmetry boundary will produce a reducible boundary condition $\mathfrak{B}'$ for the reduced topological center $\mathcal{Z}(\mathrm{Vec}_{\mathbb{Z}_2 \times \mathbb{Z}_2})$. From the completed algebra (B.10) one can deduce that the reducible boundary $\mathfrak{B}'$ will be of the form

$$\mathfrak{B}' = \mathfrak{B}_e \oplus \mathfrak{B}_m\,, \tag{C.14}$$

where $\mathfrak{B}_e$ is an irreducible topological boundary with associated Lagrangian algebra

$$\mathcal{L}_e = 1 \oplus e_1 \oplus e_2 \oplus e_1 e_2\,, \tag{C.15}$$

and $\mathfrak{B}_m$ is an irreducible topological boundary with associated Lagrangian algebra

$$\mathcal{L}_m = 1 \oplus m_1 \oplus m_2 \oplus m_1 m_2\,. \tag{C.16}$$

We can thus see that the resulting gSSB will have $n = 2$ universes which was to be expected from considerations about the dimensions. Also note that generally, there could be a relative

Euler term between $\mathfrak{B}_e$ and $\mathfrak{B}_m$ which captures different linking actions of the symmetry generators, however, in this case, it will be trivial.

The topological line operators on $\mathfrak{B}'$ are

$$1_{ii}, \quad (P_1)_{ii}, \quad (P_2)_{ii}, \quad (P_1 P_2)_{ii}, \quad S_{ij}, \tag{C.17}$$

where $i, j \in \{e, m\}$ and $i \neq j$. The line $1_{ii}$ is the identity line on each boundary $\mathfrak{B}_i$, the lines $(P_1)_{ii}$, $(P_2)_{ii}$ and $(P_1 P_2)_{ii} \equiv (P_1)_{ii}(P_2)_{ii}$ are the $\mathbb{Z}_2$ generators of the $\mathbb{Z}_2 \times \mathbb{Z}_2$ symmetry on the boundary $\mathfrak{B}_i$, whereas the line $S_{ij}$ changes the boundary $\mathfrak{B}_i$ to the boundary $\mathfrak{B}_j$, with fusion rules

$$
\begin{aligned}
(P_k)_{ii} \otimes S_{ij} &= S_{ij}(P_k)_{jj} = S_{ij}, \\
S_{ij} \otimes S_{ji} &= 1_{ii} \oplus (P_1)_{ii} \oplus (P_2)_{ii} \oplus (P_1 P_2)_{ii},
\end{aligned}
\tag{C.18}
$$

where all $P$'s are of dimension 1 while $S_{em}$ and $S_{me}$ are of dimension 2. One can recognize these are also precisely the fusion rules of $\mathrm{Rep}(D_8)$ which is also equivalent to the Tambara-Yamagami category $\mathrm{TY}(\mathbb{Z}_2 \times \mathbb{Z}_2)$.

With these considerations we can now draw the club quiche picture as

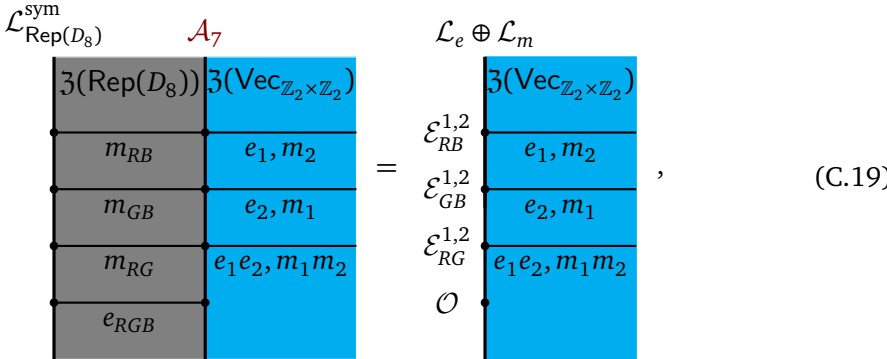

$$\tag{C.19}$$

where the commas signify the fact that e.g. the line $m_{RB}$ splits at $\mathcal{A}_7$ into $e_1$ and $m_2$.

By considering the local operators that arise after the compactification one can construct the vacua/identity operators of each universe and then study the linking actions on them by the $\mathrm{Rep}(D_8)$ symmetry generators. One then finds that lines realizing the $\mathrm{Rep}(D_8)$ symmetry on $\mathfrak{B}'$ are

$$
\begin{aligned}
\phi(1) &= 1_{ee} \oplus 1_{mm}, \\
\phi(R) &= (P_1)_{ee} \oplus (P_2)_{mm}, \\
\phi(G) &= (P_2)_{ee} \oplus (P_1)_{mm}, \\
\phi(RG) &= (P_1 P_2)_{ee} \oplus (P_1 P_2)_{mm}, \\
\phi(B) &= S_{em} \oplus S_{me}.
\end{aligned}
\tag{C.20}
$$

One can also verify that these are indeed consistent with the $\mathrm{Rep}(D_8)$ fusion rules, particularly that

$$
\begin{aligned}
\phi(\rho)\phi(\rho') &= \phi(\rho \circ \rho'), \\
\phi(\rho)\phi(B) &= \phi(B), \\
\phi(B^2) &= \phi(B)^2,
\end{aligned}
\tag{C.21}
$$

for any 1d representations $\rho, \rho' \in \{1, R, G, RG\} \subset \mathrm{Rep}(D_8)$. One can see that the two vacua stay invariant under the $\mathbb{Z}_2 \times \mathbb{Z}_2$ subsymmetry of $\mathrm{Rep}(D_8)$, but change under the generator $B$, thus we refer to this phase as a $\mathrm{Rep}(D_8)/(\mathbb{Z}_2 \times \mathbb{Z}_2)$ gSSB phase.

Mathematically, we have thus provided information about this $\mathrm{Rep}(D_8)/(\mathbb{Z}_2 \times \mathbb{Z}_2)$ gSSB by defining a pivotal tensor functor

$$\phi : \ \mathrm{Rep}(D_8) \to \mathrm{Rep}(D_8)^1_{2 \times 2}, \tag{C.22}$$

where $\text{Rep}(D_8)_{2\times 2}^{e^{-\lambda}=1}$ is the pivotal multi-fusion category formed by $2 \times 2$ matrices in $\text{Rep}(D_8)$ which is also the category formed by topological lines living on the underlying boundary $\mathfrak{B}'$ (with a trivial relative Euler term $e^{-\lambda} = 1$).

## C.2 Condensable algebras of dimension 4

### C.2.1 gSPT 1

By taking the condensable algebra $\mathcal{A}_{10}$, (and analogously $\mathcal{A}_{11}$, $\mathcal{A}_{13}$, $\mathcal{A}_{14}$, $\mathcal{A}_{16}$, or $\mathcal{A}_{17}$), one finds the reduced topological order to be that of $\mathcal{Z}(\text{Vec}_{\mathbb{Z}_2})$. Furthermore, as found in the previous section, these algebras also define gSPT phases, and as such no new non-trivial local operators are introduced after compactification with $\mathcal{L}_{\text{Rep}(D_8)}^{\text{sym}}$. Colliding the interface with the symmetry boundary $B_{\text{Rep}(D_8)}^{\text{sym}}$ produces an irreducible boundary $\mathfrak{B}'$ which is now $\mathbb{Z}_2$-symmetric.

From the completed algebras in (B.15)-(B.23), one can identify $\mathfrak{B}'$ as an irreducible topological boundary condition lying either in the deformation class $[\mathfrak{B}](\mathcal{L}_e)$ (or $[\mathfrak{B}](\mathcal{L}_m)$) associated either to the Lagrangian algebra

$$\mathcal{L}_e = 1 \oplus e \in \mathcal{Z}(\text{Vec}_{\mathbb{Z}_2}),$$
$$\text{or} \tag{C.23}$$
$$\mathcal{L}_m = 1 \oplus m \in \mathcal{Z}(\text{Vec}_{\mathbb{Z}_2}).$$

The unbroken symmetry in this case can be described by the simple topological lines

$$\mathcal{S}' = \text{Vec}_{\mathbb{Z}_2} = \{1, P\}, \tag{C.24}$$

where 1 is the identity line on $\mathfrak{B}'$ and $P$ generates $\mathbb{Z}_2$ symmetry of $\mathfrak{B}'$.

The club quiche picture in this case becomes

$$
\begin{array}{c}
\mathcal{L}_{\text{Rep}(D_8)}^{\text{sym}} \qquad \mathcal{A}_{10} \qquad\qquad\qquad \mathcal{L}_e \\[4pt]
\boxed{\begin{array}{c|c} \mathfrak{Z}(\text{Rep}(D_8)) & \mathfrak{Z}(\text{Vec}_{\mathbb{Z}_2}) \\ \hline m_{GB} & e \end{array}} \quad = \mathcal{E}_e \quad \boxed{\begin{array}{c} \mathfrak{Z}(\text{Vec}_{\mathbb{Z}_2}) \\ \hline e \end{array}} .
\end{array}
\tag{C.25}
$$

It is again not hard to see that the symmetry generators of $\text{Rep}(D_8)$ will be projected down to just $\text{Vec}_{\mathbb{Z}_2}$ on $\mathfrak{B}'$ as

$$\phi(1) = 1,$$
$$\phi(R) = 1,$$
$$\phi(G) = P, \tag{C.26}$$
$$\phi(RG) = P,$$
$$\phi(B) = 1 \oplus P,$$

as the charges associated with $m_{GB}$ multiplet are left invariant by the action of $R$ but transform by sign $-1$ under $G$ and $RG$. Note that by starting with a different algebra where the role of $m_{GB}$ was played by e.g. $m_{RG}$, eq. (C.26) needs to be modified accordingly.

This non-trivial homomorphism indeed ensures that the $\text{Rep}(D_8)$ fusion rules are still respected, especially

$$\phi(B)^2 = 2(1 \oplus P)$$
$$= \phi(1) \oplus \phi(R) \oplus \phi(G) \oplus \phi(RG) \tag{C.27}$$
$$= \phi(B^2).$$

Mathematically, we have thus provided a pivotal tensor functor that describes a gSPT phase,

$$\phi : \text{Rep}(D_8) \to \text{Vec}_{\mathbb{Z}_2} \,, \tag{C.28}$$

where $\text{Vec}_{\mathbb{Z}_2}$ is the fusion category formed by topological lines living on the boundary $\mathfrak{B}'$.

### C.2.2   gSPT 2

One can also find a gSPT phase by taking the condensable algebra $\mathcal{A}_{21}$ (and analogously $\mathcal{A}_{22}$, or $\mathcal{A}_{23}$) which again produces a reduced topological order $\mathcal{Z}(\text{Vec}_{\mathbb{Z}_2})$ but with a different functor. This is the case as this time it is not one of the $m$'s that go on as $e$ or $m$ after passing the interface but it is $e_{RGB}$ instead. The picture will thus be very similar but now all functor images of 1d symmetry generators, $R$, $G$ and $RG$ will be trivial as $e_{RGB}$ is uncharged under these generators. On the other hand, $B$ should act by linking on the $e_{RGB}$ charge by a factor of $-2$.

The club quiche picture is now instead

$$\tag{C.29}$$

The $\text{Rep}(D_8)$ symmetry will then be projected down to $\text{Vec}_{\mathbb{Z}_2}$ on $\mathfrak{B}'$ differently than in (C.26), specifically

$$\begin{aligned}
\phi(1) &= 1 \,, \\
\phi(R) &= 1 \,, \\
\phi(G) &= 1 \,, \\
\phi(RG) &= 1 \,, \\
\phi(B) &= P \oplus P \,.
\end{aligned} \tag{C.30}$$

One can then check these fusion rules are again consistent and thus we have provided a different pivotal tensor functor that also describes a gSPT phase,

$$\phi : \text{Rep}(D_8) \to \text{Vec}_{\mathbb{Z}_2} \,, \tag{C.31}$$

where $\text{Vec}_{\mathbb{Z}_2}$ is the fusion category formed by topological lines living on the boundary $\mathfrak{B}'$.

### C.2.3   igSPT

As mentioned in the main text, one can find an igSPT phase with $\text{Rep}(D_8)$ symmetry by choosing the condensable algebra to be $\mathcal{A}_8$. This case is quite similar to the gSPT 2 case just above as it involves the line $e_{RGB}$ which is invariant under $R$, $G$, $RG$. The main difference lies in the fact that the reduced center is the double semion model, which is the SymTFT $\mathfrak{Z}(\text{Vec}_{\mathbb{Z}_2}^\omega)$ with $\mathbb{Z}_2$ symmetry and a non-trivial 't Hooft anomaly described by

$$\omega \neq 0 \in H^3(\mathbb{Z}_2, U(1)) = \mathbb{Z}_2 \,. \tag{C.32}$$

As collapsing $\mathcal{A}_8$ does not produce new non-trivial topological local operators, the resulting boundary $\mathfrak{B}'$ will be irreducible with an associated Lagrangian algebra $\mathcal{L}_{s\bar{s}} = 1 \oplus s\bar{s}$. The topological lines living on $\mathfrak{B}'$ will be

$$1 \,, \quad P' \,, \tag{C.33}$$

where $P'$ generates the anomalous $\mathbb{Z}_2$ symmetry.

The club quiche picture is then the following

$$
\begin{array}{ccc}
\mathcal{L}^{\text{sym}}_{\text{Rep}(D_8)} & \mathcal{A}_8 & \mathcal{L}_{s\bar{s}} \\
\end{array}
\quad = \mathcal{E}_{s\bar{s}} \quad . \tag{C.34}
$$

The $\text{Rep}(D_8)$ symmetry will then be projected down to $\text{Vec}^{\omega}_{\mathbb{Z}_2}$ on $\mathfrak{B}'$ similarly to (C.30), specifically

$$
\begin{aligned}
\phi(1) &= 1, \\
\phi(R) &= 1, \\
\phi(G) &= 1, \\
\phi(RG) &= 1, \\
\phi(B) &= P' \oplus P',
\end{aligned} \tag{C.35}
$$

however, to fully specify these homomorphisms, one also needs to pick a consistent set of local junction operators as in [10] which encodes the anomalous symmetry generated by $P'$. Having done so, then one has found a pivotal tensor functor that describes the igSPT phase,

$$
\phi : \text{Rep}(D_8) \to \text{Vec}^{\omega}_{\mathbb{Z}_2}, \tag{C.36}
$$

where $\text{Vec}^{\omega}_{\mathbb{Z}_2}$ is the fusion category formed by topological lines living on the boundary $\mathfrak{B}'$.

### C.2.4 $\mathbb{Z}_2$ gSSB

As we have seen above for gSPT phases, we again find two possible variations of the pivotal functor for the $n = 2$ gSSB phases here. We first start by taking the condensable algebra $\mathcal{A}_9$ (and analogously $\mathcal{A}_{12}$, or $\mathcal{A}_{15}$) which again produces a reduced topological order $\mathcal{Z}(\text{Vec}_{\mathbb{Z}_2})$. However, now there is a non-trivial intersection with the symmetry boundary of the line $m_{GB}$ (and analogously $m_{RB}$ or $m_{RG}$), and thus the resulting boundary $\mathfrak{B}'$ will be reducible of the form

$$
\mathfrak{B}' = \mathfrak{B}^e_0 \oplus \mathfrak{B}^e_1, \tag{C.37}
$$

where $\mathfrak{B}^e_i$ is an irreducible topological boundary condition lying in the deformation class $[\mathfrak{B}](\mathcal{L}_e)$ associated to Lagrangian algebra $\mathcal{L}_e = 1 \oplus e$ (where we can again exchange $e \leftrightarrow m$).

The topological line operators on $\mathfrak{B}'$ are

$$
1_{ij}, \quad (P)_{ij}, \tag{C.38}
$$

where $i, j \in \{0, 1\}$. The line $1_{ii}$ is the identity line on each boundary $\mathfrak{B}_i$, the lines $P_{ii}$ are the $\mathbb{Z}_2$ generators on the boundary $\mathfrak{B}_i$, whereas the line $1_{ij}$ for $i \neq j$ changes the boundary $\mathfrak{B}_i$ to the boundary $\mathfrak{B}_j$, with fusion rules

$$
1_{ij} \otimes 1_{jk} = 1_{ik}, \tag{C.39}
$$

and $P_{ij}$ are obtained by fusing

$$
P_{ij} = P_{ii} \otimes 1_{ij} = 1_{ij} \otimes P_{jj}, \tag{C.40}
$$

where all $1_{ij}$ and $P_{ij}$ are of dimension 1 as again one finds that there is no non-trivial relative Euler term.

The club quiche picture in this case is

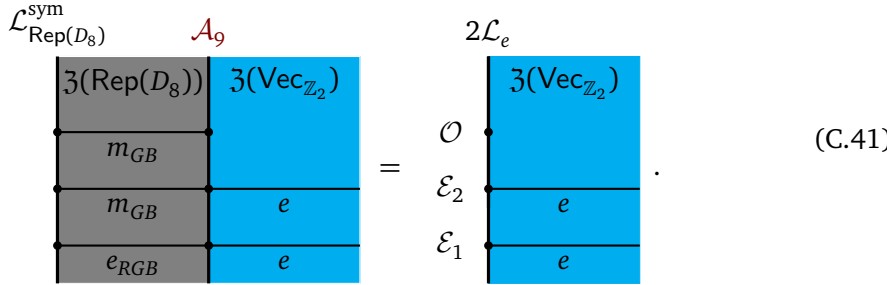

$$(C.41)$$

The products of these operators can be determined to be

$$
\begin{aligned}
&\mathcal{O}^2 = 1, && \mathcal{O}\mathcal{E}_1 = \mathcal{E}_2, && \mathcal{O}\mathcal{E}_2 = \mathcal{E}_1, \\
&\mathcal{E}_1\mathcal{E}_1 = 1, && \mathcal{E}_2\mathcal{E}_2 = 1, && \mathcal{E}_1\mathcal{E}_2 = \mathcal{O},
\end{aligned}
\tag{C.42}
$$

From these we can construct the operators

$$
\begin{aligned}
v_i &= \frac{1 + (-1)^i \mathcal{O}}{2}, \\
\widehat{\mathcal{E}}_i &= \frac{\mathcal{E}_1 + (-1)^i \mathcal{E}_2}{2} = \mathcal{E}_1 v_i,
\end{aligned}
\tag{C.43}
$$

with $i \in \{0, 1\}$ where $v_i$ are the identity local operators on $\mathfrak{B}_i^e$, and $\widehat{\mathcal{E}}_i$ are the ends of $e$ along $\mathfrak{B}_i^e$ which can be seen as

$$
v_i v_j = \delta_{ij} v_j, \qquad v_i \widehat{\mathcal{E}}_j = \delta_{ij} \widehat{\mathcal{E}}_j, \qquad \widehat{\mathcal{E}}_i \widehat{\mathcal{E}}_j = \delta_{ij} v_j.
\tag{C.44}
$$

By looking at the linking action of the $\mathrm{Rep}(D_8)$ generators, one can again deduce their images on the reducible boundary $\mathfrak{B}'$ as

$$
\begin{aligned}
\phi(1) &= 1_{00} \oplus 1_{11}, \\
\phi(R) &= 1_{00} \oplus 1_{11}, \\
\phi(G) &= 1_{01} \oplus 1_{10}, \\
\phi(RG) &= 1_{01} \oplus 1_{10}, \\
\phi(B) &= P_{00} \oplus P_{01} \oplus P_{10} \oplus P_{11},
\end{aligned}
\tag{C.45}
$$

which is again consistent with $\mathrm{Rep}(D_8)$ fusion rules. One can see there is a broken $\mathbb{Z}_2$ subsymmetry that exchanges vacua $v_0 \leftrightarrow v_1$, hence this phase can be described as a $\mathbb{Z}_2$ gSSB phase.

Mathematically, we have provided information on a pivotal tensor functor describing a $\mathbb{Z}_2$ gSSB phase

$$
\phi : \mathrm{Rep}(D_8) \to \mathrm{Mat}_2(\mathrm{Vec}_{\mathbb{Z}_2}),
\tag{C.46}
$$

where $\mathrm{Mat}_2(\mathrm{Vec}_{\mathbb{Z}_2})$ is the multi-fusion category formed by $2 \times 2$ matrices in $\mathrm{Vec}_{\mathbb{Z}_2}$, which is the category formed by topological line operators living on the boundary $\mathfrak{B}'$.

### C.2.5 $\mathrm{Rep}(D_8)/(\mathbb{Z}_2 \times \mathbb{Z}_2)$ gSSB

By instead starting with the condensable algebra $\mathcal{A}_{24}$ (and analogously $\mathcal{A}_{25}$, or $\mathcal{A}_{26}$) one finds the other pivotal functor that describes an $n = 2$ gSSB. The analysis is very similar, the only

major difference now is that the intersection of $\mathcal{A}_{24}$ with the symmetry boundary is the line $e_{RGB}$. This then changes the club quiche picture to be

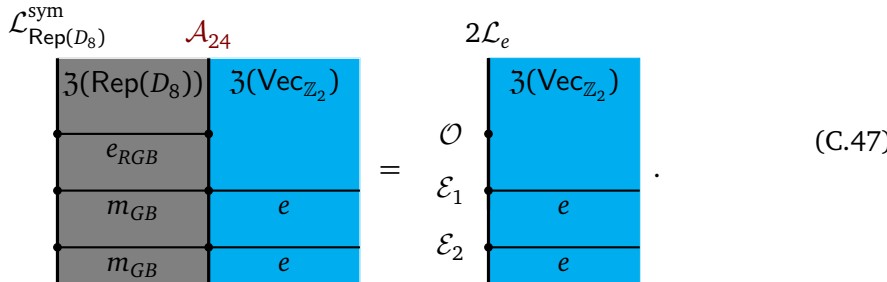

$$\tag{C.47}$$

The products of these operators can be determined to be

$$
\begin{aligned}
\mathcal{O}^2 &= 1, & \mathcal{O}\mathcal{E}_1 &= \mathcal{E}_2, & \mathcal{O}\mathcal{E}_2 &= \mathcal{E}_1, \\
\mathcal{E}_1\mathcal{E}_1 &= \mathcal{O}, & \mathcal{E}_2\mathcal{E}_2 &= \mathcal{O}, & \mathcal{E}_1\mathcal{E}_2 &= 1.
\end{aligned}
\tag{C.48}
$$

From these we can again construct the operators

$$
\begin{aligned}
v_0 &= \frac{1+\mathcal{O}}{2}, & \widehat{\mathcal{E}}_0 &= \frac{\mathcal{E}_1 + \mathcal{E}_2}{2}, \\
v_1 &= \frac{1-\mathcal{O}}{2}, & \widehat{\mathcal{E}}_1 &= i\frac{\mathcal{E}_1 - \mathcal{E}_2}{2},
\end{aligned}
\tag{C.49}
$$

where $v_i$ with $i \in \{0,1\}$ are the identity local operators on $\mathfrak{B}^e_i$, and $\widehat{\mathcal{E}}_i$ are the ends of $e$ along $\mathfrak{B}^e_i$ which can be seen as

$$
v_i v_j = \delta_{ij} v_j, \qquad v_i \widehat{\mathcal{E}}_j = \delta_{ij} \widehat{\mathcal{E}}_j, \qquad \widehat{\mathcal{E}}_i \widehat{\mathcal{E}}_j = \delta_{ij} v_j.
\tag{C.50}
$$

Note that these operators look identical to the $\mathbb{Z}_4$ club quiche operators for algebra $1 \oplus e^2$ in [10]. However, in this case, we find a different symmetry action on these operators as the underlying symmetry is a homomorphism from $\mathrm{Rep}(D_8)$ which does not have a $\mathbb{Z}_4$ subsymmetry. In this case, one instead finds that the (non-trivial) linking action that descends from $\mathrm{Rep}(D_8)$ acts as

$$
\begin{aligned}
G, RG : \quad & \mathcal{E}_i \to -\mathcal{E}_i, & \mathcal{O} \to \mathcal{O}, & \quad 1 \to 1, \\
B : \quad & \mathcal{E}_i \to 0, & \mathcal{O} \to -2\mathcal{O}, & \quad 1 \to 2,
\end{aligned}
\tag{C.51}
$$

implying that we have

$$
\begin{aligned}
G, RG : \quad & v_i \to v_i, & \widehat{\mathcal{E}}_i \to -\widehat{\mathcal{E}}_i, \\
B : \quad & \widehat{\mathcal{E}}_i \to 0, & v_0 \to 2v_1, & \quad v_1 \to 2v_0.
\end{aligned}
\tag{C.52}
$$

One finds that $\phi(G) = \phi(RG)$ act as $P^2$ in the $\mathbb{Z}_4$ center which is also consistent with how the vacua $v_i$ are left invariant under $P^2$. The only non-trivial action of $\mathbb{Z}_2 \times \mathbb{Z}_2$ which survives in the $\mathbb{Z}_4$ is $P_1 P_2$ as it gets mapped to $P^2$.

From this linking action of the $\mathrm{Rep}(D_8)$ generators, one can deduce their images on the reducible boundary $\mathfrak{B}'$ to be

$$
\begin{aligned}
\phi(1) &= 1_{00} \oplus 1_{11}, \\
\phi(R) &= 1_{00} \oplus 1_{11}, \\
\phi(G) &= P_{00} \oplus P_{11}, \\
\phi(RG) &= P_{00} \oplus P_{11}, \\
\phi(B) &= 1_{01} \oplus 1_{10} \oplus P_{01} \oplus P_{10},
\end{aligned}
\tag{C.53}
$$

which is again consistent with $\text{Rep}(D_8)$ fusion rules and a trivial relative Euler term. As the $\mathbb{Z}_2 \times \mathbb{Z}_2$ subsymmetry of $\text{Rep}(D_8)$ is spontaneously preserved in this phase, it can be described as a $\text{Rep}(D_8)/(\mathbb{Z}_2 \times \mathbb{Z}_2)$ gSSB phase.

Mathematically, we have thus provided information on a pivotal tensor functor describing a $\text{Rep}(D_8)/(\mathbb{Z}_2 \times \mathbb{Z}_2)$ gSSB

$$\phi : \text{Rep}(D_8) \to \text{Mat}_2(\text{Vec}_{\mathbb{Z}_2}), \tag{C.54}$$

where $\text{Mat}_2(\text{Vec}_{\mathbb{Z}_2})$ is the multi-fusion category formed by $2 \times 2$ matrices in $\text{Vec}_{\mathbb{Z}_2}$, which is the category formed by topological line operators living on the boundary $\mathfrak{B}'$.

### C.2.6  Ising **igSSB**

Finally, we study the last remaining condensable algebras of dimension 4 which produce $n = 3$ igSSB phases for the $\text{Rep}(D_8)$ symmetric theory which we show to be Ising igSSB phases. This feature is unique to the gapless story of these phases as by further condensing to a Lagrangian algebra for a $\text{Rep}(D_8)$ symmetric theory, one can either end up with a $\mathbb{Z}_2 \times \mathbb{Z}_2$ SSB phase or a $\text{Rep}(D_8)$ SSB phase. The Ising SSB has 3 universes, then further condensing can only increase the number of universes to 4 or 5 respectively which means this is an intrinsic gSSB phase as the number of universes must strictly increase by further condensing.

By taking the condensable algebra $\mathcal{A}_{18}$ (and analogously $\mathcal{A}_{19}$, or $\mathcal{A}_{20}$) one finds two nontrivial intersections, $e_{RGB}$ and $m_{RG}$, and thus 2 non-trivial local operators after collapsing the interval which leads to $n = 3$ universes. The boundary $\mathfrak{B}'$ will be reducible and take form

$$\mathfrak{B}' = \mathfrak{B}^m \oplus \mathfrak{B}_1^e \oplus \mathfrak{B}_2^e, \tag{C.55}$$

where again $\mathfrak{B}_i^e$ and $\mathfrak{B}^m$ are irreducible topological boundaries, $\mathfrak{B}_i^e$ associated to $\mathcal{L}_e = 1 \oplus e$ and $\mathfrak{B}^m$ associated to $\mathcal{L}_m = 1 \oplus m$ (where the roles of $e$ and $m$ may be swapped).

The topological line operators on $\mathfrak{B}'$ form are

$$1_{00}, \quad 1_{ij}, \quad P_{00}, \quad P_{ij}, \quad S_{0i}, \quad S_{i0}, \tag{C.56}$$

where $i, j \in \{1, 2\}$ label the two $e$-universes while $0$ labels the one $m$-universe. The nontrivial lines start with $1_{ij}$ for $i \neq j$ which is a boundary changing operator between the two $e$-universes, $P_{00}$ is the generator of the $\mathbb{Z}_2$ symmetry on $\mathfrak{B}^m$, $P_{ii}$ is the generator of the $\mathbb{Z}_2$ symmetry on $\mathfrak{B}_i^e$, $P_{ij}$ is again obtained by fusion $P_{ij} = P_{ii} 1_{ij} = 1_{ij} P_{jj}$, finally the lines $S_{0i}$ (and $S_{i0}$) change the boundary $\mathfrak{B}^m$ to the boundary $\mathfrak{B}_i^e$ (and vice versa), with fusion rules

$$\begin{aligned}
P_{00} \otimes S_{0i} &= S_{0i}, & P_{ii} \otimes S_{i0} &= S_{i0}, \\
S_{i0} \otimes P_{00} &= S_{i0}, & S_{0i} \otimes P_{ii} &= S_{0i}, \\
S_{0i} \otimes S_{i0} &= 1_{00} \oplus P_{00}, & S_{i0} \otimes S_{0j} &= 1_{ij} \oplus P_{ij},
\end{aligned} \tag{C.57}$$

where the linking actions of $S$-boundary-changing lines are

$$S_{0i} : v_0 \to \sqrt{2} e^{-\lambda} v_i, \qquad S_{i0} : v_i \to \sqrt{2} e^{\lambda} v_0, \tag{C.58}$$

where $\lambda \in \mathbb{R}$ captures the relative Euler term between the boundaries $\mathfrak{B}^m$ and $\mathfrak{B}_i^e$. There could also be a relative Euler term between the two $e$-boundaries coming from $1_{ij}$, however, one finds it is trivial so here we only focus on the one above.

The relative Euler term $e^{-\lambda} = \sqrt{2}$ between $\mathfrak{B}^m$ and $\mathfrak{B}_i^e$ is again a direct consequence of non-invertibility of $\text{Rep}(D_8)$. This relative Euler term also gives rise to the non-trivial pivotal structure on this multi-fusion category which is reflected in quantum dimensions of $S_{0i}$ and $S_{i0}$ being

$$\dim(S_{0i}) = 2, \qquad \dim(S_{i0}) = 1, \tag{C.59}$$

which differ by a factor of $\sqrt{2}$ from their naive quantum dimensions $\dim = \sqrt{2}$ in the standard Ising setup.

With these considerations we can now draw the club quiche picture as

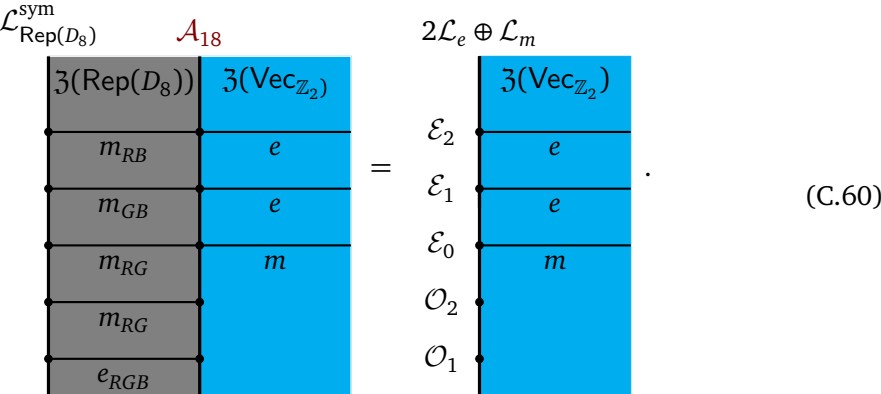

$$(C.60)$$

The operators that arise by this interval collapse have the following fusion rules

$$
\begin{aligned}
\mathcal{O}_1^2 &= 1\,, & \mathcal{O}_2^2 &= (1+\mathcal{O}_1)/2\,, \\
\mathcal{O}_1\mathcal{O}_2 &= \mathcal{O}_2\,, & \mathcal{O}_2\mathcal{E}_0 &= 0\,, \\
\mathcal{O}_1\mathcal{E}_0 &= -\mathcal{E}_0\,, & \mathcal{O}_2\mathcal{E}_1 &= \mathcal{E}_2\,, \\
\mathcal{O}_1\mathcal{E}_i &= \mathcal{E}_i\,, & \mathcal{O}_2\mathcal{E}_2 &= \mathcal{E}_1\,, \\
\mathcal{E}_0^2 &= (1-\mathcal{O}_1)/2\,, & \mathcal{E}_i^2 &= (1+\mathcal{O}_1)/2\,, \\
\mathcal{E}_0\mathcal{E}_i &= 0\,, & \mathcal{E}_1\mathcal{E}_2 &= \mathcal{O}_2\,,
\end{aligned}
\tag{C.61}
$$

for $i \in \{1,2\}$.

From these we can again construct the operators

$$
\begin{aligned}
v_0 &= \frac{1-\mathcal{O}_1}{2}\,, & \widehat{\mathcal{E}}_0 &= \mathcal{E}_0 = \mathcal{E}_0 v_0\,, \\
v_1 &= \frac{1+\mathcal{O}_1+2\mathcal{O}_2}{4}\,, & \widehat{\mathcal{E}}_1 &= \frac{\mathcal{E}_1+\mathcal{E}_2}{2} = \mathcal{E}_1 v_1\,, \\
v_2 &= \frac{1+\mathcal{O}_1-2\mathcal{O}_2}{4}\,, & \widehat{\mathcal{E}}_2 &= \frac{\mathcal{E}_1-\mathcal{E}_2}{2} = \mathcal{E}_1 v_2\,,
\end{aligned}
\tag{C.62}
$$

where $v_0$ is the identity local operator on $\mathfrak{B}^m$ and $v_i$ on $\mathfrak{B}^e_i$, and $\widehat{\mathcal{E}}_0$ is the end $m$ on $\mathfrak{B}^m$ and $\widehat{\mathcal{E}}_i$ are the ends for $e$ on $\mathfrak{B}^e_i$ which can be seen as

$$
v_a v_b = \delta_{ab} v_b\,, \qquad v_a \widehat{\mathcal{E}}_b = \delta_{ab} \widehat{\mathcal{E}}_b\,, \qquad \widetilde{\mathcal{E}}_a \widehat{\mathcal{E}}_b = \delta_{ab} v_b\,,
\tag{C.63}
$$

for $a, b \in \{0,1,2\}$.

By looking at the linking action of the $\text{Rep}(D_8)$ generators, i.e.

$$
\begin{aligned}
R: \quad & v_0 \to v_0\,, & v_1 &\leftrightarrow v_2\,, & & \\
& \widehat{\mathcal{E}}_0 \to -\widehat{\mathcal{E}}_0\,, & \widehat{\mathcal{E}}_1 &\to \widehat{\mathcal{E}}_2\,, & \widehat{\mathcal{E}}_2 &\to \widehat{\mathcal{E}}_1\,, \\
G: \quad & v_0 \to v_0\,, & v_1 &\leftrightarrow v_2\,, & & \\
& \widehat{\mathcal{E}}_0 \to -\widehat{\mathcal{E}}_0\,, & \widehat{\mathcal{E}}_1 &\to -\widehat{\mathcal{E}}_2\,, & \widehat{\mathcal{E}}_2 &\to -\widehat{\mathcal{E}}_1\,, \\
RG: \quad & v_0 \to v_0\,, & v_1 &\to v_1\,, & v_2 &\to v_2\,, \\
& \widehat{\mathcal{E}}_0 \to \widehat{\mathcal{E}}_0\,, & \widehat{\mathcal{E}}_1 &\to -\widehat{\mathcal{E}}_1\,, & \widehat{\mathcal{E}}_2 &\to -\widehat{\mathcal{E}}_2\,, \\
B: \quad & v_0 \to 2(v_1+v_2)\,, & v_1, v_2 &\to v_0\,, & & \\
& \widehat{\mathcal{E}}_0, \widehat{\mathcal{E}}_1, \widehat{\mathcal{E}}_2 \to 0\,, & & & &
\end{aligned}
\tag{C.64}
$$

one can deduce their images on the reducible boundary $\mathfrak{B}'$ to be

$$
\begin{aligned}
\phi(1) &= 1_{00} \oplus 1_{11} \oplus 1_{22}\,, \\
\phi(R) &= P_{00} \oplus 1_{12} \oplus 1_{21}\,, \\
\phi(G) &= P_{00} \oplus P_{12} \oplus P_{21}\,, \\
\phi(RG) &= 1_{00} \oplus P_{11} \oplus P_{22}\,, \\
\phi(B) &= S_{01} \oplus S_{02} \oplus S_{10} \oplus S_{20}\,,
\end{aligned}
\tag{C.65}
$$

which is again consistent with $\mathsf{Rep}(D_8)$ fusion rules and now a non-trivial relative Euler term $e^{-\lambda} = \sqrt{2}$ when we compare the $S$ action with (C.58). One can see this describes an Ising gSSB phase.

Mathematically, we thus have provided information on a pivotal tensor functor describing an $n = 3$ intrinsic gSSB with broken Ising symmetry,

$$
\phi : \; \mathsf{Rep}(D_8) \to \mathcal{S}(\mathfrak{B}')\,,
\tag{C.66}
$$

where $\mathcal{S}(\mathfrak{B}')$ is the multi-fusion category formed by topological line operators living on the boundary $\mathfrak{B}'$.

## C.3 Lagrangian algebras

If we now focus on Lagrangian algebras, rather than starting from scratch we can simply work out different cases by starting with the 4d algebras and their phases that we have already calculated and further condense down to get the phases for the 8d (Lagrangian) algebras.

### C.3.1 SPT

The Lagrangian algebras that lead to SPT phases are $\mathcal{A}_{27}$, $\mathcal{A}_{30}$ and $\mathcal{A}_{32}$. We now focus on $\mathcal{A}_{27}$ to represent the problem, with the other two algebras following analogously. To get to $\mathcal{A}_{27}$ one can see from the Hasse diagram in figure 3 that there are three paths to $\mathcal{A}_{27}$ via $\mathcal{A}_{11}$, $\mathcal{A}_{14}$ and $\mathcal{A}_{21}$. The algebras $\mathcal{A}_{11}$ and $\mathcal{A}_{14}$ lead to gSPT of the type in (C.25) while $\mathcal{A}_{21}$ leads to a gSPT of the type in (C.29). In both cases if one then further condenses $1 \oplus m$ which is a Lagrangian algebra of the $\mathcal{Z}(\mathsf{Vec}_{\mathbb{Z}_2})$ center then that is equivalent to simply condensing $\mathcal{A}_{27}$ in one go.

In such a case the club quiche picture becomes a (club) sandwich, e.g. for $\mathcal{A}_{11}$ after imposing the Lagrangian condition on the right for $\mathcal{Z}(\mathsf{Vec}_{\mathbb{Z}_2})$ one finds

$$
\begin{array}{cc}
\mathcal{L}^{\text{sym}}_{\mathsf{Rep}(D_8)} \quad \mathcal{A}_{11} \quad\quad 1 \oplus m \quad\quad \mathcal{L}^{\text{sym}}_{\mathsf{Rep}(D_8)} \quad\quad \mathcal{A}_{27} & \\
\boxed{\mathfrak{Z}(\mathsf{Rep}(D_8)) \; \mathfrak{Z}(\mathsf{Vec}_{\mathbb{Z}_2})} \quad = \quad \boxed{\mathfrak{Z}(\mathsf{Rep}(D_8))} & ,
\end{array}
\tag{C.67}
$$

where there is no intersection between $\mathcal{L}^{\text{sym}}_{\mathsf{Rep}(D_8)} = \mathcal{A}_{33}$ and $\mathcal{A}_{27}$ and thus no non-trivial local operators are created by collapsing the sandwich and hence this setup gives rise to a SPT phase.

As the symmetry generators of $\mathsf{Rep}(D_8)$ will act trivially on the charges in the multiplets of $\mathcal{A}_{27}$ and there is only one vacuum, after collapsing the sandwich the target category for the pivotal functor will be just $\mathsf{Vec}$ with only one trivial line 1 and thus one finds the trivial

homomorphisms

$$
\begin{aligned}
\phi(1) &= 1\,, \\
\phi(R) &= 1\,, \\
\phi(G) &= 1\,, \\
\phi(RG) &= 1\,, \\
\phi(B) &= 1 \oplus 1\,.
\end{aligned}
\tag{C.68}
$$

This amounts to sending $P \to 1$ in (C.26) or (C.30), which is consistent with how $\mathsf{Vec}_{\mathbb{Z}_2}$ gets projected down to $\mathsf{Vec}$.

Mathematically, we have provided information on a pivotal tensor functor describing a SPT phase,

$$
\phi : \ \mathsf{Rep}(D_8) \to \mathsf{Vec}\,,
\tag{C.69}
$$

which is in fact a fiber functor as expected.

### C.3.2 $\mathbb{Z}_2$ SSB

One can arrive at a $n = 2$ SSB for the Lagrangian algebras with two different symmetry structures. The symmetry Lagrangian $\mathcal{L}^{\mathrm{sym}}_{\mathsf{Rep}(D_8)}$ may have a single intersection of one of $m_{RG}$, $m_{GB}$, $m_{RB}$ or it can have an intersection of $e_{RGB}$. The former leads to a phase where a $\mathbb{Z}_2$ subsymmetry of $\mathsf{Rep}(D_8)$ is spontaneously broken in both vacua. In the latter case, one finds that the $\mathbb{Z}_2 \times \mathbb{Z}_2$ subsymmetry of $\mathsf{Rep}(D_8)$ is spontaneously preserved in both vacua. Thus for gapped (or gapless) phases we would call these in the former case $\mathbb{Z}_2$ SSB phases and in the latter case $\mathsf{Rep}(D_8)/(\mathbb{Z}_2 \times \mathbb{Z}_2)$ SSB phases.

Here we study first the Lagrangian algebras with a single intersection being one of $m_{RG}$, $m_{GB}$, or $m_{RB}$, these are $\mathcal{A}_{34}$, $\mathcal{A}_{35}$ and $\mathcal{A}_{36}$ respectively. As a representative example let us pick $\mathcal{A}_{34}$ for which we can see from the Hasse diagram in figure 3 that there are three paths via $\mathcal{A}_{15}$, $\mathcal{A}_{16}$ and $\mathcal{A}_{17}$. The algebra $\mathcal{A}_{15}$ is described by a club quiche (C.41) which produces an $n = 2$ gSSB thus in this case one chooses the Lagrangian $1 \oplus m$ for the right boundary to not produce any more non-trivial topological local operators after collapsing the (club) sandwich and thus

$$
\mathcal{E}_{\mathcal{O}} \qquad\qquad\qquad\qquad\qquad
\tag{C.70}
$$

On the other hand, algebras $\mathcal{A}_{16}$ and $\mathcal{A}_{17}$ are associated to the gSPTs described by (C.25) hence for these one picks a Lagrangian $1 \oplus e$ to produce one non-trivial topological local operator and thus

$$
\mathcal{E}_{\mathcal{E}_e} \qquad\qquad\qquad\qquad\qquad
\tag{C.71}
$$

As expected both paths ultimately lead to the same sandwich picture with $\mathcal{A}_{34}$ on the right.

Going down the $\mathcal{A}_{15}$ path results in keeping the operator $\mathcal{O}$ and discarding $\mathcal{E}_1$ and $\mathcal{E}_2$ in (C.42). From this one finds the only non-trivial fusion being $\mathcal{O}^2 = 1$, and the two vacua

$$
v_i = \frac{1 + (-1)^i \mathcal{O}}{2}\,,
\tag{C.72}
$$

where $i \in \{0, 1\}$.

By looking at the linking action of the $\text{Rep}(D_8)$ generators, one can deduce their images on the collapsed sandwich as

$$
\begin{aligned}
\phi(1) &= 1_{00} \oplus 1_{11}, \\
\phi(R) &= 1_{01} \oplus 1_{10}, \\
\phi(G) &= 1_{01} \oplus 1_{10}, \\
\phi(RG) &= 1_{00} \oplus 1_{11}, \\
\phi(B) &= 1_{00} \oplus 1_{01} \oplus 1_{10} \oplus 1_{11},
\end{aligned}
\tag{C.73}
$$

which is again consistent with $\text{Rep}(D_8)$ fusion rules and the projection $P_{ij} \to 1_{ij}$. Hence we see there is a $\mathbb{Z}_2$ SSB in both vacua which is generated by the images of $R$ and $G$ which act by exchanging these two vacua. We also see that $B$ is spontaneously broken in both vacua as $B : v_i \to v_0 + v_1$, however, both vacua are physically indistinguishable as there is no non-trivial relative Euler term.

Alternatively, by going down the $\mathcal{A}_{16}$ or $\mathcal{A}_{17}$ route one finds the same vacua, symmetry actions, and ultimately the same pivotal functor. Hence the routes lead to the same $\mathbb{Z}_2$ SSB phase.

Mathematically, we have provided information on a pivotal tensor functor describing an $n = 2$ SSB

$$
\phi : \text{Rep}(D_8) \to \text{Mat}_2(\text{Vec}),
\tag{C.74}
$$

where $\text{Mat}_2(\text{Vec})$ is the multi-fusion category formed by $2 \times 2$ matrices in $\text{Vec}$ with no non-trivial relative Euler terms.

### C.3.3   $\text{Rep}(D_8)/(\mathbb{Z}_2 \times \mathbb{Z}_2)$ SSB

In this case by considering the Lagrangian algebra $\mathcal{A}_{37}$ one also finds an $n = 2$ SSB but with a different pivotal functor than in the cases described just above as we show this describes a $\text{Rep}(D_8)/(\mathbb{Z}_2 \times \mathbb{Z}_2)$ SSB phase. This is the case as the intersection with the symmetry boundary is formed by the line $e_{RGB}$ whose multiplet is uncharged under $R$, $G$, $RG$ but charged under $B$.

There are 7 paths in the Hasse diagram in figure 3 that lead to $\mathcal{A}_{37}$ via the 7 4d condensable algebras: $\mathcal{A}_{21}$, $\mathcal{A}_{22}$ and $\mathcal{A}_{23}$ describing a gSPT phase; $\mathcal{A}_{24}$, $\mathcal{A}_{25}$ and $\mathcal{A}_{26}$ describing a $\text{Rep}(D_8)/(\mathbb{Z}_2 \times \mathbb{Z}_2)$ gSSB phase; $\mathcal{A}_8$ describing the igSPT phase. Starting with the gSPT we can take $\mathcal{A}_{21}$ as the example, where we now need to end the $e$ line on the right to produce a non-trivial local operator and thus choosing $1 \oplus e$ as the boundary condition to close the (club) quiche as

$$
\begin{array}{c}
\mathcal{L}^{\text{sym}}_{\text{Rep}(D_8)} \quad \mathcal{A}_{21} \quad 1 \oplus e \\[4pt]
\boxed{\begin{array}{c|c} \mathfrak{Z}(\text{Rep}(D_8)) & \mathfrak{Z}(\text{Vec}_{\mathbb{Z}_2}) \\ \hline e_{RGB} & e \end{array}}
\end{array}
\;=\; \mathcal{E}_{\mathcal{E}_e}
\begin{array}{c}
\mathcal{L}^{\text{sym}}_{\text{Rep}(D_8)} \quad \mathcal{A}_{37} \\[4pt]
\boxed{\begin{array}{c} \mathfrak{Z}(\text{Rep}(D_8)) \\ \hline e_{RGB} \end{array}}
\end{array}
\;.
\tag{C.75}
$$

Similarly, for the igSPT phase, one needs to end the $s\bar{s}$ line on the right to produce a non-trivial topological local operator after collapsing the sandwich and thus one uses $1 \oplus s\bar{s}$ on the right as

$$
\begin{array}{c}
\mathcal{L}^{\text{sym}}_{\text{Rep}(D_8)} \quad \mathcal{A}_8 \quad 1 \oplus s\bar{s} \\[4pt]
\boxed{\begin{array}{c|c} \mathfrak{Z}(\text{Rep}(D_8)) & \mathfrak{Z}(\text{Vec}^{\omega}_{\mathbb{Z}_2}) \\ \hline e_{RGB} & s\bar{s} \end{array}}
\end{array}
\;=\; \mathcal{E}_{\mathcal{E}_{s\bar{s}}}
\begin{array}{c}
\mathcal{L}^{\text{sym}}_{\text{Rep}(D_8)} \quad \mathcal{A}_{37} \\[4pt]
\boxed{\begin{array}{c} \mathfrak{Z}(\text{Rep}(D_8)) \\ \hline e_{RGB} \end{array}}
\end{array}
\;.
\tag{C.76}
$$

Finally for the $\mathbb{Z}_2$ gSSB phase one can take $\mathcal{A}_{24}$ as the example, hence to end up with a $\mathbb{Z}_2$ SSB one needs to pick $1 \oplus m$ as the boundary to not give rise to more non-trivial topological local operators after collapsing the sandwich and thus picks

$$
\begin{array}{cc}
\mathcal{L}^{\text{sym}}_{\text{Rep}(D_8)} \quad \mathcal{A}_{24} \quad\quad 1 \oplus m & \mathcal{L}^{\text{sym}}_{\text{Rep}(D_8)} \quad \mathcal{A}_{37} \\
\end{array}
= \mathcal{E}_{\mathcal{O}} \qquad\qquad\qquad . \tag{C.77}
$$

In all of the cases above, after collapsing the sandwich we find only one non-trivial topological local operator $\mathcal{E} \to \mathcal{O}$ which follows $\mathcal{O}^2 = 1$ from which, together with the identity, one can build the two vacua of this SSB phase

$$
v_i = \frac{1 + (-1)^i \mathcal{O}}{2}, \tag{C.78}
$$

where $i \in \{0, 1\}$.

By looking at the linking action of the $\text{Rep}(D_8)$ generators, one can deduce their images on the collapsed sandwich as

$$
\begin{aligned}
\phi(1) &= 1_{00} \oplus 1_{11}, \\
\phi(R) &= 1_{00} \oplus 1_{11}, \\
\phi(G) &= 1_{00} \oplus 1_{11}, \\
\phi(RG) &= 1_{00} \oplus 1_{11}, \\
\phi(B) &= 2(1_{01} \oplus 1_{10}),
\end{aligned} \tag{C.79}
$$

which is again consistent with $\text{Rep}(D_8)$ fusion rules and e.g. the homomorphism $1 \to (1_{00} \oplus 1_{11})$, $P \to (1_{01} \oplus 1_{10})$ when compared to (C.30).

We can now see there is a clear difference between this $n = 2$ SSB phase and the $\mathbb{Z}_2$ SSB phase seen above. In the previous phase, it was $\phi(R)(= \phi(G))$ that acted to exchange the vacua, which was the spontaneously broken $\mathbb{Z}_2$ subsymmetry of $\text{Rep}(D_8)$. Now $\phi(R) = \phi(G) = 1$ and instead it is $\phi(B)$ that exchanges the vacua but also in the process multiplies them by a factor of 2, i.e.

$$
B: \quad v_0 \to 2v_1, \quad v_1 \to 2v_0. \tag{C.80}
$$

As the $\mathbb{Z}_2 \times \mathbb{Z}_2$ subsymmetry is preserved in this phase, we can describe this phase as a $\text{Rep}(D_8)/(\mathbb{Z}_2 \times \mathbb{Z}_2)$ SSB phase.

Mathematically, we have provided information on another pivotal tensor functor describing a different $n = 2$ SSB phase, in this case the $\text{Rep}(D_8)/(\mathbb{Z}_2 \times \mathbb{Z}_2)$ SSB phase,

$$
\phi: \text{Rep}(D_8) \to \text{Mat}_2(\text{Vec}), \tag{C.81}
$$

where $\text{Mat}_2(\text{Vec})$ is the multi-fusion category formed by $2 \times 2$ matrices in Vec with no non-trivial relative Euler terms.

### C.3.4 $\mathbb{Z}_2 \times \mathbb{Z}_2$ SSB

For the Lagrangian algebras $\mathcal{A}_{28}$, $\mathcal{A}_{29}$ and $\mathcal{A}_{31}$, one finds a $n = 4$ SSB phase. Let us again work with a representative example by picking $\mathcal{A}_{28}$ for which we see there are 3 paths in the Hasse diagram in figure 3 via the 4d algebras $\mathcal{A}_{15}$, $\mathcal{A}_{18}$ and $\mathcal{A}_{26}$. The algebra $\mathcal{A}_{15}$ is described by a club quiche (C.41) which produces an $\mathbb{Z}_2$ gSSB thus in this case one chooses the Lagrangian

$1 \oplus e$ for the right boundary to produce 2 additional non-trivial topological local operators after collapsing the (club) sandwich and thus

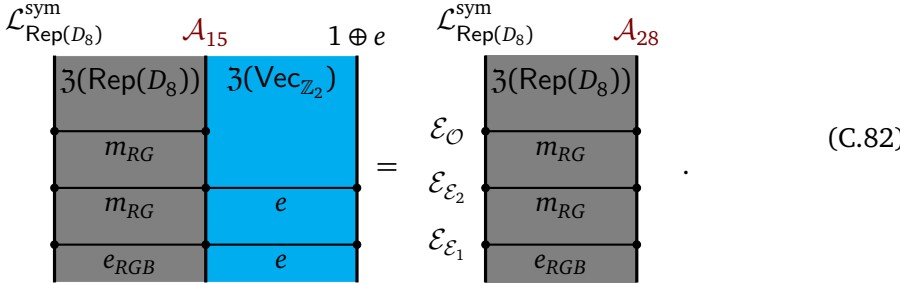

$$(C.82)$$

The algebra $\mathcal{A}_{26}$ follows very similarly as it is described by a club quiche (C.47) which produces the other $n = 2$ gSSB phase (which is in fact a $\text{Rep}(D_8)/(\mathbb{Z}_2 \times \mathbb{Z}_2)$ gSSB phase) thus in this case one chooses the Lagrangian $1 \oplus e$ again for the right boundary to produce 2 additional non-trivial topological local operators after collapsing the (club) sandwich and thus

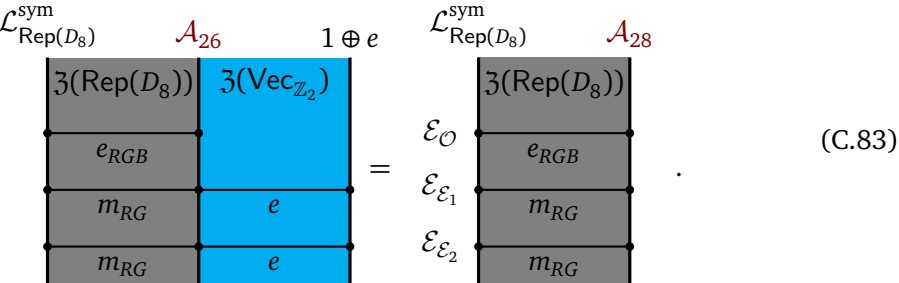

$$(C.83)$$

Finally, the algebra $\mathcal{A}_{18}$ is described by a club quiche (C.60) which produces the Ising igSSB phase thus in this case one chooses the Lagrangian $1 \oplus m$ for the right boundary to produce one more non-trivial topological local operator after collapsing the (club) sandwich and thus

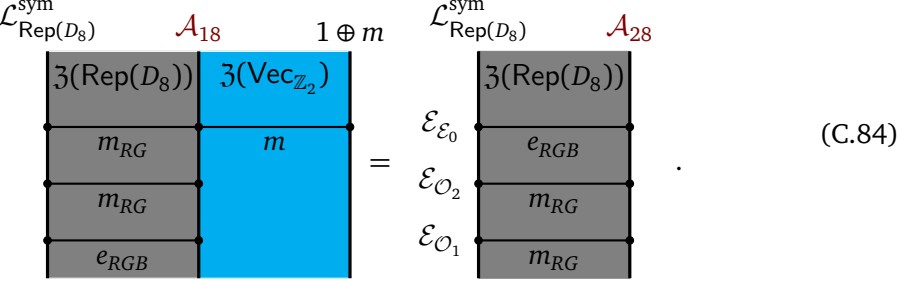

$$(C.84)$$

One can see all three ways of ultimately condensing the Lagrangian algebra $\mathcal{A}_{28}$ as captured in the Hasse diagram.

In the case of algebra $\mathcal{A}_{15}$, one finds that after collapsing the sandwich all $\mathcal{E}$'s become local operators related to the original operators in (C.42) where $\mathcal{E}_{\mathcal{O}}$ descends to operator $\mathcal{O}$, $\mathcal{E}_{\mathcal{E}_1}$ descends to $\mathcal{O}_1$ which is the collapsed operator $\mathcal{E}_1$ and $\mathcal{E}_{\mathcal{E}_2}$ descends to $\mathcal{O}_2$ which is the collapsed operator $\mathcal{E}_2$. The fusion rules for the new set of local operators $\{1, \mathcal{O}, \mathcal{O}_1, \mathcal{O}_2\}$ will then follow the one in (C.42) with $\mathcal{O}_1 \leftrightarrow \mathcal{E}_1$ and $\mathcal{O}_2 \leftrightarrow \mathcal{E}_2$. Now we can again form more canonical-looking operators by defining

$$\widehat{\mathcal{O}}_1 \equiv \frac{\mathcal{O}_1 + \mathcal{O}_2}{2}, \qquad \widehat{\mathcal{O}}_2 \equiv \frac{\mathcal{O}_1 - \mathcal{O}_2}{2}, \tag{C.85}$$

where $\widehat{\mathcal{O}}_1$ and $\widehat{\mathcal{O}}_2$ are related to the canonical ends $\widehat{\mathcal{E}}_1$ and $\widehat{\mathcal{E}}_2$ respectively. Now after introducing $\mathcal{O}_1$ and $\mathcal{O}_2$, or equivalently $\widehat{\mathcal{O}}_1$ and $\widehat{\mathcal{O}}_2$, one finds that the original vacua $v_0$ and $v_1$ are

no longer good vacua as $\widehat{\mathcal{O}}_1^2 = v_0$ and $\widehat{\mathcal{O}}_2^2 = v_1$, and the newly broken symmetry exchanges $\widehat{\mathcal{O}}_1 \leftrightarrow \widehat{\mathcal{O}}_2$. One instead finds that the two original vacua now split into two each as

$$v_0^\pm = \frac{v_0 \pm \widehat{\mathcal{O}}_1}{2}, \qquad v_1^\pm = \frac{v_1 \pm \widehat{\mathcal{O}}_2}{2}, \tag{C.86}$$

which means we further break down another $\mathbb{Z}_2$ subsymmetry of $\mathrm{Rep}(D_8)$ in both original vacua and thus produce 4 vacua in total which are exchanged by $\mathbb{Z}_2 \times \mathbb{Z}_2$ spontaneously broken subsymmetry of $\mathrm{Rep}(D_8)$. The originally broken $\mathbb{Z}_2$ exchanged the vacua $v_0 \leftrightarrow v_1$ and now it is enlarged as it exchanges $v_0^+ \leftrightarrow v_1^+$ and $v_0^- \leftrightarrow v_1^-$.

One can proceed similarly for $\mathcal{A}_{26}$ with corresponding notations. Yet in this case the original vacua $v_0$ and $v_1$ are invariant under the entire $\mathbb{Z}_2 \times \mathbb{Z}_2$ subsymmetry of $\mathrm{Rep}(D_8)$ as this is condensing from a $\mathrm{Rep}(D_8)/(\mathbb{Z}_2 \times \mathbb{Z}_2)$ gSSB phase. By analogously introducing the operators $\widehat{\mathcal{O}}_i \sim \widehat{\mathcal{E}}_i$ for (C.48), one finds that the newly broken symmetry sends $\widehat{\mathcal{O}}_i \to -\widehat{\mathcal{O}}_i$ while $\widehat{\mathcal{O}}_1^2 = v_0$ and $\widehat{\mathcal{O}}_2^2 = v_1$. One then needs to construct new vacua from the old which end up being the same as (C.86) but the symmetry action will be $v_0^+ \leftrightarrow v_0^-$ and $v_1^+ \leftrightarrow v_1^-$. Thus suddenly one finds a phase where the $\mathbb{Z}_2 \times \mathbb{Z}_2$ subsymmetry is spontaneously broken again. Thus by condensing the last $\mathbb{Z}_2$ symmetry suddenly the phase where the $\mathbb{Z}_2 \times \mathbb{Z}_2$ subsymmetry was preserved is entirely broken, i.e. the $\mathrm{Rep}(D_8)/(\mathbb{Z}_2 \times \mathbb{Z}_2)$ gSSB goes to $\mathbb{Z}_2 \times \mathbb{Z}_2$ SSB. Encouragingly, by relabeling $v_0^- \leftrightarrow v_1^+$ one finds a complete agreement with the result for $\mathcal{A}_{15}$.

Finally, for $\mathcal{A}_{18}$ the situation is slightly different as in this case the ends $\widehat{\mathcal{E}}_1$ and $\widehat{\mathcal{E}}_2$ will not be collapsed to non-trivial operators but $\widehat{\mathcal{E}}_0 \sim \widehat{\mathcal{O}}_0$ will. The newly created operator $\widehat{\mathcal{O}}_0$ will transform as $\widehat{\mathcal{O}}_0 \to -\widehat{\mathcal{O}}_0$ and $\widehat{\mathcal{O}}_0^2 = v_0$. Hence the vacua $v_1$ and $v_2$ in (C.62) stay the same but the vacuum $v_0$ which was invariant under $\mathbb{Z}_2 \times \mathbb{Z}_2$ now splits as

$$v_0^\pm = \frac{v_0 \pm \widehat{\mathcal{O}}_0}{2}. \tag{C.87}$$

Now the newly broken symmetry acts to exchange vacua $v_1 \leftrightarrow v_2$ and $v_0^+ \leftrightarrow v_0^-$. So in the end one again finds the same phase with a broken $\mathbb{Z}_2 \times \mathbb{Z}_2$ subsymmetry after a relabeling as expected. This is the case where the remaining symmetry of the Ising igSSB phase gets further broken down, increasing the number of vacua by 1, as it ends up as a $\mathbb{Z}_2 \times \mathbb{Z}_2$ SSB phase.

One can then conclude that all three ways of sequentially condensing various algebras ultimately lead to the same result. However, as we have shown, by utilizing the club quiche/sandwich picture, one can observe how these condensations affect the given phases in finer detail. For example in the case of $\mathcal{A}_{26}$, which descends from the $\mathbb{Z}_4$ symmetric club quiche (C.4), before imposing the $1 \oplus e$ boundary condition to close the club quiche, the phase has two vacua which are invariant under the entire $\mathbb{Z}_2 \times \mathbb{Z}_2$ subsymmetry of $\mathrm{Rep}(D_8)$, but right after it has 4 vacua with the $\mathbb{Z}_2 \times \mathbb{Z}_2$ fully broken.

By fixing the notation in (C.86) to be $v_0 = v_0^+$, $v_1 = v_1^+$, $v_2 = v_0^-$, $v_3 = v_1^-$ and by looking at the linking action of the $\mathrm{Rep}(D_8)$ generators, one can deduce their images on the collapsed sandwich as

$$\begin{aligned}
\phi(1) &= 1_{00} \oplus 1_{11} \oplus 1_{22} \oplus 1_{33}, \\
\phi(R) &= 1_{01} \oplus 1_{10} \oplus 1_{23} \oplus 1_{32}, \\
\phi(G) &= 1_{01} \oplus 1_{10} \oplus 1_{23} \oplus 1_{32}, \\
\phi(RG) &= 1_{00} \oplus 1_{11} \oplus 1_{22} \oplus 1_{33}, \\
\phi(B) &= 1_{02} \oplus 1_{03} \oplus 1_{12} \oplus 1_{13} \oplus 1_{20} \oplus 1_{21} \oplus 1_{30} \oplus 1_{31},
\end{aligned} \tag{C.88}$$

which is again consistent with $\mathrm{Rep}(D_8)$ fusion rules.

Mathematically, we have provided information on a pivotal tensor functor describing a $n = 4$ SSB phase

$$\phi : \mathrm{Rep}(D_8) \to \mathrm{Mat}_4(\mathrm{Vec}), \tag{C.89}$$

where $\mathrm{Mat}_4(\mathrm{Vec})$ is the multi-fusion category formed by $4 \times 4$ matrices in Vec with no non-trivial relative Euler terms. We refer to a phase with such symmetry structure as a $\mathbb{Z}_2 \times \mathbb{Z}_2$ SSB phase.

### C.3.5   Rep($D_8$) SSB

If one chooses the Rep($D_8$) symmetry Lagrangian $\mathcal{L}^{\mathrm{sym}}_{\mathrm{Rep}(D_8)} = \mathcal{A}_{33}$ itself as the condensable algebra one ends up breaking the entire Rep($D_8$) symmetry in that phase. We can see from the Hasse diagram in figure 3 that there are 3 paths to $\mathcal{A}_{33}$ via the 4d algebras $\mathcal{A}_{18}$, $\mathcal{A}_{19}$ and $\mathcal{A}_{20}$. All three of these 4d condensable algebras are described by the quiche in (C.60) which is the Ising igSSB phase, hence we focus on $\mathcal{A}_{18}$ to be the representative example. In this case, we impose the $1 \oplus e$ boundary condition on the right to produce 2 new non-trivial topological local operators after collapsing the (club) sandwich, thus bringing the total number of non-trivial operators to 4. The club sandwich picture will then be

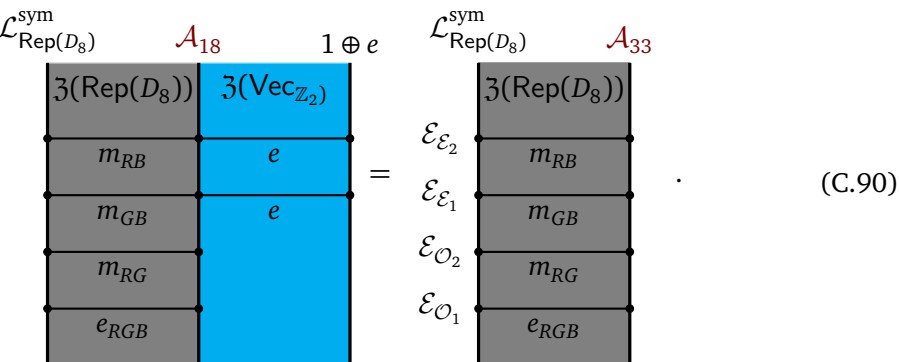

$$\tag{C.90}$$

One finds that after collapsing the sandwich all $\mathcal{E}$'s become local operators related to the original operators in (C.61) where $\mathcal{E}_{\mathcal{O}_i}$ descend to operator $\mathcal{O}_i$, $\mathcal{E}_{\mathcal{E}_i}$ descend to $\mathcal{O}_i^{\mathcal{E}}$ which are the collapsed operators $\mathcal{E}_i$, whereas the original end $\mathcal{E}_0$ will not end on the right and thus not descend to a non-trivial local operator. The fusion rules for the new set of local operators $\{1, \mathcal{O}_1, \mathcal{O}_2, \mathcal{O}_1^{\mathcal{E}}, \mathcal{O}_2^{\mathcal{E}}\}$ will then follow the one in (C.61) with $\mathcal{O}_i^{\mathcal{E}} \sim \mathcal{E}_i$. The new set of local operators now follows the fusion rules

$$
\begin{aligned}
\mathcal{O}_1^2 &= 1, & \mathcal{O}_2 \mathcal{O}_1^{\mathcal{E}} &= \mathcal{O}_2^{\mathcal{E}}, \\
\mathcal{O}_1 \mathcal{O}_2 &= \mathcal{O}_2, & \mathcal{O}_2 \mathcal{O}_2^{\mathcal{E}} &= \mathcal{O}_1^{\mathcal{E}}, \\
\mathcal{O}_1 \mathcal{O}_i^{\mathcal{E}} &= \mathcal{O}_i^{\mathcal{E}}, & (\mathcal{O}_i^{\mathcal{E}})^2 &= (1 + \mathcal{O}_1)/2, \\
\mathcal{O}_2^2 &= (1 + \mathcal{O}_1)/2, & \mathcal{O}_1^{\mathcal{E}} \mathcal{O}_2^{\mathcal{E}} &= \mathcal{O}_2.
\end{aligned}
\tag{C.91}
$$

Now we can again form more canonical-looking operators by defining

$$
\begin{aligned}
v_0 &= \frac{1 - \mathcal{O}_1}{2}, \\
v_1 &= \frac{1 + \mathcal{O}_1 + 2\mathcal{O}_2}{4}, & \widehat{\mathcal{O}}_1^{\mathcal{E}} &= \frac{\mathcal{O}_1^{\mathcal{E}} + \mathcal{O}_2^{\mathcal{E}}}{2}, \\
v_2 &= \frac{1 + \mathcal{O}_1 - 2\mathcal{O}_2}{4}, & \widehat{\mathcal{O}}_2^{\mathcal{E}} &= \frac{\mathcal{O}_1^{\mathcal{E}} - \mathcal{O}_2^{\mathcal{E}}}{2},
\end{aligned}
\tag{C.92}
$$

with this redefinition one finds

$$
v_a v_b = \delta_{ab} v_b, \qquad v_a \widehat{\mathcal{O}}_i^{\mathcal{E}} = \delta_{ai} \widehat{\mathcal{O}}_i^{\mathcal{E}}, \qquad \widehat{\mathcal{O}}_i^{\mathcal{E}} \widehat{\mathcal{O}}_j^{\mathcal{E}} = \delta_{ij} v_j,
\tag{C.93}
$$

for $a, b \in \{0, 1, 2\}$ and $i, j \in \{1, 2\}$.

As the original end of the $m$ line on $\mathfrak{B}'$ denoted by $\widehat{\mathcal{E}}_0$ is now trivial, one finds that the vacuum $v_0$ remains a good vacuum. However, as the original ends of the two $e$ lines $\widehat{\mathcal{E}}_1$ and $\widehat{\mathcal{E}}_2$ now become local operators $\widehat{\mathcal{O}}_1^{\mathcal{E}}$ and $\widehat{\mathcal{O}}_2^{\mathcal{E}}$ after collapsing the sandwich, the original vacua $v_1$ and $v_2$ both split in two, so one finds a new set of vacua

$$v_0 = v_0\,, \qquad v_1^\pm = \frac{v_1 \pm \widehat{\mathcal{O}}_1^{\mathcal{E}}}{2}\,, \qquad v_2^\pm = \frac{v_2 \pm \widehat{\mathcal{O}}_2^{\mathcal{E}}}{2}\,. \tag{C.94}$$

However for our purposes, it may be easier to take these vacua and relabel them as

$$v_1 = v_1^+\,, \qquad v_R = v_2^+\,, \qquad v_G = v_2^-\,, \qquad v_{RG} = v_1^-\,, \qquad v_B = v_0\,, \tag{C.95}$$

as this will make it easier the effect of the linking actions of $\mathrm{Rep}(D_8)$ on these vacua

$$\begin{aligned} R: &\quad v_B \to v_B\,, &\quad v_1 \leftrightarrow v_R\,, &\quad v_G \leftrightarrow v_{RG}\,, \\ G: &\quad v_B \to v_B\,, &\quad v_1 \leftrightarrow v_G\,, &\quad v_R \leftrightarrow v_{RG}\,, \\ RG: &\quad v_B \to v_B\,, &\quad v_1 \leftrightarrow v_{RG}\,, &\quad v_R \leftrightarrow v_G\,, \\ B: &\quad v_B \to v_1 + v_R + v_G + v_{RG}\,, &\quad v_1, v_R, v_G, v_{RG} \to v_B\,, \end{aligned} \tag{C.96}$$

from which we can see that the vacua indeed transform among themselves according to the broken $\mathrm{Rep}(D_8)$ symmetry, with no non-trivial relative Euler terms between the vacua. From these linking actions of the $\mathrm{Rep}(D_8)$ generators, one can deduce their images on the collapsed sandwich as

$$\begin{aligned} \phi(1) &= 1_{BB} \oplus 1_{11} \oplus 1_{RR} \oplus 1_{GG} \oplus 1_{(RG)(RG)}\,, \\ \phi(R) &= 1_{BB} \oplus 1_{1R} \oplus 1_{R1} \oplus 1_{G(RG)} \oplus 1_{(RG)G}\,, \\ \phi(G) &= 1_{BB} \oplus 1_{1G} \oplus 1_{G1} \oplus 1_{R(RG)} \oplus 1_{(RG)R}\,, \\ \phi(RG) &= 1_{BB} \oplus 1_{1(RG)} \oplus 1_{(RG)1} \oplus 1_{RG} \oplus 1_{GR}\,, \\ \phi(B) &= 1_{B1} \oplus 1_{BR} \oplus 1_{BG} \oplus 1_{B(RG)} \oplus 1_{1B} \oplus 1_{RB} \oplus 1_{GB} \oplus 1_{(RG)B}\,, \end{aligned} \tag{C.97}$$

which is again consistent with $\mathrm{Rep}(D_8)$ fusion rules.

Mathematically, we have provided information on a pivotal tensor functor describing a full $n = 5$ SSB

$$\phi: \ \mathrm{Rep}(D_8) \to \mathrm{Mat}_5(\mathrm{Vec})\,, \tag{C.98}$$

where $\mathrm{Mat}_5(\mathrm{Vec})$ is the multi-fusion category formed by $5 \times 5$ matrices in Vec with no non-trivial relative Euler terms. We refer to a phase with such symmetry structure as a $\mathrm{Rep}(D_8)$ SSB phase.

# D  Kennedy-Tasaki (KT) transformations for $\mathrm{Rep}(D_8)$ and phase transitions

In this Appendix, we discuss a few representative examples of Kennedy-Tasaki (KT) transformations for theories with $\mathrm{Rep}(D_8)$ symmetry using the SymTFT Quiches found in the preceding Appendix. These examples provide further details on the results shown in Table 4, Table 5 and Table 6.

The maps constructed below are inspired by the Kennedy–Tasaki (KT) unitary [161], which converts the $\mathbb{Z}_2 \times \mathbb{Z}_2$ Haldane SPT into a symmetry-broken phase with the symmetry. Here we adopt the term in a broader sense: our "generalized KT" transformations, introduced in [10, 162], realize a categorical analogue of the original KT duality of the Haldane chain.

Table 8: Operators in the Ising CFT.

|  | Name | $\mathbb{Z}_2$-charge | $\mathbb{Z}_2$-sector |
|---|---|---|---|
| $\sigma$ | spin/order | $-1$ | untwisted |
| $\mu$ | disorder | $+1$ | twisted |
| $\epsilon$ | energy | $+1$ | untwisted |

Let $\mathcal{S}$ denote the target symmetry category and let $\mathcal{I}_\mathcal{A}$ be the gapped interface determined by a condensable algebra $\mathcal{A} \in \mathcal{Z}(\mathcal{S})$. Collapsing the two intervals in the diagram

$$\text{SymTFT}(\mathcal{S}) \xleftarrow{\mathcal{I}_\mathcal{A}} \text{SymTFT}(\mathcal{S}') \dashrightarrow \mathfrak{B}^{\text{phys}},$$

implements a functorial map

$$\mathcal{K}^{\mathcal{I}_\mathcal{A}}_{\mathcal{S},\mathcal{S}'} : \text{QFT}_{\mathcal{S}'} \longrightarrow \text{QFT}_\mathcal{S}, \tag{D.1}$$

which sends any $\mathcal{S}'$-symmetric phase – gapped or gapless – to an $\mathcal{S}$-symmetric one. On the level of topological lines this map is the charge-conversion functor

$$F_{\mathcal{I}_\mathcal{A}} : \mathcal{Z}(\mathcal{S}') \longrightarrow \mathcal{Z}(\mathcal{S}), \qquad F_{\mathcal{I}_\mathcal{A}}\big(\mathbf{Q}'_{a'}\big) = \bigoplus_a n_{a,a'} \mathbf{Q}_a, \tag{D.2}$$

where the integers $n_{a,a'}$ are the multiplicities appearing in the minimal Lagrangian $\mathcal{L}_{\mathcal{I}_\mathcal{A}} = \bigoplus_{a,a'} n_{a,a'} \mathbf{Q}_a \overline{\mathbf{Q}}'_{a'} \in \mathcal{Z}(\mathcal{S}) \boxtimes \overline{\mathcal{Z}}(\mathcal{S}')$ that completes the interface. Thus the transformation simply re-labels symmetry defects (and hence phase data) without closing the gap, thereby reproducing the structural role of the lattice KT unitary [161] in a fully topological framework.

When $\mathcal{L}_{\mathcal{I}_\mathcal{A}}$ is minimal, the map $\mathcal{K}^{\mathcal{I}_\mathcal{A}}_{\mathcal{S},\mathcal{S}'}$ preserves irreducibility: it carries irreducible $\mathcal{S}'$-symmetric gapped phases and critical points onto irreducible $\mathcal{S}$-symmetric ones. Whether the resulting theory is gapped or gapless is dictated solely by the choice of physical boundary $\mathfrak{B}^{\text{phys}}$; the KT transformation itself simply transports a known critical point (such as the Ising CFT) between different symmetry categories.

Thus, generalized KT transformations are a powerful tool that allows one to construct a map from a theory with an $\mathcal{S}'$ symmetry to a theory with a larger symmetry $\mathcal{S}$. The main use of such a tool can then be found in realizing new phase transition theories from ones which are known. A classic example of phase transition theory is the 2d Ising CFT

$$\mathfrak{T}^{\mathcal{S}'}_C = \text{Ising} \,\overset{\curvearrowleft}{\phantom{x}}\, \mathbb{Z}_2 \quad, \tag{D.3}$$

which serves the role of a $\mathbb{Z}_2$-symmetric phase transition. It turns out that for $\text{Rep}(D_8)$ symmetry one can effectively construct any phase transition using the Ising CFT or a few copies of it.

The Ising CFT has three operators of interest for our discussion which are listed in Table 8. where a twisted-sector operator is attached to the line $P$ generating $\mathbb{Z}_2$ symmetry. If we take the SymTFT $\mathfrak{Z}(\text{Vec}_{\mathbb{Z}_2})$ with

$$B^{\text{sym}}_{\text{Vec}_{\mathbb{Z}_2}} = \mathcal{L}_e = 1 \oplus e, \tag{D.4}$$

then from table 8, we deduce that the generalized charges are [10]:

$$q(\sigma) = e, \qquad q(\mu) = m, \qquad q(\epsilon) = 1. \tag{D.5}$$

As is well-known, the Ising CFT describes a phase transition between an SPT phase for $\mathbb{Z}_2$ symmetry and a $\mathbb{Z}_2$ SSB phase, which we denote by $\mathfrak{T}_{\text{SPT}}^{\text{Vec}_{\mathbb{Z}_2}}$ and $\mathfrak{T}_{\text{SSB}}^{\text{Vec}_{\mathbb{Z}_2}}$ respectively. The former consists of a single vacuum on which $\mathbb{Z}_2$ is realized trivially

$$\mathfrak{T}_{\text{SPT}}^{\text{Vec}_{\mathbb{Z}_2}} = \text{Trivial} \, \circlearrowleft \, \mathbb{Z}_2 \quad , \tag{D.6}$$

which has a string order parameter given by $\mu$, whereas the latter presents two vacua exchanged by the spontaneously broken $\mathbb{Z}_2$ symmetry generator

$$\mathfrak{T}_{\text{SSB}}^{\text{Vec}_{\mathbb{Z}_2}} = \underbrace{\text{Trivial}_0 \oplus \text{Trivial}_1}_{\mathbb{Z}_2} . \tag{D.7}$$

We denote the identity local operators (also called vacua) in each theory as $v_i$, $i \in \{0, 1\}$, which satisfy

$$v_i v_j = \delta_{ij} v_i , \qquad 1 \equiv v_0 + v_1 . \tag{D.8}$$

The broken $\mathbb{Z}_2$ generator $P$ acts by exchanging the two vacua

$$P : \; v_0 \leftrightarrow v_1 . \tag{D.9}$$

The order parameter for SSB of $\mathbb{Z}_2$ symmetry is the spin operator $\sigma$ of the Ising CFT, which acquires a non-zero vacuum expectation value (vev) in each vacuum. It can be identified as the operator

$$\sigma \equiv v_0 - v_1 , \tag{D.10}$$

where the coefficients are the vev that $\sigma$ takes in each vacuum. We note that the action of the $\mathbb{Z}_2$ generator $P$, eq. (D.9), sends $\sigma \to -\sigma$, as required.

## D.1 Condensable algebras of dimension 4

Here we provide an example of a simple $\text{Rep}(D_8)$-symmetric transition using a KT transformation from $\mathbb{Z}_2$ to $\text{Rep}(D_8)$ on the Ising CFT. Other phase transitions can be found analogously using the machinery found in [10] or using similar calculations in this Appendix. We also provide the KT transformation describing the igSPT with non-invertible symmetry which is one of the main heroes of this work which we note is not a phase transition as it can only flow to a single gapped phase.

### D.1.1 KT from $\mathbb{Z}_2$ to $\text{Rep}(D_8)$ symmetry I

Let us discuss the KT transformation associated to the condensable algebra $\mathcal{A}_{11}$, (and analogously $\mathcal{A}_{10}$, $\mathcal{A}_{13}$, $\mathcal{A}_{14}$, $\mathcal{A}_{16}$, or $\mathcal{A}_{17}$), whose properties we described in subsection C.2.1.

From the folded Lagrangian algebra $\mathcal{L}_{11}$, eq. (B.16) and associated to the condensable algebra $\mathcal{A}_{11}$, we learn that the map of generalized charges

$$\mathcal{Z}(\text{Vec}_{\mathbb{Z}_2}) \to \mathcal{Z}(\text{Rep}(D_8)) \tag{D.11}$$

is the following:

$$\begin{aligned}
1 &\to 1 \oplus e_R \oplus m_B , & e &\to m_G \oplus m_{GB} , \\
m &\to e_G \oplus e_{RG} \oplus m_B , & em &\to f_G \oplus f_{GB} .
\end{aligned} \tag{D.12}$$

According to this map, one finds the following operators:

- An untwisted operator $\mathcal{O}'_1$ of $\mathfrak{T}^{\mathcal{S}'}$ uncharged under $\mathcal{S}' = \mathbb{Z}_2$ descends to four operators of $\mathfrak{T}^{\mathcal{S}}$ that are all uncharged under $\mathcal{S} = \mathsf{Rep}(D_8)$: an untwisted operator $\mathcal{O}_1$, an $R$-twisted operator $\mathcal{O}_{e_R}$, a $B$-twisted sector operators $\mathcal{O}^{(1)}_{m_B,1}$ and $\mathcal{O}^{(1)}_{m_B,2}$:

$$
\begin{aligned}
\mathcal{O}_1 &= \mathcal{O}'_1, & \mathcal{O}_{e_R} &= \mathcal{O}'_1, \\
\mathcal{O}^{(1)}_{m_B,1} &= \mathcal{O}'_1, & \mathcal{O}^{(1)}_{m_B,2} &= \mathcal{O}'_1.
\end{aligned}
\tag{D.13}
$$

  Recalling the homomorphism given in eq. (C.26), $\mathcal{O}_{e_R}$ is the operator $\mathcal{O}'_1$ recognized as sitting at the end of the $R$ line since $\phi(R) = 1$. Similarly, from $\phi(B) = 1 \oplus P$, we interpret $\mathcal{O}^{(1)}_{m_B,1}$ and $\mathcal{O}^{(1)}_{m_B,2}$ as living at the end of the $B$ line as the identity line of the $\mathbb{Z}_2$ symmetry sitting within $\phi(B)$.

- An untwisted operator $\mathcal{O}'_e$ of $\mathfrak{T}^{\mathcal{S}'}$ charged under $\mathcal{S}' = \mathbb{Z}_2$ descends to four operators of $\mathfrak{T}^{\mathcal{S}}$ charged under $G$, $RG$, $B$: $\mathcal{O}_{m_G,1}$ and $\mathcal{O}_{m_G,2}$ in the $B$-twisted sector, $\mathcal{O}_{m_{GB},1}$ which is untwisted, and $\mathcal{O}_{m_{GB},2}$ in $R$-twisted sector:

$$
\begin{aligned}
\mathcal{O}_{m_G,1} &= \mathcal{O}'_e, & \mathcal{O}_{m_G,2} &= \mathcal{O}'_e, \\
\mathcal{O}_{m_{GB},1} &= \mathcal{O}'_e, & \mathcal{O}_{m_{GB},2} &= \mathcal{O}'_e.
\end{aligned}
\tag{D.14}
$$

  $\mathcal{O}_{m_G,1}$ and $\mathcal{O}_{m_G,2}$ are interpreted as local operators at the end of the $B$ line since $\phi(B) = 1 \oplus P$ contains $1$, $\mathcal{O}_{m_{GB},1}$ is untwisted and $\mathcal{O}_{m_{GB},2}$ lives at the end of the $\phi(R) = 1$ line.

- A P-twisted operator $\mathcal{O}'_m$ of $\mathfrak{T}^{\mathcal{S}'}$ uncharged under $\mathcal{S}' = \mathbb{Z}_2$ descends to four operators of $\mathfrak{T}^{\mathcal{S}}$ uncharged under $\mathcal{S} = \mathsf{Rep}(D_8)$: $\mathcal{O}_{e_G}$ and $\mathcal{O}_{e_{RG}}$, respectively in $G$ and $RG$ twisted sectors, and $B$-twisted sector operators $\mathcal{O}^{(m)}_{m_B,1}$ and $\mathcal{O}^{(m)}_{m_B,2}$:

$$
\begin{aligned}
\mathcal{O}_{e_G} &= \mathcal{O}'_m, & \mathcal{O}_{e_{RG}} &= \mathcal{O}'_m, \\
\mathcal{O}^{(m)}_{m_B,1} &= \mathcal{O}'_m, & \mathcal{O}^{(m)}_{m_B,2} &= \mathcal{O}'_m.
\end{aligned}
\tag{D.15}
$$

  From eq. (C.26), we deduce that $\mathcal{O}_{e_G}$ and $\mathcal{O}_{e_{RG}}$ are local operators respectively at the end of $\phi(G) = P$ and $\phi(RG) = P$, while both $\mathcal{O}^{(m)}_{m_B,1}$ and $\mathcal{O}^{(m)}_{m_B,2}$ live at the end of $\phi(B) = 1 \oplus P$.

- A P-twisted operator $\mathcal{O}'_{em}$ of $\mathfrak{T}^{\mathcal{S}'}$ charged under $\mathcal{S}' = \mathbb{Z}_2$ descends to four operators of $\mathfrak{T}^{\mathcal{S}}$ charged under $G$, $RG$, $B$: $\mathcal{O}_{f_G,1}$ and $\mathcal{O}_{f_G,2}$, both $B$-twisted operators; $\mathcal{O}_{f_{GB},1}$ and $\mathcal{O}_{f_{GB},2}$, that are in $G$ and $RG$-twisted sectors, respectively:

$$
\begin{aligned}
\mathcal{O}_{f_G,1} &= \mathcal{O}'_{em}, & \mathcal{O}_{f_G,2} &= \mathcal{O}'_{em}, \\
\mathcal{O}_{f_{GB},1} &= \mathcal{O}'_{em}, & \mathcal{O}_{f_{GB},2} &= \mathcal{O}'_{em}.
\end{aligned}
\tag{D.16}
$$

  We interpret $\mathcal{O}_{f_G,1}$ and $\mathcal{O}_{f_G,2}$ as operators $\mathcal{O}'_{em}$ living at the end of $\phi(B) = 1 \oplus P$; similarly, the operators $\mathcal{O}_{f_{GB},1}$ and $\mathcal{O}_{f_{GB},2}$ as operators $\mathcal{O}'_{em}$ living at the ends of $\phi(G) = P$ and $\phi(RG) = P$ respectively.

### D.1.2 Example phase transition between SPT and $\mathbb{Z}_2$ SSB phase for $\mathsf{Rep}(D_8)$ symmetry

In this short example, we discuss a phase transition after obtaining the relevant KT transformation. The condensable algebra $\mathcal{A}_{11}$, (and analogously $\mathcal{A}_{10}$, $\mathcal{A}_{13}$, $\mathcal{A}_{14}$, $\mathcal{A}_{16}$, or $\mathcal{A}_{17}$), describes a phase transition between an SPT and $\mathbb{Z}_2$ SSB phase for $\mathsf{Rep}(D_8)$ symmetry, as can be seen from the Hasse diagram in figure 3 and phase table 3. Indeed, the club quiche (C.25) has reduced center $\mathcal{Z}(\mathsf{Vec}_{\mathbb{Z}_2})$, which admits two topological boundary conditions, listed in equation

(C.23). We can thus obtain two gapped phases after compactifying the club sandwich: picking $\mathcal{L}_m$ as $\mathfrak{B}^{\text{phys}}$ gives an SPT phase (i.e. with $\mathcal{S} = \text{Rep}(D_8)$ fully preserved), as described in subsection C.3.1, whereas by choosing $\mathcal{L}_e$ as $\mathfrak{B}^{\text{phys}}$ the final gapped phase is a $\mathbb{Z}_2$ SSB, discussed in subsection C.3.2. In the $\mathbb{Z}_2$ SSB to which the gapless phase associated to $\mathcal{A}_{11}$ can flow, the broken $\text{Rep}(D_8)$ generators are $G, RG, B$ and their action on the two vacua $v_0, v_1$ is:

$$
\begin{aligned}
G, RG : & \quad v_0 \leftrightarrow v_1 \,, \\
B : & \quad v_i \to v_0 + v_1 \,, \quad i \in \{0, 1\} \,,
\end{aligned}
$$

whereas $R$ is preserved in both vacua. The gapless phase responsible for the transition between $\text{Rep}(D_8)$-SPT and $\mathbb{Z}_2$ SSB phases is simply the Ising CFT regarded as a $\text{Rep}(D_8)$-symmetric theory:

$$
\mathcal{T}_C^{\mathcal{S}} = \text{Ising} \,\, \circlearrowleft \text{Rep}(D_8) \,, \tag{D.17}
$$

where the homomorphism $\phi$ between the $\text{Rep}(D_8)$ generators $R, G, RG, B$ and the $\mathbb{Z}_2$ generator $P$ is given in equation (C.26). The relevant operator responsible for the $\text{Rep}(D_8)$-symmetric transition is $\epsilon$.

The SPT phase for $\text{Rep}(D_8)$ symmetry is obtained by closing the club quiche with $\mathcal{L}_m = 1 \oplus m$, therefore, from equations (D.13) and (D.15), the order parameters for this phase are realized by the operators:

$$
\begin{aligned}
\mathcal{O}_1 &= \mathcal{O}_{e_R} = \mathcal{O}^{(1)}_{m_B, 1} = \mathcal{O}^{(1)}_{m_B, 2} = \epsilon \,, \\
\mathcal{O}_{e_G} &= \mathcal{O}_{e_{RG}} = \mathcal{O}^{(2)}_{m_B, 1} = \mathcal{O}^{(2)}_{m_B, 2} = \mu \,,
\end{aligned}
\tag{D.18}
$$

where, for $\mathcal{O}^{(1)}_{m_B, 2}$, we used that in the SPT phase $P \equiv 1$. $\mu$ is a string order parameter, however, since in the SPT phase all $\text{Rep}(D_8)$ generators are trivial, the image of $\mu$ in the IR is the identity local operator at the end of the $P$ line:

$$
\mathcal{O}_{e_G} = \mathcal{O}_{e_{RG}} = \mathcal{O}^{(2)}_{m_B, 1} = \mathcal{O}^{(2)}_{m_B, 2} \equiv 1 \,. \tag{D.19}
$$

The $\mathbb{Z}_2$ SSB phase is instead obtained by closing the club quiche with $\mathcal{L} = 1 \oplus e$, which, along with equations (D.13) and (D.14), implies that the local order parameters for the $\mathbb{Z}_2$ SSB phase are:

$$
\begin{aligned}
\mathcal{O}_1 &= \mathcal{O}_{e_R} = \mathcal{O}^{(1)}_{m_B, 1} = \mathcal{O}^{(1)}_{m_B, 2} = \epsilon \,, \\
\mathcal{O}_{m_G, 1} &= \mathcal{O}_{m_G, 2} = \mathcal{O}_{m_{GB}, 1} = \mathcal{O}_{m_{GB}, 2} = \sigma \,.
\end{aligned}
\tag{D.20}
$$

The IR images of these operators follow from equations (D.8) and (D.10):

$$
\begin{aligned}
\mathcal{O}_1 &= \mathcal{O}_{e_R} = \mathcal{O}^{(1)}_{m_B, 1} = \mathcal{O}^{(1)}_{m_B, 2} \equiv v_0 + v_1 \,, \\
\mathcal{O}_{m_G, 1} &= \mathcal{O}_{m_G, 2} = \mathcal{O}_{m_{GB}, 1} = \mathcal{O}_{m_{GB}, 2} \equiv v_0 - v_1 \,.
\end{aligned}
\tag{D.21}
$$

### D.1.3 KT from $\mathbb{Z}_2^\omega$ to $\text{Rep}(D_8)$ symmetry

Let us discuss the KT transformation associated to the condensable algebra $\mathcal{A}_8$ of the igSPT, whose properties we described in subsection C.2.3. In this case we are not describing any phase transition as the igSPT has only a single gapped phase it can flow to. Nevertheless, it is still interesting to study how operators map from a theory with anomalous group symmetry to a theory with non-invertible symmetry. It also allows one to combine two known/existing KT transformations to produce a new KT transformation which may not be known or easy to compute.

From the folded Lagrangian algebra $\mathcal{L}_8$, (B.11) and associated to the condensable algebra $\mathcal{A}_8$, we learn that the map of generalized charges

$$\mathcal{Z}(\text{Vec}_{\mathbb{Z}_2}^\omega) \to \mathcal{Z}(\text{Rep}(D_8)) \tag{D.22}$$

is the following:

$$\begin{aligned}
1 &\to 1 \oplus e_{RG} \oplus e_{RB} \oplus e_{GB}\,, & s\bar{s} &\to e_R \oplus e_{RGB} \oplus e_G \oplus e_B\,, \\
s &\to 2s_{RGB}\,, & \bar{s} &\to 2\bar{s}_{RGB}\,.
\end{aligned} \tag{D.23}$$

According to this map, one finds the following operators:

- An untwisted operator $\mathcal{O}_1'$ of $\mathfrak{T}^{\mathcal{S}'}$ uncharged under $\mathcal{S}' = \mathbb{Z}_2^\omega$ descends to four operators of $\mathfrak{T}^{\mathcal{S}}$: an untwisted operator $\mathcal{O}_1$ and an $RG$-twisted operator $\mathcal{O}_{e_{RG}}$ which are uncharged under $\text{Rep}(D_8)$, and a $G$-twisted operator $\mathcal{O}_{e_{RB}}$ and an $R$-twisted operator $\mathcal{O}_{e_{GB}}$ which are both charged under $B \in \text{Rep}(D_8)$:

$$\begin{aligned}
\mathcal{O}_1 &= \mathcal{O}_1'\,, & \mathcal{O}_{e_{RG}} &= \mathcal{O}_1'\,, \\
\mathcal{O}_{e_{RB}} &= \mathcal{O}_1'\,, & \mathcal{O}_{e_{GB}} &= \mathcal{O}_1'\,,
\end{aligned} \tag{D.24}$$

where $\mathcal{O}_{e_{RG}}$ is the operator $\mathcal{O}_1'$ living at the end of $\phi(RG) = 1$, while $\mathcal{O}_{e_{RB}}$ and $\mathcal{O}_{e_{GB}}$ are operators $\mathcal{O}_1'$ living at the end of $\phi(G) = 1$ and $\phi(R) = 1$ respectively. The fact that $\mathcal{O}_{e_{RB}}$ and $\mathcal{O}_{e_{GB}}$ are charged under $\phi(B) = P' \oplus P'$ can be traced to the KT map in [10]:

$$\mathcal{Z}(\text{Vec}_{\mathbb{Z}_2}^\omega) \to \mathcal{Z}(\text{Vec}_{\mathbb{Z}_4})\,. \tag{D.25}$$

Effectively by taking the KT transformation we have already found in (D.31) for

$$\mathcal{Z}(\text{Vec}_{\mathbb{Z}_4}) \to \mathcal{Z}(\text{Rep}(D_8))\,, \tag{D.26}$$

one can combine two known KTs to produce the KT which we are after

$$\mathcal{Z}(\text{Vec}_{\mathbb{Z}_2}^\omega) \to \mathcal{Z}(\text{Vec}_{\mathbb{Z}_4}) \to \mathcal{Z}(\text{Rep}(D_8))\,. \tag{D.27}$$

From this one finds that while $RG$ is regarded as 1 in $\mathbb{Z}_4$, $R$ and $G$ as $P^2$ and $B$ as $P \oplus P^3$, both $\mathcal{O}_{e_{RB}}$ and $\mathcal{O}_{e_{GB}}$ are identified with $\mathcal{O}_{e^2 m^2}$ at the end of $P^2$. This then gives rise to non-trivial $\mathbb{Z}_4$ junctions characteristic of the anomalous phase even though $\mathcal{O}'$ on its own is not uncharged under $P'$:

$$\tag{D.28}$$

- An untwisted operator $\mathcal{O}'_{s\bar{s}}$ of $\mathfrak{T}^{\mathcal{S}'}$ with charge 1 under $\mathcal{S}' = \mathbb{Z}_2^\omega$ descends to four operators of $\mathfrak{T}^{\mathcal{S}}$: a $G$-twisted operator $\mathcal{O}_{e_G}$ and an $R$-twisted operator $\mathcal{O}_{e_R}$ which are uncharged under $\text{Rep}(D_8)$, and a untwisted operator $\mathcal{O}_{e_{RGB}}$ and an $RG$-twisted operator $\mathcal{O}_{e_B}$ which are both charged under $B \in \text{Rep}(D_8)$:

$$
\begin{aligned}
\mathcal{O}_{e_G} &= \mathcal{O}'_{s\bar{s}}, & \mathcal{O}_{e_R} &= \mathcal{O}'_{s\bar{s}}, \\
\mathcal{O}_{e_{RGB}} &= \mathcal{O}'_{s\bar{s}}, & \mathcal{O}_{e_B} &= \mathcal{O}'_{s\bar{s}},
\end{aligned}
\tag{D.29}
$$

where $\mathcal{O}_{e_B}$ is the operator $\mathcal{O}'_{s\bar{s}}$ living at the end of $\phi(RG) = 1$, while $\mathcal{O}_{e_G}$ and $\mathcal{O}_{e_R}$ are operators $\mathcal{O}'_{s\bar{s}}$ living at the end of $\phi(G) = 1$ and $\phi(R) = 1$ respectively. Now we see as $\mathcal{O}'_{s\bar{s}}$ is charged under $P'$ that both $\mathcal{O}_{e_{RGB}}$ and $\mathcal{O}_{e_B}$ will be charged under $\phi(B) = P' \oplus P'$. On the other hand, $\mathcal{O}_{e_G}$ and $\mathcal{O}_{e_R}$

- A $P'$-twisted operator $\mathcal{O}'_s$ of $\mathfrak{T}^{\mathcal{S}'}$ with charge $\frac{1}{2}$ under $\mathcal{S}' = \mathbb{Z}_2^\omega$ descends to four operators of $\mathfrak{T}^{\mathcal{S}}$, all transforming in the $s_{RGB}$ multiplet under $\text{Rep}(D_8)$: $B$-twisted operators $\mathcal{O}^{(1)}_{s_{RGB},i}$ and $\mathcal{O}^{(2)}_{s_{RGB},j}$ for $i, j = 1, 2$. These operators can be seen as

$$
\begin{aligned}
\mathcal{O}^{(1)}_{s_{RGB},1} &= \mathcal{O}'_s, & \mathcal{O}^{(1)}_{s_{RGB},2} &= \mathcal{O}'_s, \\
\mathcal{O}^{(2)}_{s_{RGB},1} &= \mathcal{O}'_s, & \mathcal{O}^{(2)}_{s_{RGB},2} &= \mathcal{O}'_s,
\end{aligned}
\tag{D.30}
$$

where all $\mathcal{O}^{(1)}_{s_{RGB},i}$ and $\mathcal{O}^{(2)}_{s_{RGB},j}$ should be regarded as operator $\mathcal{O}'_s$ living at the end of line $\phi(B) = P' \oplus P'$. Similarly to (D.28), the presence of non-trivial junctions ensures that all operators above are charged under $R$ and $G$, even though $\phi(R) = 1$ and $\phi(G) = 1$, while the action of $B$ exchanges $\mathcal{O}^{(1)}_{s_{RGB},1}$ and $\mathcal{O}^{(1)}_{s_{RGB},2}$, and similarly for $\mathcal{O}^{(2)}_{s_{RGB},j}$.

- Finally, very analogously a $P'$-twisted operator $\mathcal{O}'_{\bar{s}}$ of $\mathfrak{T}^{\mathcal{S}'}$ with charge $\frac{1}{2}$ under $\mathcal{S}' = \mathbb{Z}_2^\omega$ descends to four operators of $\mathfrak{T}^{\mathcal{S}}$, all transforming in the $\bar{s}_{RGB}$ multiplet under $\text{Rep}(D_8)$.

## D.2 Condensable algebras of dimension 2

Here we provide a few examples of KT transformations from $\mathbb{Z}_2 \times \mathbb{Z}_2$ to $\text{Rep}(D_8)$ as well as the one example of KT transformation from $\mathbb{Z}_4$ to $\text{Rep}(D_8)$. The input phase transition for $\mathbb{Z}_2 \times \mathbb{Z}_2$ symmetry is simply an Ising $\otimes$ Ising CFT whereas for $\mathbb{Z}_4$ it is the 4-State Potts model. Notably for the $\mathbb{Z}_2 \times \mathbb{Z}_2$ case, we show the KT transformation describing the $\text{Rep}(D_8)/(\mathbb{Z}_2 \times \mathbb{Z}_2)$ gSSB phase which can be used as a guide for finding other complicated phase transitions within the $\text{Rep}(D_8)$ system.

### D.2.1 KT from $\mathbb{Z}_4$ to $\text{Rep}(D_8)$ symmetry

From the expression for the condensable algebra $\mathcal{A}_1$ in (B.3) (and analogously $\mathcal{A}_2$ and $\mathcal{A}_3$) we observe that the map

$$
\mathcal{Z}(\text{Vec}_{\mathbb{Z}_4}) \to \mathcal{Z}(\text{Rep}(D_8))
\tag{D.31}
$$

of generalized charges is

$$
\begin{aligned}
1 &\to 1 \oplus e_{RG}, & e^2 &\to e_B \oplus e_{RGB}, \\
m^2 &\to e_R \oplus e_G, & e^2 m^2 &\to e_{GB} \oplus e_{RB}, \\
e, e^3 &\to m_{RG}, & m, m^3 &\to m_B, \\
e m^2, e^3 m^2 &\to f_{RG}, & e^2 m, e^2 m^3 &\to f_B, \\
e m, e^3 m^3 &\to s_{RGB}, & e^3 m, e m^3 &\to \bar{s}_{RGB}.
\end{aligned}
\tag{D.32}
$$

According to this map, one finds the following operators:

- An untwisted operator $\mathcal{O}'_1$ of $\mathfrak{T}^{\mathcal{S}'}$ uncharged under $\mathbb{Z}_4$ descends to two operators of $\mathfrak{T}^{\mathcal{S}}$ that are uncharged under $\text{Rep}(D_8)$: an untwisted operator $\mathcal{O}_1$ and an $RG$-twisted operator $\mathcal{O}_{e_{RG}}$. It is then straightforward to recognize these operators as

$$\mathcal{O}_1 = \mathcal{O}'_1, \qquad \mathcal{O}_{e_{RG}} = \mathcal{O}'_1, \tag{D.33}$$

where $\mathcal{O}_{e_{RG}}$ is the operator $\mathcal{O}'_1$ viewed as sitting at the end of $\phi(RG) = 1$.

- An untwisted operator $\mathcal{O}'_{e^2}$ of $\mathfrak{T}^{\mathcal{S}'}$ with charge 2 under $\mathbb{Z}_4$ descends to two operators of $\mathfrak{T}^{\mathcal{S}}$ charged under $B \in \text{Rep}(D_8)$: an untwisted operator $\mathcal{O}_{e_{RGB}}$ and an $RG$-twisted operator $\mathcal{O}_{e_B}$. These operators can be recognized as

$$\mathcal{O}_{e_{RGB}} = \mathcal{O}'_{e^2}, \qquad \mathcal{O}_{e_B} = \mathcal{O}'_{e^2}, \tag{D.34}$$

where $\mathcal{O}_{e_B}$ is the operator $\mathcal{O}'_{e^2}$ viewed as sitting at the end of $\phi(RG) = 1$.

- An operator $\mathcal{O}'_{m^2}$ in $P^2$-twisted sector of $\mathfrak{T}^{\mathcal{S}'}$ and uncharged under $\mathbb{Z}_4$ descends to two operators of $\mathfrak{T}^{\mathcal{S}}$ that are uncharged under $\text{Rep}(D_8)$: an $R$-twisted operator $\mathcal{O}_{e_R}$ and a $G$-twisted operator $\mathcal{O}_{e_G}$. We can recognize these operators as

$$\mathcal{O}_{e_R} = \mathcal{O}'_{m^2}, \qquad \mathcal{O}_{e_G} = \mathcal{O}'_{m^2}, \tag{D.35}$$

where $\mathcal{O}_{e_R}$ and $\mathcal{O}_{e_G}$ are both the operator $\mathcal{O}'_{m^2}$ viewed as sitting at the end of $\phi(R) = P^2$ and $\phi(G) = P^2$ respectively.

- An operator $\mathcal{O}'_{e^2 m^2}$ in $P^2$-twisted sector of $\mathfrak{T}^{\mathcal{S}'}$ and with charge 2 under $\mathbb{Z}_4$ descends to two operators of $\mathfrak{T}^{\mathcal{S}}$ that are charged under $B \in \text{Rep}(D_8)$: an $R$-twisted operator $\mathcal{O}_{e_{GB}}$ and a $G$-twisted operator $\mathcal{O}_{e_{RB}}$. We can recognize these operators as

$$\mathcal{O}_{e_{GB}} = \mathcal{O}'_{e^2 m^2}, \qquad \mathcal{O}_{e_{RB}} = \mathcal{O}'_{e^2 m^2}, \tag{D.36}$$

where $\mathcal{O}_{e_{GB}}$ and $\mathcal{O}_{e_{RB}}$ are both the operator $\mathcal{O}'_{e^2 m^2}$ viewed as sitting at the end of $\phi(R) = P^2$ and $\phi(G) = P^2$ respectively.

- An untwisted operator $\mathcal{O}'_{e^k}$ of $\mathfrak{T}^{\mathcal{S}'}$ with charge $k = 1$ or $k = 3$ under $\mathbb{Z}_4$ descends to two classes of operators of $\mathfrak{T}^{\mathcal{S}}$, all charged under $R, G, B \in \text{Rep}(D_8)$: an untwisted operator $\mathcal{O}^{(e^k)}_{m_{RG},1}$ and an $RG$-twisted operator $\mathcal{O}^{(e^k)}_{m_{RG},2}$. These operators can be recognized as

$$\begin{aligned}
\mathcal{O}^{(e)}_{m_{RG},1} &= \mathcal{O}'_e, & \mathcal{O}^{(e)}_{m_{RG},2} &= \mathcal{O}'_e, \\
\mathcal{O}^{(e^3)}_{m_{RG},1} &= \mathcal{O}'_{e^3}, & \mathcal{O}^{(e^3)}_{m_{RG},2} &= \mathcal{O}'_{e^3},
\end{aligned} \tag{D.37}$$

where $\mathcal{O}^{(e^k)}_{m_{RG},2}$ for $k = 1, 3$ is the operator $\mathcal{O}'_{e^k}$ viewed as sitting at the end of $\phi(RG) = 1$.

- A $P^k$-twisted operator $\mathcal{O}'_{m^k}$ of $\mathfrak{T}^{\mathcal{S}'}$ with $k = 1, 3$ descends to two classes of operators of $\mathfrak{T}^{\mathcal{S}}$, all uncharged under $\text{Rep}(D_8)$: $B$-twisted operators $\mathcal{O}^{(m^k)}_{m_B,1}$ and $\mathcal{O}^{(m^k)}_{m_B,2}$. These operators can be recognized as

$$\begin{aligned}
\mathcal{O}^{(m)}_{m_B,1} &= \mathcal{O}'_m, & \mathcal{O}^{(m)}_{m_B,2} &= \mathcal{O}'_m, \\
\mathcal{O}^{(m^3)}_{m_B,1} &= \mathcal{O}'_{m^3}, & \mathcal{O}^{(m^3)}_{m_B,2} &= \mathcal{O}'_{m^3},
\end{aligned} \tag{D.38}$$

where both $\mathcal{O}^{(m^k)}_{m_B,1}$ and $\mathcal{O}^{(m^k)}_{m_B,2}$ for $k = 1, 3$ are the operator $\mathcal{O}'_{m^k}$ viewed as sitting at the end of $\phi(B) = P \oplus P^3$ which includes $P^k$.

- A $P^2$-twisted operator $\mathcal{O}'_{e^k m^2}$ of $\mathfrak{T}^{\mathcal{S}'}$ with charge $k = 1$ or $k = 3$ under $\mathbb{Z}_4$ descends to two classes of operators of $\mathfrak{T}^{\mathcal{S}}$, all charged under $R, G, B \in \mathrm{Rep}(D_8)$: an $R$-twisted operator $\mathcal{O}^{(e^k m^2)}_{f_{RG},1}$ and a $G$-twisted operator $\mathcal{O}^{(e^k m^2)}_{f_{RG},2}$. These operators can be recognized as

$$
\begin{aligned}
\mathcal{O}^{(em^2)}_{f_{RG},1} &= \mathcal{O}'_{em^2}, & \mathcal{O}^{(em^2)}_{f_{RG},2} &= \mathcal{O}'_{em^2}, \\
\mathcal{O}^{(e^3 m^2)}_{f_{RG},1} &= \mathcal{O}'_{e^3 m^2}, & \mathcal{O}^{(e^3 m^2)}_{f_{RG},2} &= \mathcal{O}'_{e^3 m^2},
\end{aligned}
\tag{D.39}
$$

where $\mathcal{O}^{(e^k m^2)}_{f_{RG},1}$ and $\mathcal{O}^{(e^k m^2)}_{f_{RG},2}$ for $k = 1, 3$ are the operators $\mathcal{O}'_{e^k m^2}$ viewed as sitting at the end of $\phi(R) = P^2$ and $\phi(G) = P^2$ respectively.

- A $P^k$-twisted operator $\mathcal{O}'_{e^2 m^k}$ of $\mathfrak{T}^{\mathcal{S}'}$ with $k = 1, 3$ and charge 2 under $\mathbb{Z}_4$ descends to two classes of operators of $\mathfrak{T}^{\mathcal{S}}$, all charged under $B \in \mathrm{Rep}(D_8)$: $B$-twisted operators $\mathcal{O}^{(e^2 m^k)}_{f_B,1}$ and $\mathcal{O}^{(e^2 m^k)}_{f_B,2}$. These operators can be recognized as

$$
\begin{aligned}
\mathcal{O}^{(e^2 m)}_{f_B,1} &= \mathcal{O}'_{e^2 m}, & \mathcal{O}^{(e^2 m)}_{f_B,2} &= \mathcal{O}'_{e^2 m}, \\
\mathcal{O}^{(e^2 m^3)}_{f_B,1} &= \mathcal{O}'_{e^2 m^3}, & \mathcal{O}^{(e^2 m^3)}_{f_B,2} &= \mathcal{O}'_{e^2 m^3},
\end{aligned}
\tag{D.40}
$$

where $\mathcal{O}^{(e^2 m^k)}_{f_B,1}$ and $\mathcal{O}^{(e^2 m^k)}_{f_B,2}$ for $k = 1, 3$ are the operator $\mathcal{O}'_{e^2 m^k}$ viewed as sitting at the end of $\phi(B) = P \oplus P^3$.

- A $P^k$-twisted operator $\mathcal{O}'_{e^k m^k}$ of $\mathfrak{T}^{\mathcal{S}'}$ with charge $k = 1, 3$ under $\mathbb{Z}_4$ descends to two classes of operators of $\mathfrak{T}^{\mathcal{S}}$, all charged under $R, G, B \in \mathrm{Rep}(D_8)$: $B$-twisted operators $\mathcal{O}^{(e^k m^k)}_{s_{RGB},1}$ and $\mathcal{O}^{(e^k m^k)}_{s_{RGB},2}$. These operators can be recognized as

$$
\begin{aligned}
\mathcal{O}^{(em)}_{s_{RGB},1} &= \mathcal{O}'_{em}, & \mathcal{O}^{(em)}_{s_{RGB},2} &= \mathcal{O}'_{em}, \\
\mathcal{O}^{(e^3 m^3)}_{s_{RGB},1} &= \mathcal{O}'_{e^3 m^3}, & \mathcal{O}^{(e^3 m^3)}_{s_{RGB},2} &= \mathcal{O}'_{e^3 m^3},
\end{aligned}
\tag{D.41}
$$

where $\mathcal{O}^{(e^k m^k)}_{s_{RGB},1}$ and $\mathcal{O}^{(e^k m^k)}_{s_{RGB},2}$ for $k = 1, 3$ are the operator $\mathcal{O}'_{e^k m^k}$ viewed as sitting at the end of $\phi(B) = P \oplus P^3$.

- Finally, there is the $P$-twisted operator $\mathcal{O}'_{e^3 m}$ with charge 3 under $\mathbb{Z}_4$ and the $P^3$-twisted operator $\mathcal{O}'_{e m^3}$ with charge 1 under $\mathbb{Z}_4$ which descend to operators of $\mathfrak{T}^{\mathcal{S}}$, all charged under $R, G, B \in \mathrm{Rep}(D_8)$: the $B$-twisted operators $\mathcal{O}^{(e^3 m)}_{\bar{s}_{RGB},i}$ and $\mathcal{O}^{(em^3)}_{\bar{s}_{RGB},j}$ for $i, j = 1, 2$. These operators can be recognized as

$$
\begin{aligned}
\mathcal{O}^{(e^3 m)}_{\bar{s}_{RGB},1} &= \mathcal{O}'_{e^3 m}, & \mathcal{O}^{(e^3 m)}_{\bar{s}_{RGB},2} &= \mathcal{O}'_{e^3 m}, \\
\mathcal{O}^{(em^3)}_{\bar{s}_{RGB},1} &= \mathcal{O}'_{em^3}, & \mathcal{O}^{(em^3)}_{\bar{s}_{RGB},2} &= \mathcal{O}'_{em^3},
\end{aligned}
\tag{D.42}
$$

where $\mathcal{O}^{(e^3 m)}_{\bar{s}_{RGB},i}$ and $\mathcal{O}^{(em^3)}_{\bar{s}_{RGB},j}$ for $i, j = 1, 2$ are the operator $\mathcal{O}'_{e^3 m}$ and $\mathcal{O}'_{em^3}$ respectively viewed as sitting at the end of $\phi(B) = P \oplus P^3$.

### D.2.2 KT from $\mathbb{Z}_2 \times \mathbb{Z}_2$ to $\mathrm{Rep}(D_8)$ symmetry

From the expression for the condensable algebra $\mathcal{A}_4$ in (B.7) (and analogously $\mathcal{A}_5$ and $\mathcal{A}_6$) we observe that the map

$$
\mathcal{Z}(\mathrm{Vec}_{\mathbb{Z}_2 \times \mathbb{Z}_2}) \to \mathcal{Z}(\mathrm{Rep}(D_8))
\tag{D.43}
$$

of generalized charges is

$$
\begin{aligned}
1 &\to 1 \oplus e_R \,, \\
e_1 e_2 &\to e_{GB} \oplus e_{RGB} \,, \\
m_1 m_2 &\to e_G \oplus e_{RG} \,, \\
e_1 e_2 m_1 m_2 &\to e_B \oplus e_{RB} \,, \\
e_1, e_2 &\to m_{GB} \,, \\
m_1, m_2 &\to m_B \,, \\
e_1 m_2, e_2 m_1 &\to m_G \,, \\
e_1 m_1 m_2, e_2 m_1 m_2 &\to f_{GB} \,, \\
e_1 e_2 m_1, e_1 e_2 m_2 &\to f_B \,, \\
e_1 m_1, e_2 m_2 &\to f_G \,.
\end{aligned}
\tag{D.44}
$$

According to this map, one finds the following operators:

- An untwisted operator $\mathcal{O}'_1$ of $\mathfrak{T}^{\mathcal{S}'}$ uncharged under $\mathbb{Z}_2 \times \mathbb{Z}_2$ descends to two operators of $\mathfrak{T}^{\mathcal{S}}$ that are uncharged under $\mathrm{Rep}(D_8)$: an untwisted operator $\mathcal{O}_1$ and an $R$-twisted operator $\mathcal{O}_{e_R}$. It is then straightforward to recognize these operators as

$$
\mathcal{O}_1 = \mathcal{O}'_1 \,, \qquad \mathcal{O}_{e_R} = \mathcal{O}'_1 \,,
\tag{D.45}
$$

  where $\mathcal{O}_{e_{RG}}$ is the operator $\mathcal{O}'_1$ viewed as sitting at the end of $\phi(R) = 1$.

- An untwisted operator $\mathcal{O}'_{e_1 e_2}$ of $\mathfrak{T}^{\mathcal{S}'}$ charged under $\mathbb{Z}_2 \times \mathbb{Z}_2$ descends to two operators of $\mathfrak{T}^{\mathcal{S}}$ charged under $B \in \mathrm{Rep}(D_8)$: an untwisted operator $\mathcal{O}_{e_{RGB}}$ and an $R$-twisted operator $\mathcal{O}_{e_{GB}}$. These operators can be recognized as

$$
\mathcal{O}_{e_{RGB}} = \mathcal{O}'_{e_1 e_2} \,, \qquad \mathcal{O}_{e_{GB}} = \mathcal{O}'_{e_1 e_2} \,,
\tag{D.46}
$$

  where $\mathcal{O}_{e_{GB}}$ is the operator $\mathcal{O}'_{e_1 e_2}$ viewed as sitting at the end of $\phi(R) = 1$.

- An operator $\mathcal{O}'_{m_1 m_2}$ in $P_1 P_2$-twisted sector of $\mathfrak{T}^{\mathcal{S}'}$ and uncharged under $\mathbb{Z}_2 \times \mathbb{Z}_2$ descends to two operators of $\mathfrak{T}^{\mathcal{S}}$ that are uncharged under $\mathrm{Rep}(D_8)$: an $G$-twisted operator $\mathcal{O}_{e_G}$ and a $RG$-twisted operator $\mathcal{O}_{e_{RG}}$. We can recognize these operators as

$$
\mathcal{O}_{e_G} = \mathcal{O}'_{m_1 m_2} \,, \qquad \mathcal{O}_{e_{RG}} = \mathcal{O}'_{m_1 m_2} \,,
\tag{D.47}
$$

  where $\mathcal{O}_{e_G}$ and $\mathcal{O}_{e_{RG}}$ are both the operator $\mathcal{O}'_{m_1 m_2}$ viewed as sitting at the end of $\phi(G) = P_1 P_2$ and $\phi(RG) = P_1 P_2$ respectively.

- An operator $\mathcal{O}'_{e_1 e_2 m_1 m_2}$ in $P_1 P_2$-twisted sector of $\mathfrak{T}^{\mathcal{S}'}$ and charged under $\mathbb{Z}_2 \times \mathbb{Z}_2$ descends to two operators of $\mathfrak{T}^{\mathcal{S}}$ that are charged under $B \in \mathrm{Rep}(D_8)$: an $G$-twisted operator $\mathcal{O}_{e_{RB}}$ and a $RG$-twisted operator $\mathcal{O}_{e_B}$. We can recognize these operators as

$$
\mathcal{O}_{e_{RB}} = \mathcal{O}'_{e_1 e_2 m_1 m_2} \,, \qquad \mathcal{O}_{e_B} = \mathcal{O}'_{e_1 e_2 m_1 m_2} \,,
\tag{D.48}
$$

  where $\mathcal{O}_{e_{RB}}$ and $\mathcal{O}_{e_B}$ are both the operator $\mathcal{O}'_{e_1 e_2 m_1 m_2}$ viewed as sitting at the end of $\phi(G) = P_1 P_2$ and $\phi(RG) = P_1 P_2$ respectively.

- An untwisted operator $\mathcal{O}'_{e_k}$ of $\mathfrak{T}^{\mathcal{S}'}$ with $k = 1$ or $k = 2$ charged under $\mathbb{Z}_2 \times \mathbb{Z}_2$ descends to two classes of operators of $\mathfrak{T}^{\mathcal{S}}$, all charged under $R, G, B \in \mathrm{Rep}(D_8)$: an untwisted

operator $\mathcal{O}^{(e_k)}_{m_{GB},1}$ and an $R$-twisted operator $\mathcal{O}^{(e_k)}_{m_{GB},2}$. These operators can be recognized as

$$
\begin{aligned}
\mathcal{O}^{(e_1)}_{m_{GB},1} &= \mathcal{O}'_{e_1}, & \mathcal{O}^{(e_1)}_{m_{GB},2} &= \mathcal{O}'_{e_1}, \\
\mathcal{O}^{(e_2)}_{m_{GB},1} &= \mathcal{O}'_{e_2}, & \mathcal{O}^{(e_2)}_{m_{GB},2} &= \mathcal{O}'_{e_2},
\end{aligned}
\tag{D.49}
$$

where $\mathcal{O}^{(e_k)}_{m_{GB},2}$ for $k = 1,2$ is the operator $\mathcal{O}'_{e_k}$ viewed as sitting at the end of $\phi(R) = 1$.

- A $P_k$-twisted operator $\mathcal{O}'_{m_k}$ of $\mathfrak{T}^{\mathcal{S}'}$ with $k = 1,2$ descends to two classes of operators of $\mathfrak{T}^{\mathcal{S}}$, all uncharged under $\mathrm{Rep}(D_8)$: $B$-twisted operators $\mathcal{O}^{(m_k)}_{m_B,1}$ and $\mathcal{O}^{(m_k)}_{m_B,2}$. These operators can be recognized as

$$
\begin{aligned}
\mathcal{O}^{(m_1)}_{m_B,1} &= \mathcal{O}'_{m_1}, & \mathcal{O}^{(m_1)}_{m_B,2} &= \mathcal{O}'_{m_1}, \\
\mathcal{O}^{(m_2)}_{m_B,1} &= \mathcal{O}'_{m_2}, & \mathcal{O}^{(m_2)}_{m_B,2} &= \mathcal{O}'_{m_2},
\end{aligned}
\tag{D.50}
$$

where $\mathcal{O}^{(m_k)}_{m_B,1}$ and $\mathcal{O}^{(m_k)}_{m_B,2}$ for $k = 1,2$ are the operator $\mathcal{O}'_{m_k}$ viewed as sitting at the end of $P^k \supset \phi(B) = P_1 \oplus P_2$.

- A $P_2$-twisted operator $\mathcal{O}'_{e_1 m_2}$ and $P_1$-twisted operator $\mathcal{O}'_{e_2 m_1}$ of $\mathfrak{T}^{\mathcal{S}'}$ charged under $\mathbb{Z}_2 \times \mathbb{Z}_2$ descend to two classes of operators of $\mathfrak{T}^{\mathcal{S}}$, all charged under $G, B \in \mathrm{Rep}(D_8)$: $B$-twisted operators $\mathcal{O}^{(e_1 m_2)}_{m_G,i}$ and $\mathcal{O}^{(e_2 m_1)}_{m_G,j}$ for $i,j = 1,2$. These operators can be recognized as

$$
\begin{aligned}
\mathcal{O}^{(e_1 m_2)}_{m_G,1} &= \mathcal{O}'_{e_1 m_2}, & \mathcal{O}^{(e_1 m_2)}_{m_G,2} &= \mathcal{O}'_{e_1 m_2}, \\
\mathcal{O}^{(e_2 m_1)}_{m_G,1} &= \mathcal{O}'_{e_2 m_1}, & \mathcal{O}^{(e_2 m_1)}_{m_G,2} &= \mathcal{O}'_{e_2 m_1},
\end{aligned}
\tag{D.51}
$$

where $\mathcal{O}^{(e_1 m_2)}_{m_G,i}$ and $\mathcal{O}^{(e_2 m_1)}_{m_G,j}$ are the operators $\mathcal{O}'_{e_1 m_2}$ and $\mathcal{O}'_{e_2 m_1}$ respectively viewed as sitting at the end of $\phi(B) \supset P_2$ and $\phi(B) \supset P_1$ respectively.

- A $P_1 P_2$-twisted operator $\mathcal{O}'_{e_k m_1 m_2}$ of $\mathfrak{T}^{\mathcal{S}'}$ with charged under $\mathbb{Z}_2 \times \mathbb{Z}_2$ descends to two classes of operators of $\mathfrak{T}^{\mathcal{S}}$, all charged under $G, B \in \mathrm{Rep}(D_8)$: a $G$-twisted operator $\mathcal{O}^{(e_k m_1 m_2)}_{f_{GB},1}$ and a $RG$-twisted operator $\mathcal{O}^{(e_k m_1 m_2)}_{f_{GB},2}$. These operators can be recognized as

$$
\mathcal{O}^{(e_k m_1 m_2)}_{f_{GB},1} = \mathcal{O}'_{e_k m_1 m_2}, \qquad \mathcal{O}^{(e_k m_1 m_2)}_{f_{GB},2} = \mathcal{O}'_{e_k m_1 m_2},
\tag{D.52}
$$

where $\mathcal{O}^{(e_k m_1 m_2)}_{f_{GB},1}$ and $\mathcal{O}^{(e_k m_1 m_2)}_{f_{GB},2}$ for $k = 1,2$ are the operators $\mathcal{O}'_{e_k m_1 m_2}$ viewed as sitting at the end of $\phi(G) = P_1 P_2$ and $\phi(RG) = P_1 P_2$ respectively.

- A $P_k$-twisted operator $\mathcal{O}'_{e_1 e_2 m_k}$ of $\mathfrak{T}^{\mathcal{S}'}$ with $k = 1,2$ charged under $\mathbb{Z}_2 \times \mathbb{Z}_2$ descends to two classes of operators of $\mathfrak{T}^{\mathcal{S}}$, charged under $B \in \mathrm{Rep}(D_8)$: $B$-twisted operators $\mathcal{O}^{(e_1 e_2 m_k)}_{f_B,1}$ and $\mathcal{O}^{(e_1 e_2 m_k)}_{f_B,2}$. These operators can be recognized as

$$
\begin{aligned}
\mathcal{O}^{(e_1 e_2 m_1)}_{f_B,1} &= \mathcal{O}'_{e_1 e_2 m_1}, & \mathcal{O}^{(e_1 e_2 m_1)}_{f_B,2} &= \mathcal{O}'_{e_1 e_2 m_1}, \\
\mathcal{O}^{(e_1 e_2 m_2)}_{f_B,1} &= \mathcal{O}'_{e_1 e_2 m_2}, & \mathcal{O}^{(e_1 e_2 m_2)}_{f_B,2} &= \mathcal{O}'_{e_1 e_2 m_2},
\end{aligned}
\tag{D.53}
$$

where $\mathcal{O}^{(e_1 e_2 m_k)}_{f_B,1}$ and $\mathcal{O}^{(e_1 e_2 m_k)}_{f_B,2}$ for $k = 1,2$ are the operator $\mathcal{O}'_{e_1 e_2 m_k}$ viewed as sitting at the end of $\phi(B) = P_1 \oplus P_2 \supset P_k$.

- A $P_1$-twisted operator $\mathcal{O}'_{e_1 m_1}$ and $P_2$-twisted operator $\mathcal{O}'_{e_2 m_2}$ of $\mathfrak{T}^{\mathcal{S}'}$ charged under $\mathbb{Z}_2 \times \mathbb{Z}_2$ descend to two classes of operators of $\mathfrak{T}^{\mathcal{S}}$, all charged under $G, B \in \mathrm{Rep}(D_8)$: $B$-twisted operators $\mathcal{O}^{(e_1 m_1)}_{f_G,i}$ and $\mathcal{O}^{(e_2 m_2)}_{f_G,j}$ with $i, j = 1, 2$. These operators can be recognized as

$$
\begin{aligned}
\mathcal{O}^{(e_1 m_1)}_{f_G,1} &= \mathcal{O}'_{e_1 m_1}, & \mathcal{O}^{(e_1 m_1)}_{f_G,2} &= \mathcal{O}'_{e_1 m_1}, \\
\mathcal{O}^{(e_2 m_2)}_{f_G,1} &= \mathcal{O}'_{e_2 m_2}, & \mathcal{O}^{(e_2 m_2)}_{f_G,2} &= \mathcal{O}'_{e_2 m_2},
\end{aligned}
\tag{D.54}
$$

where $\mathcal{O}^{(e_1 m_1)}_{f_G,i}$ and $\mathcal{O}^{(e_2 m_2)}_{f_G,j}$ are the operators $\mathcal{O}'_{e_1 m_1}$ and $\mathcal{O}'_{e_2 m_2}$ respectively viewed as living at the end of $\phi(B) \supset P_1, P_2$.

### D.2.3 KT from $\mathbb{Z}_2 \times \mathbb{Z}_2$ to $\mathrm{Rep}(D_8)$ symmetry II

From the expression for the condensable algebra $\mathcal{A}_7$ in (B.10) we observe that the map

$$
\mathcal{Z}(\mathrm{Vec}_{\mathbb{Z}_2 \times \mathbb{Z}_2}) \to \mathcal{Z}(\mathrm{Rep}(D_8))
\tag{D.55}
$$

of generalized charges is

$$
\begin{aligned}
1 &\to 1 \oplus e_{RGB}, \\
e_2 m_1 &\to e_R \oplus e_{GB}, \\
e_1 m_2 &\to e_G \oplus e_{RB}, \\
e_1 e_2 m_1 m_2 &\to e_B \oplus e_{RG}, \\
e_1, m_2 &\to m_{RB}, \\
e_2, m_1 &\to m_{GB}, \\
e_1 e_2, m_1 m_2 &\to m_{RG}, \\
e_2 m_1 m_2, e_1 e_2 m_1 &\to f_{RB}, \\
e_1 m_1 m_2, e_1 e_2 m_2 &\to f_{GB}, \\
e_1 m_1, e_2 m_2 &\to f_{RG}.
\end{aligned}
\tag{D.56}
$$

Now one can again write out all the operators and their maps, however, given we have already shown these for $\mathbb{Z}_2 \times \mathbb{Z}_2 \to \mathrm{Rep}(D_8)$ we will only show a few examples of dealing with some complications here while the rest can be computed similarly. The main difficulty here lies in the fact that the SymTFT Quiche will have a reducible symmetry boundary of the form

$$
\mathfrak{B}' = \mathfrak{B}_e \oplus \mathfrak{B}_m,
\tag{D.57}
$$

where $\mathfrak{B}_m = \mathfrak{B}_e/(\mathbb{Z}_2 \times \mathbb{Z}_2)$ mixes two gauge-opposite irreducible boundaries of the SymTFT for $\mathbb{Z}_2^2$ symmetry, with a trivial relative Euler term between the two boundaries.

The $\mathrm{Rep}(D_8)$-symmetric, KT-transformed phase transition theory can be written in terms of an input $\mathbb{Z}_2^2$-symmetric theory $\mathfrak{T}^{\mathbb{Z}_2^2}$ as

$$
\mathfrak{T}^{\mathrm{Rep}(D_8)} = \rho \; \underleftrightarrow{\curvearrowright \underbrace{\mathfrak{T}^{\mathbb{Z}_2^2} \oplus (\mathfrak{T}^{\mathbb{Z}_2}/\mathbb{Z}_2^2)}_{B} \curvearrowleft} \; \rho,
\tag{D.58}
$$

where $\rho = 1, R, G, RG \subset \mathrm{Rep}(D_8)$. In fact, we can recognize $\mathfrak{T}^{\mathbb{Z}_2^2}$ as a product of two copies of a $\mathbb{Z}_2$-symmetric input theory $\mathfrak{T}^{\mathbb{Z}_2}$. For our minimal case, we can take $\mathfrak{T}^{\mathbb{Z}_2}$ to be the 2d Ising CFT, and therefore our $\mathbb{Z}_2^2$ input transition $\mathfrak{T}^{\mathbb{Z}_2^2}$ will simply be $\mathrm{Ising} \otimes \mathrm{Ising} = \mathrm{Ising}^2$.

Now we are in a position to discuss how operators of an input theory map to operators of the KT-transformed theory. For example:

- An untwisted operator $\mathcal{O}'_1$ of $\mathfrak{T}^{\mathbb{Z}_2^2}$ uncharged under $\mathbb{Z}_2 \times \mathbb{Z}_2$ descends to two operators: an untwisted operator $\mathcal{O}_1$ uncharged under $\mathrm{Rep}(D_8)$ and an untwisted operator $\mathcal{O}_{e_{RGB}}$ charged under $B \in \mathrm{Rep}(D_8)$. These operators can be recognized as

$$\mathcal{O}_1 = (\mathcal{O}'_1)_e + (\mathcal{O}'_1)_m, \qquad \mathcal{O}_{e_{RGB}} = (\mathcal{O}'_1)_e - (\mathcal{O}'_1)_m, \tag{D.59}$$

where $(\mathcal{O}'_1)_e$ is a copy of $\mathcal{O}'_1$ in $\mathfrak{T}^e := \mathfrak{T}^{\mathbb{Z}_2^2}$ and $(\mathcal{O}'_1)_m$ is a copy of $\mathcal{O}'_1$ in $\mathfrak{T}^m := \mathfrak{T}^{\mathbb{Z}_2^2}/\mathbb{Z}_2^2$ of (D.58). This is consistent with the (passing-through) symmetry action seen in (C.20) as

$$S_{em} : (\mathcal{O}'_1)_e \to (\mathcal{O}'_1)_m, \qquad S_{me} : (\mathcal{O}'_1)_m \to (\mathcal{O}'_1)_e, \tag{D.60}$$

while $\mathcal{O}'_1$ is invariant under $P_1$ and $P_2$, hence the operators $\mathcal{O}_1$ and $\mathcal{O}_{e_{RGB}}$ are uncharged under $R$ and $G$ but under $\phi(B) = S_{em} \oplus S_{me}$ one finds

$$\phi(B) : \mathcal{O}_1 \to 2\mathcal{O}_1, \qquad \mathcal{O}_{e_{RGB}} \to -2\mathcal{O}_{e_{RGB}}, \tag{D.61}$$

as expected.

- A $P_2$-twisted operator $\mathcal{O}'_{e_1 m_2}$ of $\mathfrak{T}^{\mathbb{Z}_2^2}$ with charge $(1,0)$ under $\mathbb{Z}_2 \times \mathbb{Z}_2$ descends to two operators of $\mathfrak{T}^{\mathrm{Rep}(D_8)}$: a $G$-twisted operator $\mathcal{O}_{e_G}$ uncharged under $\mathrm{Rep}(D_8)$ and a $G$-twisted operator $\mathcal{O}_{e_{RB}}$ charged under $B \in \mathrm{Rep}(D_8)$. These operators can be recognized as

$$\begin{aligned} \mathcal{O}_{e_G} &= (\mathcal{O}'_{e_1 m_2})_e + (\mathcal{O}'_{e_1 m_2})_m, \\ \mathcal{O}_{e_{RB}} &= (\mathcal{O}'_{e_1 m_2})_e - (\mathcal{O}'_{e_1 m_2})_m, \end{aligned} \tag{D.62}$$

where $(\mathcal{O}'_{e_1 m_2})_e$ is a copy of $\mathcal{O}'_{e_1 m_2}$ in $\mathfrak{T}^e$ which is $(P_2)_{ee}$-twisted and $(\mathcal{O}'_{e_1 m_2})_m$ is a copy of $\mathcal{O}'_{e_1 m_2}$ in $\mathfrak{T}^m$ which is $(P_1)_{mm}$-twisted after gauging of the $\mathbb{Z}_2^2$ symmetry. Both operators can thus be viewed as sitting at the end of $\phi(G) \supset (P_2)_{ee}, (P_1)_{mm}$ as required. This is consistent with the $B$ symmetry action (and hence also the linking action) seen in (C.20) as

$$\begin{aligned} S_{em} &: (\mathcal{O}'_{e_1 m_2})_e \to (\mathcal{O}'_{e_1 m_2})_m, \\ S_{me} &: (\mathcal{O}'_{e_1 m_2})_m \to (\mathcal{O}'_{e_1 m_2})_e, \end{aligned} \tag{D.63}$$

however, we also require $\mathcal{O}_{e_G}$ and $\mathcal{O}_{e_{RB}}$ to be invariant under $\phi(R)$ and $\phi(G)$, while in its current form it would seem they are both charged under $\phi(R)$ and $\phi(RG)$. To remedy this, one chooses a particular non-trivial junction between the $\phi(G)$ and $\phi(R)$ lines to cancel this inconsistency. The presence of such a junction is then important for other operators in the theory to be charged correctly as we will see shortly with the $m_{RB}$ multiplet.

Hence the operators $\mathcal{O}_{e_G}$ and $\mathcal{O}_{e_{RB}}$ are then uncharged under $R$ and $G$ but under $\phi(B) = S_{em} \oplus S_{me}$ one finds the linking action

$$\phi(B) : \mathcal{O}_{e_G} \to 2\mathcal{O}_{e_G}, \qquad \mathcal{O}_{e_{RB}} \to -2\mathcal{O}_{e_{RB}}, \tag{D.64}$$

which is consistent with the results for $\mathcal{O}_1$ and $\mathcal{O}_{e_{RGB}}$ above as $\mathcal{O}_{e_{RB}} = \mathcal{O}_{e_G} \otimes \mathcal{O}_{e_{RGB}}$.

- Finally for a more complicated multiplet, we can pick the untwisted operator $\mathcal{O}'_{e_1}$ of $\mathfrak{T}^{\mathbb{Z}_2^2}$ with charge $(1,0)$ under $\mathbb{Z}_2 \times \mathbb{Z}_2$ which descends to two operators of $\mathfrak{T}^{\mathrm{Rep}(D_8)}$ in the $m_{RB}$ multiplet: an untwisted operator $\mathcal{O}^{e_1}_{m_{RB},1}$ and a $G$-twisted operator $\mathcal{O}^{e_1}_{m_{RB},2}$ both charged under $R, RG, B \in \mathrm{Rep}(D_8)$. These operators can be recognized as

$$\mathcal{O}^{e_1}_{m_{RB},1} = (\mathcal{O}'_{e_1})_e, \qquad \mathcal{O}^{e_1}_{m_{RB},2} = (\mathcal{O}'_{e_1})_m, \tag{D.65}$$

where $(\mathcal{O}'_{e_1})_e$ is a copy of $\mathcal{O}'_{e_1}$ in $\mathfrak{T}^e$ which untwisted and $(\mathcal{O}'_{e_1})_m$ is a copy of $\mathcal{O}'_{e_1}$ in $\mathfrak{T}^m$ which is $(P_1)_{mm}$-twisted after gauging of the $\mathbb{Z}_2^2$ symmetry. Thus $\mathcal{O}^{e_1}_{m_{RB},2}$ should be viewed as an operator living at the end of $\phi(G) \supset (P_1)_{mm}$, while $\mathcal{O}^{e_1}_{m_{RB},1}$ is untwisted as required.

Both $\mathcal{O}^{e_1}_{m_{RB},1}$ and $\mathcal{O}^{e_1}_{m_{RB},2}$ are charged under $R$ but not $G$, while the symmetry action of $B$ exchanges $\mathcal{O}^{e_1}_{m_{RB},1}$ and $\mathcal{O}^{e_1}_{m_{RB},2}$ as $S_{em} : (\mathcal{O}'_{e_1})_e \to (\mathcal{O}'_{e_1})_m$ and similarly for $(\mathcal{O}'_{e_1})_m$. Note that the charge of $\mathcal{O}^{e_1}_{m_{RB},2}$ under $R$ can be obtained from a choice of a non-trivial junction operator between the symmetry line $\phi(R)$ and twist line $\phi(G)$ as was the case for $\mathcal{O}_{e_G}$ and $\mathcal{O}_{e_{RB}}$.

Other examples can be computed similarly.

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
