# Peer review of "Hasse Diagrams for Gapless SPT and SSB Phases with Non-Invertible Symmetries"

_SciPost Physics, doi:SciPost Phys. 19, 113 (2025)_

## Round 1 · Author Response

Report url: https://scipost.org/submissions/2403.00905v3/

We thank both Referees for their detailed reviews and very insightful comments. We reply to reach referee below.

Reply to Referee 1:

We thank the Referee for their many comments, demonstrating a careful reading of our work. We very much appreciate this. We agree on basically all of the changes and questions raised and the implementation can be found below.

Let us first address the requested changes:

1) References [3,4] suggested by the referee were added to [1-2]. In addition to what used to be [6-8], the references [11,12] by Kong et al. were added. Also on page 1, some mathematical references regarding the notions of condensable algebras were added as [14-15] and [16-17]. On page 3, in addition to what used to be [21-23], earlier references on the role of the Drinfel’d centre as a classifying tool for sectors/charges were added as [5-7,19,31-33]. On page 7, an earlier reference than [52] discussing the correspondence between the topological orders was added as [55]. On page 15, when discussing SSB phases, the references mentioned by the Referee [3,4] were highlighted, establishing a direct connection between module categories and gapped phases, also see footnote [60].

2) Footnote [13] was added discussing the assumptions behind the systematic exploration of gapped phases using the SymTFT description.

3) At the top of page 3, a sentence was added clarifying what is meant by the word charge in this paper: we employ ‘generalized charge’ in the sense of [19] for any object in the Drinfeld center. We would like to point out that the case that the referee mentions "fluxes" and "charges" is very specific to the case of group-symmetries. More generally, for a categorical symmetry, the anyons in the Drinfeld center that can end on the physical boundary are precisely the generalized charges -- they are acted upon by the (not necessarily group-like) symmetry generators.

4) On page 4, a clarifying sentence was added to the definition of confined charges. For comments about the pivotal structure we added footnotes [42] and [45]. This should clarify the use of pivotal structures and issues with treatment in higher dimensions.

5) On page 7, around (II.19) we now say that the Drinfeld centers of D8 and Rep(D8) are not the same but equivalent as suggested. We also added a paragraph towards the end of Introduction on page 3, clarifying our use of condensable algebras at the level objects and therefore the possible multiplication structures of the algebras are not considered in this work, hence we decided to omit the formula for group-theoretical condensable algebras suggested by the Referee for simplicity and consistency.

6) A paragraph starting with "Physically, ..." on page 9 was added to explain more physically the relationship between gaplessness and the notion of (de)confined charges.

7) On page 10, footnote [59] was added to explain what is meant by suitable physical boundary of the SymTFT. We clarify how gaplessness comes into the picture via the physical boundary.

8) A sentence was added under (III.13) on page 10 to make it clear what in the pictures is meant as a charge and what is meant as a domain wall.

9) On page 12, at the start of the subsection, references [24,26] were added highlighting the well-known example of (1+1)d igSPT for Z4 symmetry.

10) At the top of page 15, a short paragraph was added to explain the terminology behind the use of Euler terms and how these supersede thinking of purely quantum dimensions in relation to labeling vacua.

11) The paragraph on page 15 starting with "Mathematically, ..." has been changed to avoid the statements of this paragraph being misinterpreted as module categories over the symmetry category which do not in general have the structure of fusion or multi-fusion categories. We also clarified how charges and confined charges correspond to order parameters and excitations under (IV.3) on the following page.

12) On page 17, footnote [63] was added to address potential confusion about gapless domain walls. On the same page, footnote [64] defines the use of the word 'critical' in the present context.

13) A typo was corrected on page 18, we thank the Referee for pointing that out.

14) On page 43, three short paragraphs were added to briefly introduce the concept of a generalized KT transformation. We point the reader to the more recent use of the term in our context in [10] while explaining the connection between the two and justifying the use of the term.

Regarding the pointed out weaknesses:

1) Hopefully we have addressed all the concerns expressed in this point in the changes we have made above including the connection of gaplessness and condensable algebras, including the motivation, e.g. on page 9. We classify these phases using functorial maps so the heavy use of 'maps' is only natural in our context. Finally, please also see top of page 12 where we explain the limitations of the categorical framework we are working with.

2) This point we address in a few places where required changes were made but also in a dedicated paragraph at the bottom of page 3 where we clearly state the limitations of our current approach treating condensable algebras only on the level of objects.

3) We agree some of the original statements were perhaps a bit too dimensionally ambiguous and we have addressed those concerns throughout the requested changes and the work, especially by addressing some points made by Referee 2, which can be found below. Furthermore, in a forthcoming work a subset of the authors compute the algebra multiplication for all group-theoretical cases and confirm that no additional algebras arise for the examples considered in this work. Also we should mention that in fact in the meantime many of the concepts have been extended to 2+1d where although the technical details of the categories involved are more complex (fusion 2-categories), the main ideas do in fact follow through and thus we do not think our comments pointing to higher dimensions are "lacunar".

Regarding the specific questions/remarks:

1) Please see requested change 6) to answer this query and also top of page 12 where there is a short paragraph addressing this issue of the nature of the excitations and the nature of the spectra.

2) We address the pivotal properties in the requested changes above and below in changes for Referee 2.

3) Given the sentence in the abstract "gSSB phases are classified by functors from fusion to multi-fusion categories" we do not necessarily make an equivalance between functors and gSSB phases, rather we use the former to categorize the latter, i.e. we do not necessarily claim that for any functor from fusion to multi-fusion category will necessarily describe a gSSB.

4) As mentioned in requested change 5) we added a paragraph at the bottom of page 3, clarifying our use of condensable algebras at the level objects and therefore the possible multiplication structures of the algebras are not concretely considered in this work, hence we decided to omit such formula for simplicity and consistency.

Reply to Referee 2:

We thank the Referee for their comments. Let us address the requested changes:

1) In the "Generalized Superconductivity Interpretation" on page 3, we removed the previously used term "local" and added explanation of "(Such operators may correspond to extended or disorder operators in the microscopic theory.)", to make it clear what is meant by these operators.

2) In the last paragraph of page 3, we agree our current approach based on the use of condensable algebras on the level of objects (please refer to changes made above, requested by Referee 1, e.g. requested change 5) is not generally enough to capture all the subtleties of condensable algebras including the case highlighted by the Referee with SPTs with a symmetry group of order 128. We addressed the limitations of our current approach in the third paragraph on page 4 and included a reference [43].

3) The issues of the use of pivotal structures is addressed by footnotes [42] and [45] which justify their use, especially the physical justification for it.

4) We agree that symmetry protected criticality becomes a more difficult concept to deal with in higher dimensions than 2 and hence we have reflected on this by extending the paragraph that deals with this criticality to explain the limitations.

5) We have removed the term "set of charges" as indeed this may not be enough to characterize the phases entirely. We believe in 1+1d it is sufficient. But in higher dimensions, additional data, such as symmetry fractionalization, may come into this that require a refinement of this statement.

We hope we have addressed all the comments of the Referees and that the paper can now be recommended for publication.

---

## Editorial Decision

published